# Benign Overfitting in Two-layer Convolutional Neural Networks

**Yuan Cao**[*]
Department of Statistics & Actuarial Science
Department of Mathematics
The University of Hong Kong
yuancao@hku.hk

**Zixiang Chen**[*]
Department of Computer Science
University of California, Los Angeles
Los Angeles, CA 90095, USA
chenzx19@cs.ucla.edu

**Mikhail Belkin**
Haliciolu Data Science Institute
University of California San Diego
La Jolla, CA 92093, USA
mbelkin@ucsd.edu

**Quanquan Gu**
Department of Computer Science
University of California, Los Angeles
Los Angeles, CA 90095, USA
qgu@cs.ucla.edu

## Abstract

Modern neural networks often have great expressive power and can be trained to overfit the training data, while still achieving a good test performance. This phenomenon is referred to as "benign overfitting". Recently, there emerges a line of works studying "benign overfitting" from the theoretical perspective. However, they are limited to linear models or kernel/random feature models, and there is still a lack of theoretical understanding about when and how benign overfitting occurs in neural networks. In this paper, we study the benign overfitting phenomenon in training a two-layer convolutional neural network (CNN). We show that when the signal-to-noise ratio satisfies a certain condition, a two-layer CNN trained by gradient descent can achieve arbitrarily small training and test loss. On the other hand, when this condition does not hold, overfitting becomes harmful and the obtained CNN can only achieve constant level test loss. These together demonstrate a sharp phase transition between benign overfitting and harmful overfitting, driven by the signal-to-noise ratio. To the best of our knowledge, this is the first work that precisely characterizes the conditions under which benign overfitting can occur in training convolutional neural networks.

## 1 Introduction

Modern deep learning models often consist of a huge number of model parameters, which is more than the number of training data points and therefore over-parameterized. These over-parameterized models can be trained to overfit the training data (achieving a close to $100\%$ training accuracy), while still making accurate prediction on the unseen test data. This phenomenon has been observed in a number of prior works (Zhang et al., 2017; Neyshabur et al., 2019), and is often referred to as *benign overfitting* (Bartlett et al., 2020). It revolutionizes the classical understanding about the bias-variance trade-off in statistical learning theory, and has drawn great attention from the community (Belkin et al., 2018, 2019a,b; Hastie et al., 2019).

There exist a number of works towards understanding the benign overfitting phenomenon. While they offered important insights into the benign overfitting phenomenon, most of them are limited

---
[*]Equal contribution.

36th Conference on Neural Information Processing Systems (NeurIPS 2022).

to the settings of linear models (Belkin et al., 2019b; Bartlett et al., 2020; Hastie et al., 2019; Wu and Xu, 2020; Chatterji and Long, 2020; Zou et al., 2021b; Cao et al., 2021) and kernel/random features models (Belkin et al., 2018; Liang and Rakhlin, 2020; Montanari and Zhong, 2020), and cannot be applied to neural network models that are of greater interest. The only notable exceptions are (Adlam and Pennington, 2020; Li et al., 2021), which attempted to understand benign overfitting in neural network models. However, they are still limited to the "neural tagent kernel regime" (Jacot et al., 2018) where the neural network learning problem is essentially equivalent to kernel regression. Thus, it remains a largely open problem to show how and when benign overfitting can occur in neural networks.

Clearly, understanding benign overfitting in neural networks is much more challenging than that in linear models, kernel methods or random feature models. The foremost challenge stems from nonconvexity: previous works on linear models and kernel methods/random features are all in the convex setting, while neural network training is a highly nonconvex optimization problem. Therefore, while most of the previous works can study the minimum norm interpolators/maximum margin classifiers according to the *implicit bias* (Soudry et al., 2018) results for the corresponding models, existing implicit bias results for neural networks (e.g., Lyu and Li (2019)) are not sufficient and a new analysis of the neural network learning process is in demand.

In this work, we provide one such algorithmic analysis for learning two-layer convolutional neural networks (CNNs) with the second layer parameters being fixed as $+1$'s and $-1$'s and polynomial ReLU activation function: $\sigma(z) = \max\{0, z\}^q$, where $q > 2$ is a hyperparameter. We consider a setting where the input data consist of *label dependent signals* and *label independent noises*, and utilize a *signal-noise decomposition* of the CNN filters to precisely characterize the signal learning and noise memorization processes during neural network training. Our result not only demonstrates that benign overfitting can occur in learning two-layer neural networks, but also gives precise conditions under which the overfitted CNN trained by gradient descent can achieve small population loss. Our paper makes the following major contributions:

- We establish population loss bounds of overfitted CNN models trained by gradient descent, and theoretically demonstrate that benign overfitting can occur in learning over-parameterized neural networks. We show that under certain conditions on the signal-to-noise ratio, CNN models trained by gradient descent will prioritize learning the signal over memorizing the noise, and thus achieving both small training and test losses. To the best of our knowledge, this is the first result on the benign overfitting of neural networks that is beyond the neural tangent kernel regime.

- We also establish a negative result showing that when the conditions on the signal-to-noise ratio do not hold, then the overfitted CNN model will achieve at least a constant population loss. This result, together with our upper bound result, reveals an interesting phase transition between benign overfitting and harmful overfitting.

- Our analysis is based on a new proof technique namely *signal-noise decomposition*, which decomposes the convolutional filters into a linear combination of initial filters, the signal vectors and the noise vectors. We convert the neural network learning into a discrete dynamical system of the coefficients from the decomposition, and perform a two-stage analysis that decouples the complicated relation among the coefficients. This enables us to analyze the non-convex optimization problem, and bound the population loss of the CNN trained by gradient descent. We believe our proof technique is of independent interest and can potentially be applied to deep neural networks.

We note that a concurrent work (Frei et al., 2022) studies learning log-Concave mixture data with label flip noise using fully-connected two-layer neural networks with smoothed leaky ReLU activation. Notably, their risk bound matches the risk bound for linear models given in Cao et al. (2021) when the label flip noise is zero. However, their analysis only focuses on upper bounding the risk, and cannot demonstrate the phase transition between benign and harmful overfitting. Compared with (Frei et al., 2022), we focus on CNNs, and consider a different data model to better capture the nature of image classification problems. Moreover, we present both positive and negative results under different SNR regimes, and demonstrate a sharp phase transition between benign and harmful overfitting.

**Notation.** Given two sequences $\{x_n\}$ and $\{y_n\}$, we denote $x_n = O(y_n)$ if there exist some absolute constant $C_1 > 0$ and $N > 0$ such that $|x_n| \leq C_1|y_n|$ for all $n \geq N$. Similarly, we denote $x_n = \Omega(y_n)$ if there exist $C_2 > 0$ and $N > 0$ such that $|x_n| \geq C_2|y_n|$ for all $n > N$. We say

$x_n = \Theta(y_n)$ if $x_n = O(y_n)$ and $x_n = \Omega(y_n)$ both holds. We use $\widetilde{O}(\cdot)$, $\widetilde{\Omega}(\cdot)$, and $\widetilde{\Theta}(\cdot)$ to hide logarithmic factors in these notations respectively. Moreover, we denote $x_n = \text{poly}(y_n)$ if $x_n = O(y_n^D)$ for some positive constant $D$, and $x_n = \text{polylog}(y_n)$ if $x_n = \text{poly}(\log(y_n))$. Finally, for two scalars $a$ and $b$, we denote $a \vee b = \max\{a, b\}$.

## 2  Related Work

A line of recent works have attempted to understand why overfitted predictors can still achieve a good test performance. Belkin et al. (2019a) first empirically demonstrated that in many machine learning models such as random Fourier features, decision trees and ensemble methods , the population risk curve has a *double descent* shape with respect to the number of model parameters. Belkin et al. (2019b) further studied two specific data models, namely the Gaussian model and Fourier series model, and theoretically demonstrated the double descent risk curve in linear regression. Bartlett et al. (2020) studied over-parameterized linear regression to fit data produced by a linear model with additive noises, and established matching upper and lower bounds of the risk achieved by the minimum norm interpolator on the training dataset. It is shown that under certain conditions on the spectrum of the data covariance matrix, the population risk of the interpolator can be asymptotically optimal. Hastie et al. (2019); Wu and Xu (2020) studied linear regression in the setting where both the dimension and sample size grow together with a fixed ratio, and showed double descent of the risk with respect to this ratio. Chatterji and Long (2020) studied the population risk bounds of over-parameterized linear logistic regression on sub-Gaussian mixture models with label flipping noises, and showed how gradient descent can train over-parameterized linear models to achieve nearly optimal population risk. Cao et al. (2021) tightened the upper bound given by Chatterji and Long (2020) in the case without the label flipping noises, and established a matching lower bound of the risk achieved by over-parameterized maximum margin interpolators. Shamir (2022) proposed a generic data model for benign overfitting of linear predictors, and studied different problem settings under which benign overfitting can or cannot occur.

Besides the studies on linear models, several recent works also studied the benign overfitting and double descent phenomena in kernel methods or random feature models. Zhang et al. (2017) first pointed out that overfitting kernel predictors can sometimes still achieve good population risk. Liang and Rakhlin (2020) studied how interpolating kernel regression with radial basis function (RBF) kernels (and variants) can generalize and how the spectrum of the data covariance matrix affects the population risk of the interpolating kernel predictor. Li et al. (2021) studied the benign overfitting phenomenon of random feature models defined as two-layer neural networks whose first layer parameters are fixed at random initialization. Mei and Montanari (2019); Liao et al. (2020) demonstrated the double descent phenomenon for the population risk of interpolating random feature predictors with respect to the ratio between the dimensions of the random feature and the data input. Adlam and Pennington (2020) shows that neural tangent kernel (Jacot et al., 2018) based kernel regression has a triple descent risk curve with respect to the total number of trainable parameters. Montanari and Zhong (2020) further pointed out an interesting phase transition of the generalization error achieved by neural networks trained in the neural tangent kernel regime.

## 3  Problem Setup

In this section, we introduce the data generation model and the convolutional neural network we consider in this paper. We focus on binary classification, and present our data distribution $\mathcal{D}$ in the following definition.

**Definition 3.1.** *Let $\boldsymbol{\mu} \in \mathbb{R}^d$ be a fixed vector representing the signal contained in each data point. Then each data point $(\mathbf{x}, y)$ with $\mathbf{x} = [\mathbf{x}^{(1)\top}, \mathbf{x}^{(2)\top}]^\top \in \mathbb{R}^{2d}$ and $y \in \{-1, 1\}$ is generated from the following distribution $\mathcal{D}$:*

1. *The label $y$ is generated as a Rademacher random variable.*

2. *A noise vector $\boldsymbol{\xi}$ is generated from the Gaussian distribution $N(\mathbf{0}, \sigma_p^2 \cdot (\mathbf{I} - \boldsymbol{\mu}\boldsymbol{\mu}^\top \cdot \|\boldsymbol{\mu}\|_2^{-2}))$.*

3. *One of $\mathbf{x}^{(1)}, \mathbf{x}^{(2)}$ is randomly selected and then assigned as $y \cdot \boldsymbol{\mu}$, which represents the signal; the other is then given by $\boldsymbol{\xi}$, which represents noises.*

Our data generation model is inspired by image data, where the inputs consist of different patches, and only some of the patches are related to the class label of the image. In detail, the patch assigned as $y \cdot \boldsymbol{\mu}$ is the signal patch that is correlated to the label of the data, and the patch assigned as $\boldsymbol{\xi}$ is the noise patch that is independent of the label of the data and therefore is irrelevant for prediction. We assume that the noise patch is generated from the Gaussian distribution $N(\mathbf{0}, \sigma_p^2 \cdot (\mathbf{I} - \boldsymbol{\mu}\boldsymbol{\mu}^\top \cdot \|\boldsymbol{\mu}\|_2^{-2}))$ to ensure that the noise vector is orthogonal to the signal vector $\boldsymbol{\mu}$ for simplicity. Note that when the dimension $d$ is large, $\|\boldsymbol{\xi}\|_2 \approx \sigma_p \sqrt{d}$ by standard concentration bounds. Therefore, we can treat $\|\boldsymbol{\mu}\|_2/(\sigma_p\sqrt{d}) \approx \|\boldsymbol{\mu}\|_2/\|\boldsymbol{\xi}\|_2$ as the signal-to-noise ratio (SNR). For the ease of discussion, we denote $\mathrm{SNR} = \|\boldsymbol{\mu}\|_2/(\sigma_p\sqrt{d})$. Note that the Bayes risk for learning our model is zero. We can also add label flip noise similar to Chatterji and Long (2020); Frei et al. (2022) to make the Bayes risk equal to the label flip noise and therefore nonzero, but this will not change the key message of our paper.

Intuitively, if a classifier learns the signal $\boldsymbol{\mu}$ and utilizes the signal patch of the data to make prediction, it can perfectly fit a given training data set $\{(\mathbf{x}_i, y_i) : i \in [n]\}$ and at the same time have a good performance on the test data. However, when the dimension $d$ is large ($d > n$), a classifier that is a function of the noises $\boldsymbol{\xi}_i$, $i \in [n]$ can also perfectly fit the training data set, while the prediction will be totally random on the new test data. Therefore, the data generation model given in Definition 3.1 is a useful model to study the population loss of overfitted classifiers. Similar models have been studied in some recent works by Li et al. (2019); Allen-Zhu and Li (2020a,b); Zou et al. (2021a).

**Two-layer CNNs.** We consider a two-layer convolutional neural network whose filters are applied to the two patches $\mathbf{x}^{(1)}$ and $\mathbf{x}^{(2)}$ separately, and the second layer parameters of the network are fixed as $+1/m$ and $-1/m$ respectively. Then the network can be written as $f(\mathbf{W}, \mathbf{x}) = F_{+1}(\mathbf{W}_{+1}, \mathbf{x}) - F_{-1}(\mathbf{W}_{-1}, \mathbf{x})$, where $F_{+1}(\mathbf{W}_{+1}, \mathbf{x})$, $F_{-1}(\mathbf{W}_{-1}, \mathbf{x})$ are defined as:

$$F_j(\mathbf{W}_j, \mathbf{x}) = \frac{1}{m}\sum_{r=1}^m \left[ \sigma(\langle \mathbf{w}_{j,r}, \mathbf{x}^{(1)} \rangle) + \sigma(\langle \mathbf{w}_{j,r}, \mathbf{x}^{(2)} \rangle) \right] = \frac{1}{m}\sum_{r=1}^m \left[ \sigma(\langle \mathbf{w}_{j,r}, y \cdot \boldsymbol{\mu} \rangle) + \sigma(\langle \mathbf{w}_{j,r}, \boldsymbol{\xi} \rangle) \right]$$

for $j \in \{+1, -1\}$. Here, $m$ is the number of convolutional filters in $F_{+1}$ and $F_{-1}$, $\sigma(z) = (\max\{0, z\})^q$ is the ReLU$^q$ activation function where $q > 2$, $\mathbf{w}_{j,r} \in \mathbb{R}^d$ denotes the weight for the $r$-th filter (i.e., neuron), and $\mathbf{W}_j$ is the collection of model weights associated with $F_j$. We also use $\mathbf{W}$ to denote the collection of all model weights. We note that our CNN model can also be viewed as a CNN with average global pooling (Lin et al., 2013). We train the above CNN model by minimizing the empirical cross-entropy loss function

$$L_S(\boldsymbol{W}) = \frac{1}{n}\sum_{i=1}^n \ell[y_i \cdot f(\mathbf{W}, \mathbf{x}_i)],$$

where $\ell(z) = \log(1 + \exp(-z))$, and $S = \{(\mathbf{x}_i, y_i)\}_{i=1}^n$ is the training data set. We further define the true loss (test loss) $L_{\mathcal{D}}(\mathbf{W}) := \mathbb{E}_{(\mathbf{x},y)\sim\mathcal{D}} \ell[y \cdot f(\mathbf{W}, \mathbf{x})]$.

We consider gradient descent starting from Gaussian initialization, where each entry of $\mathbf{W}_{+1}$ and $\mathbf{W}_{-1}$ is sampled from a Gaussian distribution $N(0, \sigma_0^2)$, and $\sigma_0^2$ is the variance. The gradient descent update of the filters in the CNN can be written as

$$\mathbf{w}_{j,r}^{(t+1)} = \mathbf{w}_{j,r}^{(t)} - \eta \cdot \nabla_{\mathbf{w}_{j,r}} L_S(\boldsymbol{W}^{(t)})$$

$$= \mathbf{w}_{j,r}^{(t)} - \frac{\eta}{nm}\sum_{i=1}^n \ell_i'^{(t)} \cdot \sigma'(\langle \mathbf{w}_{j,r}^{(t)}, \boldsymbol{\xi}_i \rangle) \cdot jy_i\boldsymbol{\xi}_i - \frac{\eta}{nm}\sum_{i=1}^n \ell_i'^{(t)} \cdot \sigma'(\langle \mathbf{w}_{j,r}^{(t)}, y_i\boldsymbol{\mu} \rangle) \cdot j\boldsymbol{\mu} \quad (3.1)$$

for $j \in \{\pm 1\}$ and $r \in [m]$, where we introduce a shorthand notation $\ell_i'^{(t)} = \ell'[y_i \cdot f(\mathbf{W}^{(t)}, \mathbf{x}_i)]$.

# 4 Main Results

In this section, we present our main theoretical results. At the core of our analyses and results is a *signal-noise decomposition* of the filters in the CNN trained by gradient descent. By the gradient descent update rule (3.1), it is clear that the gradient descent iterate $\mathbf{w}_{j,r}^{(t)}$ is a linear combination of its random initialization $\mathbf{w}_{j,r}^{(0)}$, the signal vector $\boldsymbol{\mu}$ and the noise vectors in the training data $\boldsymbol{\xi}_i$, $i \in [n]$. Motivated by this observation, we introduce the following definition.

**Definition 4.1.** *Let* $\mathbf{w}_{j,r}^{(t)}$ *for* $j \in \{\pm 1\}$, $r \in [m]$ *be the convolution filters of the CNN at the $t$-th iteration of gradient descent. Then there exist unique coefficients* $\gamma_{j,r}^{(t)} \geq 0$ *and* $\rho_{j,r,i}^{(t)}$ *such that*

$$\mathbf{w}_{j,r}^{(t)} = \mathbf{w}_{j,r}^{(0)} + j \cdot \gamma_{j,r}^{(t)} \cdot \|\boldsymbol{\mu}\|_2^{-2} \cdot \boldsymbol{\mu} + \sum_{i=1}^n \rho_{j,r,i}^{(t)} \cdot \|\boldsymbol{\xi}_i\|_2^{-2} \cdot \boldsymbol{\xi}_i.$$

*We further denote* $\overline{\rho}_{j,r,i}^{(t)} := \rho_{j,r,i}^{(t)} \mathbb{1}(\rho_{j,r,i}^{(t)} \geq 0)$, $\underline{\rho}_{j,r,i}^{(t)} := \rho_{j,r,i}^{(t)} \mathbb{1}(\rho_{j,r,i}^{(t)} \leq 0)$. *Then we have that*

$$\mathbf{w}_{j,r}^{(t)} = \mathbf{w}_{j,r}^{(0)} + j \cdot \gamma_{j,r}^{(t)} \cdot \|\boldsymbol{\mu}\|_2^{-2} \cdot \boldsymbol{\mu} + \sum_{i=1}^n \overline{\rho}_{j,r,i}^{(t)} \cdot \|\boldsymbol{\xi}_i\|_2^{-2} \cdot \boldsymbol{\xi}_i + \sum_{i=1}^n \underline{\rho}_{j,r,i}^{(t)} \cdot \|\boldsymbol{\xi}_i\|_2^{-2} \cdot \boldsymbol{\xi}_i. \quad (4.1)$$

We refer to (4.1) as the *signal-noise decomposition* of $\mathbf{w}_{j,r}^{(t)}$. We add normalization factors $\|\boldsymbol{\mu}\|_2^{-2}, \|\boldsymbol{\xi}_i\|_2^{-2}$ in the definition so that $\gamma_{j,r}^{(t)} \approx \langle \mathbf{w}_{j,r}^{(t)}, \boldsymbol{\mu} \rangle, \rho_{j,r,i}^{(t)} \approx \langle \mathbf{w}_{j,r}^{(t)}, \boldsymbol{\xi}_i \rangle$. In this decomposition, $\gamma_{j,r}^{(t)}$ characterizes the progress of learning the signal vector $\boldsymbol{\mu}$, and $\rho_{j,r,i}^{(t)}$ characterizes the degree of noise memorization by the filter. Evidently, based on this decomposition, for some iteration $t$, (i) If some of $\gamma_{j,r}^{(t)}$'s are large enough while $|\rho_{j,r,i}^{(t)}|$ are relatively small, then the CNN will have small training and test losses; (ii) If some $\overline{\rho}_{j,r,i}^{(t)}$'s are large and all $\gamma_{j,r}^{(t)}$'s are small, then the CNN will achieve a small training loss, but a large test loss. Thus, Definition 4.1 provides a handle for us to study the convergence of the training loss as well as the the population loss of the CNN trained by gradient descent.

Our results are based on the following conditions on the dimension $d$, sample size $n$, neural network width $m$, learning rate $\eta$, initialization scale $\sigma_0$.

**Condition 4.2.** *Suppose that*

1. *Dimension $d$ is sufficiently large:* $d = \widetilde{\Omega}(m^{2 \vee [4/(q-2)]} n^{4 \vee [(2q-2)/(q-2)]})$.

2. *Training sample size $n$ and neural network width $m$ satisfy* $n, m = \Omega(\mathrm{polylog}(d))$.

3. *The learning rate $\eta$ satisfies* $\eta \leq \widetilde{O}(\min\{\|\boldsymbol{\mu}\|_2^{-2}, \sigma_p^{-2} d^{-1}\})$.

4. *The standard deviation of Gaussian initialization $\sigma_0$ is appropriately chosen such that* $\widetilde{O}(nd^{-1/2}) \cdot \min\{(\sigma_p \sqrt{d})^{-1}, \|\boldsymbol{\mu}\|_2^{-1}\} \leq \sigma_0 \leq \widetilde{O}(m^{-2/(q-2)} n^{-[1/(q-2)] \vee 1}) \cdot \min\{(\sigma_p \sqrt{d})^{-1}, \|\boldsymbol{\mu}\|_2^{-1}\}$.

A few remarks on Condition 4.2 are in order. The condition on $d$ is to ensure that the learning is in a sufficiently over-parameterized setting, and similar conditions have been made in the study of learning over-parameterized linear models (Chatterji and Long, 2020; Cao et al., 2021). For example, if we choose $q = 3$, then the condition on $d$ becomes $d = \widetilde{\Omega}(m^4 n^4)$. Furthermore, we require the sample size and neural network width to be at least polylogarithmic in the dimension $d$ to ensure some statistical properties of the training data and weight initialization to hold with probability at least $1 - d^{-1}$, which is a mild condition. Finally, the conditions on $\sigma_0$ and $\eta$ are to ensure that gradient descent can effectively minimize the training loss, and they depend on the scale of the training data points. When $\sigma_p = O(d^{-1/2})$ and $\|\boldsymbol{\mu}\|_2 = O(1)$, the step size $\eta$ can be chosen as large as $\widetilde{O}(1)$ and the initialization $\sigma_0$ can be as large as $\widetilde{O}(m^{-2/(q-2)} n^{-[1/(q-2)] \vee 1})$. In our paper, we only require $m, n = \Omega(\mathrm{polylog}(d))$, so our initialization and step-size can be chosen as an almost constant order. Based on these conditions, we give our main result on signal learning in the following theorem.

**Theorem 4.3.** *For any $\epsilon > 0$, let* $T = \widetilde{\Theta}(\eta^{-1} m \sigma_0^{-(q-2)} \|\boldsymbol{\mu}\|_2^{-q} + \eta^{-1} \epsilon^{-1} m^3 \|\boldsymbol{\mu}\|_2^{-2})$. *Under Condition 4.2, if $n \cdot \mathrm{SNR}^q = \widetilde{\Omega}(1)^*$, then with probability at least $1 - d^{-1}$, there exists $0 \leq t \leq T$ such that:*

1. *The CNN learns the signal:* $\max_r \gamma_{j,r}^{(t)} = \Omega(1)$ *for $j \in \{\pm 1\}$.*

2. *The CNN does not memorize the noises in the training data:* $\max_{j,r,i} |\rho_{j,r,i}^{(T)}| = \widetilde{O}(\sigma_0 \sigma_p \sqrt{d})$.

---

*Here the $\widetilde{\Omega}(\cdot)$ hides an $\mathrm{polylog}(\epsilon^{-1})$ factor. This applies to Theorem 4.4 as well.

3. *The training loss converges to $\epsilon$, i.e., $L_S(\mathbf{W}^{(t)}) \leq \epsilon$.*

4. *The trained CNN achieves a small test loss: $L_{\mathcal{D}}(\mathbf{W}^{(t)}) \leq 6\epsilon + \exp(-n^2)$.*

Theorem 4.3 characterizes the case of signal learning. It shows that, if $n \cdot \mathrm{SNR}^q = \widetilde{\Omega}(1)$, then at least one CNN filter can learn the signal by achieving $\gamma_{j,r_j^*}^{(t)} \geq \Omega(1)$, and as a result, the learned neural network can achieve small training and test losses. To demonstrate the sharpness of this condition, we also present the following theorem for the noise memorization by the CNN.

**Theorem 4.4.** *For any $\epsilon > 0$, let $T = \widetilde{\Theta}(\eta^{-1}m \cdot n(\sigma_p\sqrt{d})^{-q} \cdot \sigma_0^{-(q-2)} + \eta^{-1}\epsilon^{-1}nm^3d^{-1}\sigma_p^{-2})$. Under Condition 4.2, if $n^{-1} \cdot \mathrm{SNR}^{-q} = \widetilde{\Omega}(1)$, then with probability at least $1 - d^{-1}$, there exists $0 \leq t \leq T$ such that:*

1. *The CNN memorizes noises in the training data: $\max_r \overline{\rho}_{y_i,r,i}^{(t)} = \Omega(1)$.*

2. *The CNN does not sufficiently learn the signal: $\max_{j,r} \gamma_{j,r}^{(t)} \leq \widetilde{O}(\sigma_0\|\boldsymbol{\mu}\|_2)$.*

3. *The training loss converges to $\epsilon$, i.e., $L_S(\mathbf{W}^{(t)}) \leq \epsilon$.*

4. *The trained CNN has a constant order test loss: $L_{\mathcal{D}}(\mathbf{W}^{(t)}) = \Theta(1)$.*

Theorem 4.4 holds under the condition that $n^{-1} \cdot \mathrm{SNR}^{-q} = \widetilde{\Omega}(1)$. Clearly, this is the opposite regime (up to some logarithmic factors) compared with Theorem 4.3. In this case, the CNN trained by gradient descent mainly memorizes noises in the training data and does not learn enough signal. This, together with the results in Theorem 4.3, reveals a clear phase transition between signal learning and noise memorization in CNN training:

- If $n \cdot \mathrm{SNR}^q = \widetilde{\Omega}(1)$, then the CNN learns the signal and achieves a $O(\epsilon + \exp(-n^2))$ test loss. This is the regime of benign overfitting.

- If $n^{-1} \cdot \mathrm{SNR}^{-q} = \widetilde{\Omega}(1)$ then the CNN can only memorize noises and will have a $\Theta(1)$ test loss. This is the regime of harmful overfitting.

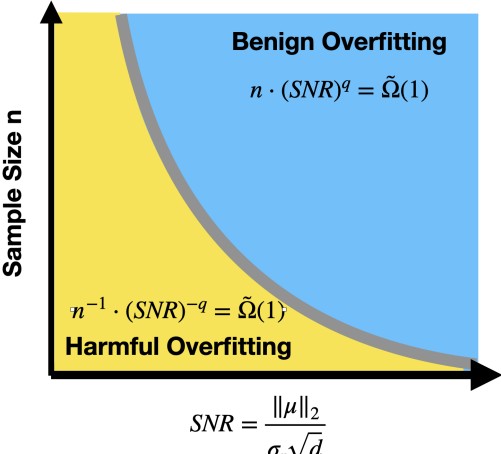

$$SNR = \frac{\|\boldsymbol{\mu}\|_2}{\sigma_p\sqrt{d}}$$

Figure 1: Illustration of the phase transition between benign and harmful overfitting. The blue region represents the setting under which the overfitted CNN trained by gradient descent is guaranteed to have small population loss, and the yellow region represents the setting under which the population loss is guaranteed to be of constant order. The slim gray band region is the setting where the population loss is not well characterized.

The phase transition is illustrated in Figure 1. Clearly, $n \cdot \mathrm{SNR}^q = \widetilde{\Omega}(1)$ is the precise condition under which benign overfitting occurs. Remarkably, in this case the population loss decreases *exponentially* with the sample size $n$. Under our condition that $n = \Omega(\mathrm{polylog}(d))$, this term can also be upper bounded by $1/\mathrm{poly}(d)$, which is small in the high-dimensional setting. Note that when $\|\boldsymbol{\mu}\|_2 = \Theta(1)$ and $\sigma_p = \Theta(d^{-1/2})$, applying standard uniform convergence based bounds (Bartlett et al., 2017; Neyshabur et al., 2018) or stability based bounds (Hardt et al., 2016; Mou et al., 2017; Chen et al., 2018) typically give $\widetilde{O}(n^{-1/2})$ bounds on the generalization gap, which are vacuous when $n = O(\mathrm{polylog}(d))$. Our bound under the same setting is $O(1/\mathrm{poly}(d))$, which is non-vacuous. This is attributed to our precise analysis of signal learning and noise memorization in Theorems 4.3 and 4.4.

**Comparison with neural tangent kernel (NTK) results.** We want to emphasize that our analysis is beyond the so-called neural tangent kernel regime. In the NTK regime, it has been shown that gradient descent can train an over-parameterized neural network to achieve good training and test accuracies (Jacot et al., 2018; Du et al., 2019b,a; Allen-Zhu et al., 2019b; Zou et al., 2019; Arora et al., 2019a; Cao and Gu, 2019a; Chen et al., 2019). However, it is widely believed in literature that the NTK analyses cannot fully explain the success of deep learning, as the neural networks in the NTK regime are almost "linearized" (Lee et al., 2019; Cao and Gu, 2019a). Our analysis and results are not in the NTK regime: In the NTK regime, the network parameters stay close to their initialization throughout training, i.e., $\|\mathbf{W}^{(t)} - \mathbf{W}^{(0)}\|_F = O(1)$, so that the NN model

can be approximated by its linearization (Allen-Zhu et al., 2019b; Cao and Gu, 2019a; Chen et al., 2019). In comparison, our analysis does not rely on linearizing the neural network function, and $\|\mathbf{W}^{(t)} - \mathbf{W}^{(0)}\|_F$ can be as large as $O(\text{poly}(m))$.

# 5 Overview of Proof Technique

In this section, we discuss the main challenges in the study of CNN training under our setting, and explain some key techniques we implement in our proofs to overcome these challenges. The complete proofs of all the results are given in the appendix.

**Main challenges.** Studying benign overfitting under our setting is a challenging task. The first challenge is the nonconvexity of the training objective function $L_S(\mathbf{W})$. Nonconvexity has introduced new challenges in the study of benign overfitting particularly because our goal is not only to show the convergence of the training loss, but also to study the population loss in the over-parameterized setting, which requires a precise algorithmic analysis of the learning problem.

## 5.1 Iterative Analysis of the Signal-Noise Decomposition

In order to study the learning process based on the nonconvex optimization problem, we propose a key technique which enables the iterative analysis of the coefficients in the signal-noise decomposition in Definition 4.1. This technique is given in the following lemma.

**Lemma 5.1.** *The coefficients* $\gamma_{j,r}^{(t)}, \overline{\rho}_{j,r,i}^{(t)}, \underline{\rho}_{j,r,i}^{(t)}$ *in Definition 4.1 satisfy the following equations:*

$$\gamma_{j,r}^{(0)}, \overline{\rho}_{j,r,i}^{(0)}, \underline{\rho}_{j,r,i}^{(0)} = 0, \tag{5.1}$$

$$\gamma_{j,r}^{(t+1)} = \gamma_{j,r}^{(t)} - \frac{\eta}{nm} \cdot \sum_{i=1}^{n} \ell_i'^{(t)} \cdot \sigma'(\langle \mathbf{w}_{j,r}^{(t)}, y_i \cdot \boldsymbol{\mu} \rangle) \cdot \|\boldsymbol{\mu}\|_2^2, \tag{5.2}$$

$$\overline{\rho}_{j,r,i}^{(t+1)} = \overline{\rho}_{j,r,i}^{(t)} - \frac{\eta}{nm} \cdot \ell_i'^{(t)} \cdot \sigma'(\langle \mathbf{w}_{j,r}^{(t)}, \boldsymbol{\xi}_i \rangle) \cdot \|\boldsymbol{\xi}_i\|_2^2 \cdot \mathbb{1}(y_i = j), \tag{5.3}$$

$$\underline{\rho}_{j,r,i}^{(t+1)} = \underline{\rho}_{j,r,i}^{(t)} + \frac{\eta}{nm} \cdot \ell_i'^{(t)} \cdot \sigma'(\langle \mathbf{w}_{j,r}^{(t)}, \boldsymbol{\xi}_i \rangle) \cdot \|\boldsymbol{\xi}_i\|_2^2 \cdot \mathbb{1}(y_i = -j). \tag{5.4}$$

**Remark 5.2.** *With the decomposition (4.1), the signal learning and noise memorization processes of a CNN can be formally studied by analyzing the dynamics of* $\gamma_{j,r}^{(t)}, \overline{\rho}_{j,r,i}^{(t)}, \underline{\rho}_{j,r,i}^{(t)}$ *based on the dynamical system (5.2)-(5.4). Note that prior to our work, several existing results have utilized the inner products* $\langle \mathbf{w}_{j,r}^{(t)}, \boldsymbol{\mu} \rangle$ *during the neural network training process in order to establish generalization bounds (Brutzkus et al., 2018; Chatterji and Long, 2020; Frei et al., 2021). Similar inner product based arguments are also implemented in Allen-Zhu and Li (2020a,b); Zou et al. (2021a), which study different topics related to learning neural networks. Compared with the inner product based argument, our method has two major advantages: (i) Based on the definition (5.2)-(5.4) and the fact that* $\ell_i'^{(t)} < 0$*, it is clear that* $\gamma_{j,r}^{(t)}, \overline{\rho}_{j,r,i}^{(t)}$ *are monotonically increasing, while* $\underline{\rho}_{j,r,i}^{(t)}$ *is monotonically decreasing throughout the whole training process. In comparison, monotonicity does not hold in the inner product based argument, especially for* $\langle \mathbf{w}_{j,r}^{(t)}, \boldsymbol{\xi}_i \rangle$*. (ii) Our signal-noise decomposition also enables a clean homogeneity-based proof for the convergence of the training loss to an arbitrarily small error rate* $\epsilon > 0$*, which will be presented in Subsection 5.2.*

With Lemma 5.1, we can reduce the study of the CNN learning process to the analysis of the discrete dynamical system given by (5.1)-(5.4). Our proof then focuses on a careful assessment of the values of the coefficients $\gamma_{j,r}^{(t)}, \overline{\rho}_{j,r,i}^{(t)}, \underline{\rho}_{j,r,i}^{(t)}$ throughout training. To prepare for more detailed analyses, we first present the following bounds of the coefficients, which hold throughout training.

**Proposition 5.3.** *Under Condition 4.2, for any* $T^* = \eta^{-1}\text{poly}(\epsilon^{-1}, \|\boldsymbol{\mu}\|_2^{-1}, d^{-1}\sigma_p^{-2}, \sigma_0^{-1}, n, m, d)$*, the following bounds hold for* $t \in [0, T^*]$*:*

1. $0 \leq \gamma_{j,r}^{(t)}, \overline{\rho}_{j,r,i}^{(t)} \leq 4\log(T^*)$ *for all* $j \in \{\pm1\}$*,* $r \in [m]$ *and* $i \in [n]$*.*

2. $0 \geq \underline{\rho}_{j,r,i}^{(t)} \geq -2\max_{i,j,r}\{|\langle \mathbf{w}_{j,r}^{(0)}, \boldsymbol{\mu} \rangle|, |\langle \mathbf{w}_{j,r}^{(0)}, \boldsymbol{\xi}_i \rangle|\} - 16n\sqrt{\frac{\log(4n^2/\delta)}{d}} \cdot 4\log(T^*)$ *for all* $j \in \{\pm1\}$*,* $r \in [m]$ *and* $i \in [n]$*.*

We can then prove the following lemma, which demonstrates that the training objective function $L_S(\mathbf{W})$ can dominate the gradient norm $\|\nabla L_S(\mathbf{W}^{(t)})\|_F$ along the gradient descent path.

**Lemma 5.4.** *Under Condition 4.2, for any $T^* = \eta^{-1}poly(\epsilon^{-1}, \|\boldsymbol{\mu}\|_2^{-1}, d^{-1}\sigma_p^{-2}, \sigma_0^{-1}, n, m, d)$, the following result holds for $t \in [0, T^*]$:*

$$\|\nabla L_S(\mathbf{W}^{(t)})\|_F^2 = O\big(\max\{\|\boldsymbol{\mu}\|_2^2, \sigma_p^2 d\}\big) \cdot L_S(\mathbf{W}^{(t)}).$$

Lemma 5.4 plays a key role in the convergence proof of training loss function. However, note that our study of benign overfitting requires carefully monitoring the changes of the coefficients in the signal-noise decomposition, which cannot be directly done by Lemma 5.4. This is quite a challenging task, due to the complicated interactions among $\gamma_{j,r}^{(t)}$, $\overline{\rho}_{j,r,i}^{(t)}$ and $\underline{\rho}_{j,r,i}^{(t)}$. Note that even $\gamma_{j,r}^{(t)}$, which has the simplest formula (5.2), depends on *all* the quantities $\gamma_{j',r'}^{(t)}$, $\overline{\rho}_{j',r',i}^{(t)}$ and $\underline{\rho}_{j',r',i}^{(t)}$ for $j' \in \{\pm 1\}$, $r' \in [m]$ and $i \in [n]$. This is because the cross-entropy loss derivative term $\ell_i'^{(t)} = \ell'[y_i \cdot f(\mathbf{W}^{(t)}, \mathbf{x}_i)]$ depends on all the neurons of the network. To overcome this challenge, we introduce in the next subsection a decoupling technique based on a two-stage analysis.

### 5.2 Decoupling with a Two-Stage Analysis.

We utilize a two-stage analysis to decouple the complicated relation among the coefficients $\gamma_{j,r}^{(t)}$, $\overline{\rho}_{j,r,i}^{(t)}$ and $\underline{\rho}_{j,r,i}^{(t)}$. Intuitively, the initial neural network weights are small enough so that the neural network at initialization has constant level cross-entropy loss derivatives on all the training data: $\ell_i'^{(0)} = \ell'[y_i \cdot f(\mathbf{W}^{(0)}, \mathbf{x}_i)] = \Theta(1)$ for all $i \in [n]$. This is guaranteed under Condition 4.2 and matches neural network training in practice. Motivated by this, we can consider the first stage of the training process where $\ell_i'^{(t)} = \Theta(1)$, in which case we can show significant scale differences among $\gamma_{j,r}^{(t)}$, $\overline{\rho}_{j,r,i}^{(t)}$ and $\underline{\rho}_{j,r,i}^{(t)}$. Based on the result in the first stage, we then proceed to the second stage of the training process where the loss derivatives are no longer at a constant level and show that the training loss can be optimized to be arbitrarily small and meanwhile, the scale differences shown in the first learning stage remain the same throughout the training process. In the following, we focus on explaining the key proof steps for Theorem 4.3. The proof idea for Theorem 4.4 is similar, so we defer the details to the appendix.

***Stage 1.*** It can be shown that, until some of the coefficients $\gamma_{j,r}^{(t)}$, $\rho_{j,r,i}^{(t)}$ reach $\Theta(1)$, we have $\ell_i'^{(t)} = \ell'[y_i \cdot f(\mathbf{W}^{(t)}, \mathbf{x}_i)] = \Theta(1)$ for all $i \in [n]$. Therefore, we first focus on this first stage of the training process, where the dynamics of the coefficients in (5.2) - (5.4) can be greatly simplified by replacing the $\ell_i'^{(t)}$ factors by their constant upper and lower bounds. The following lemma summarizes our main conclusion at stage 1 for signal learning:

**Lemma 5.5.** *Under the same conditions as Theorem 4.3, there exists $T_1 = \widetilde{O}(\eta^{-1}m\sigma_0^{2-q}\|\boldsymbol{\mu}\|_2^{-q})$ such that*

*1. $\max_r \gamma_{j,r}^{(T_1)} = \Omega(1)$ for $j \in \{\pm 1\}$.*

*2. $|\rho_{j,r,i}^{(t)}| = O(\sigma_0 \sigma_p \sqrt{d})$ for all $j \in \{\pm 1\}$, $r \in [m]$, $i \in [n]$ and $0 \le t \le T_1$.*

Lemmas 5.5 takes advantage of the training period when the loss function derivatives remain a constant order to show that the CNN can capture the signal. At the end of stage 1 in signal learning, $\max_r \gamma_{j,r}$ reaches $\Theta(1)$, and is significantly larger than $\rho_{j,r,i}^{(t)}$. After this, it is no longer guaranteed that the loss derivatives $\ell_i'^{(t)}$ will remain constant order, and thus starts the training stage 2.

***Stage 2.*** In this stage, we take into full consideration the exact definition $\ell_i'^{(t)} = \ell'[y_i \cdot f(\mathbf{W}^{(t)}, \mathbf{x}_i)]$ and show that the training loss function will converge to $L_S(\mathbf{W}^{(t)}) < \epsilon$. Thanks to the analysis in stage 1, we know that some $\gamma_{j,r}^{(t)}$ is significantly larger than all $\rho_{j,r,i}^{(t)}$'s at the end of stage 1. This scale difference is the key to our analysis in stage 2. Based on this scale difference and the monotonicity of $\gamma_{j,r}^{(t)}$, $\overline{\rho}_{j,r,i}^{(t)}$, $\underline{\rho}_{j,r,i}^{(t)}$ in the signal-noise decomposition, it can be shown that there exists $\mathbf{W}^*$ such that $y_i \cdot \langle \nabla f(\mathbf{W}^{(t)}, \mathbf{x}_i), \mathbf{W}^* \rangle \ge q \log(2q/\epsilon)$ throughout stage 2. Moreover, since the neural network

$f(\mathbf{W}, \mathbf{x})$ is $q$-homogeneous in $\mathbf{W}$, we have $\langle \nabla f(\mathbf{W}^{(t)}, \mathbf{x}), \mathbf{W}^{(t)} \rangle = q \cdot f(\mathbf{W}^{(t)}, \mathbf{x})$. Therefore,

$$
\begin{aligned}
\langle \nabla L_S(\mathbf{W}^{(t)}), \mathbf{W}^{(t)} - \mathbf{W}^* \rangle &= \frac{1}{n} \sum_{i=1}^n \ell_i'^{(t)} \cdot y_i \cdot \langle \nabla f(\mathbf{W}^{(t)}, \mathbf{x}_i), \mathbf{W}^{(t)} - \mathbf{W}^* \rangle \\
&= \frac{1}{n} \sum_{i=1}^n \ell_i'^{(t)} \cdot [y_i \cdot q \cdot f(\mathbf{W}^{(t)}, \mathbf{x}_i) - y_i \cdot \langle \nabla f(\mathbf{W}^{(t)}, \mathbf{x}_i), \mathbf{W}^* \rangle] \\
&\geq \frac{1}{n} \sum_{i=1}^n \ell'[y_i \cdot f(\mathbf{W}^{(t)}, \mathbf{x}_i)] \cdot [y_i \cdot q \cdot f(\mathbf{W}^{(t)}, \mathbf{x}_i) - q \log(2q/\epsilon)] \\
&\geq q \cdot \frac{1}{n} \sum_{i=1}^n [\ell(f(\mathbf{W}^{(t)}, \mathbf{x}_i)) - \ell(\log(2q/\epsilon))] \\
&\geq q \cdot L_S(\mathbf{W}^{(t)}) - \epsilon/2,
\end{aligned}
$$

where the second inequality follows by the convexity of the cross-entropy loss function. With the above key technique, we can prove the following lemma.

**Lemma 5.6.** *Let $T, T_1$ be defined in Theorem 4.3 and Lemma 5.5 respectively. Then under the same conditions as Theorem 4.3, for any $t \in [T_1, T]$, it holds that $|\rho_{j,r,i}^{(t)}| \leq \sigma_0 \sigma_p \sqrt{d}$ for all $j \in \{\pm 1\}$, $r \in [m]$ and $i \in [n]$. Moreover, let $\mathbf{W}^*$ be the collection of CNN parameters with convolution filters $\mathbf{w}_{j,r}^* = \mathbf{w}_{j,r}^{(0)} + 2qm \log(2q/\epsilon) \cdot j \cdot \|\boldsymbol{\mu}\|_2^{-2} \cdot \boldsymbol{\mu}$. Then the following bound holds*

$$
\frac{1}{t - T_1 + 1} \sum_{s=T_1}^t L_S(\mathbf{W}^{(s)}) \leq \frac{\|\mathbf{W}^{(T_1)} - \mathbf{W}^*\|_F^2}{(2q-1)\eta(t - T_1 + 1)} + \frac{\epsilon}{(2q-1)}
$$

*for all $t \in [T_1, T]$, where we denote $\|\mathbf{W}\|_F = \sqrt{\|\mathbf{W}_{+1}\|_F^2 + \|\mathbf{W}_{-1}\|_F^2}$.*

Lemma 5.6 states two main results on signal learning. First of all, during this training period, it is guaranteed that the coefficients of noise vectors $\rho_{j,r,i}^{(t)}$ in the signal-noise decomposition remain sufficiently small. Moreover, it also gives an optimization type result that the best iterate in $[T_1, T]$ is small as long as $T$ is large enough. Clearly, the convergence of the training loss stated in Theorems 4.3 directly follows by choosing $T$ to be sufficiently large in Lemmas 5.6. The lemma below further gives an upper bound on the test loss.

**Lemma 5.7.** *Let $T$ be defined in Theorem 4.3. Under the same conditions as Theorem 4.3, for any $t \leq T$ with $L_S(\mathbf{W}^{(t)}) \leq 1$, it holds that $L_{\mathcal{D}}(\mathbf{W}^{(t)}) \leq 6 \cdot L_S(\mathbf{W}^{(t)}) + \exp(-n^2)$.*

Below we finalize the proof of Theorem 4.3. The proofs of other results are in the appendix.

*Proof of Theorem 4.3.* The first part of Theorem 4.3 follows by Lemma 5.5 and the monotonicity of $\gamma_{j,r}^{(t)}$. The second part of Theorem 4.3 follows by Lemma 5.6. For the third part, let $\mathbf{W}^*$ be defined in Lemma 5.6. Then by the definition of $\mathbf{W}^*$, we have

$$
\begin{aligned}
\|\mathbf{W}^{(T_1)} - \mathbf{W}^*\|_F &\leq \|\mathbf{W}^{(T_1)} - \mathbf{W}^{(0)}\|_F + \|\mathbf{W}^{(0)} - \mathbf{W}^*\|_F \\
&\leq \sum_{j,r} \gamma_{j,r}^{(T_1)} \|\boldsymbol{\mu}\|_2^{-1} + \sum_{j,r,i} \frac{\overline{\rho}_{j,r,i}^{(T_1)}}{\|\boldsymbol{\xi}_i\|_2} + \sum_{j,r,i} \frac{\underline{\rho}_{j,r,i}^{(T_1)}}{\|\boldsymbol{\xi}_i\|_2} + \Theta(m^{3/2} \log(1/\epsilon)) \|\boldsymbol{\mu}\|_2^{-1} \\
&= \widetilde{O}(m^{3/2} \|\boldsymbol{\mu}\|_2^{-1}),
\end{aligned}
$$

where the first inequality is by triangle inequality, the second inequality is by the signal-noise decomposition of $\mathbf{W}^{(T_1)}$ and the definition of $\mathbf{W}^*$, and the last equality is by Proposition 5.3 and Lemma 5.5. Therefore, choosing $T = \widetilde{\Theta}(\eta^{-1}T_1 + \eta^{-1}\epsilon^{-1}m^3 \|\boldsymbol{\mu}\|_2^{-2}) = \widetilde{\Theta}(\eta^{-1}\sigma_0^{-(q-2)} \|\boldsymbol{\mu}\|_2^{-q} + \eta^{-1}\epsilon^{-1}m^3 \|\boldsymbol{\mu}\|_2^{-2})$ in Lemma 5.6 ensures that

$$
\frac{1}{T - T_1 + 1} \sum_{t=T_1}^T L_S(\mathbf{W}^{(t)}) \leq \frac{\|\mathbf{W}^{(T_1)} - \mathbf{W}^*\|_F^2}{(2q-1)\eta(T - T_1 + 1)} + \frac{\epsilon}{2q-1} \leq \frac{\widetilde{O}(m^3 \|\boldsymbol{\mu}\|_2^{-2})}{(2q-1)\eta(T - T_1 + 1)} + \frac{\epsilon}{2q-1} \leq \epsilon,
$$

and there exists $t \in [T_1, T]$ such that $L_S(\mathbf{W}^{(t)}) \leq \epsilon$. This completes the proof of the third part of Theorem 4.3. Finally, combining this bound with Lemma 5.7 gives

$$L_{\mathcal{D}}(\mathbf{W}^{(t)}) \leq 6 \cdot L_S(\mathbf{W}^{(t)}) + \exp(-n^2) \leq 6\epsilon + \exp(-n^2),$$

which proves the last part of Theorem 4.3. $\qquad\square$

## 6   Conclusion and Future Work

This paper utilizes a signal-noise decomposition to study the signal learning and noise memorization process in the training of a two-layer CNN. We precisely give the conditions under which the CNN will mainly focus on learning signals or memorizing noises, and reveals a phase transition of the population loss with respect to the sample size, signal strength, noise level, and dimension. Our result theoretically demonstrates that benign overfitting can happen in neural network training. An important future work direction is to study the benign overfitting phenomenon of neural networks in learning other data models. Moreover, it is also important to generalize our analysis to deep convolutional neural networks.

## Acknowledgments and Disclosure of Funding

We would like to thank Spencer Frei for valuable comment and discussion on the earlier version of this paper, and pointing out a related work. ZC and QG are supported in part by the National Science Foundation CAREER Award 1906169, IIS-2008981 and the Sloan Research Fellowship. MB is grateful for the support from the National Science Foundation (NSF) and the Simons Foundation for the Collaboration on the Theoretical Foundations of Deep Learning[†] through awards DMS-2031883 and #814639 as well as NSF IIS-1815697 and the TILOS institute (NSF CCF-2112665).

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
