# A   Additional Related Work

There has also been a large number of works studying the optimization and generalization of neural networks. A series of work (Li and Yuan, 2017; Soltanolkotabi, 2017; Du et al., 2018a,b; Zhong et al., 2017; Zhang et al., 2019; Cao and Gu, 2019b) studied the parameter recovery problem in two-layer neural networks, where the data are given by a teacher network and the task is to recover the parameters in the teacher network. These works either focus on the noiseless setting, or requires the number of training data points to be larger than the number of parameters in the network, and therefore does not cover the setting where the neural network can overfit the training data. Another line of works (Neyshabur et al., 2015; Bartlett et al., 2017; Neyshabur et al., 2018; Golowich et al., 2018; Arora et al., 2018) have studied the generalization gap between the training and test losses of neural networks with uniform convergence based arguments. However, these results are not algorithm-dependent and cannot explain benign overfitting. Some recent works studied the generalization gap based on stability based arguments (Bousquet and Elisseeff, 2002; Hardt et al., 2016; Mou et al., 2017; Chen et al., 2018). A more recent line of works studied the convergence (Jacot et al., 2018; Li and Liang, 2018; Du et al., 2019b; Allen-Zhu et al., 2019b; Du et al., 2019a; Zou et al., 2019) and test error bounds (Allen-Zhu et al., 2019a; Arora et al., 2019a,b; Cao and Gu, 2019a; Ji and Telgarsky, 2020; Chen et al., 2019) of over-parameterized networks in the neural tangent kernel regime. However, these works depend on the equivalence between neural network training and kernel methods, which cannot fully explain the success of deep learning. Compared with the works mentioned above, our work has a different focus which is to study the conditions for benign and harmful overfitting.

# B   Preliminary Lemmas

In this section, we present some pivotal lemmas that give some important properties of the data and the neural network parameters at their random initialization.

**Lemma B.1.** *Suppose that $\delta > 0$ and $n \geq 8\log(4/\delta)$. Then with probability at least $1 - \delta$,*

$$|\{i \in [n] : y_i = 1\}|, \ |\{i \in [n] : y_i = -1\}| \geq n/4.$$

*Proof of Lemma B.1.* By Hoeffding's inequality, with probability at least $1 - \delta/2$, we have

$$\left| \frac{1}{n} \sum_{i=1}^{n} \mathbb{1}\{y_i = 1\} - \frac{1}{2} \right| \leq \sqrt{\frac{\log(4/\delta)}{2n}}.$$

Therefore, as long as $n \geq 8\log(4/\delta)$, we have

$$|\{i \in [n] : y_i = 1\}| = \sum_{i=1}^{n} \mathbb{1}\{y_i = 1\} \geq \frac{n}{2} - n \cdot \sqrt{\frac{\log(4/\delta)}{2n}} \geq \frac{n}{4}.$$

This proves the result for $|\{i \in [n] : y_i = 1\}|$. The proof for $|\{i \in [n] : y_i = -1\}|$ is exactly the same, and we can conclude the proof by applying a union bound. $\qquad\square$

The following lemma estimates the norms of the noise vectors $\boldsymbol{\xi}_i$, $i \in [n]$, and gives an upper bound of their inner products with each other.

**Lemma B.2.** *Suppose that $\delta > 0$ and $d = \Omega(\log(4n/\delta))$. Then with probability at least $1 - \delta$,*

$$\sigma_p^2 d/2 \leq \|\boldsymbol{\xi}_i\|_2^2 \leq 3\sigma_p^2 d/2,$$
$$|\langle \boldsymbol{\xi}_i, \boldsymbol{\xi}_{i'} \rangle| \leq 2\sigma_p^2 \cdot \sqrt{d\log(4n^2/\delta)}$$

*for all $i, i' \in [n]$.*

*Proof of Lemma B.2.* By Bernstein's inequality, with probability at least $1 - \delta/(2n)$ we have

$$\left| \|\boldsymbol{\xi}_i\|_2^2 - \sigma_p^2 d \right| = O(\sigma_p^2 \cdot \sqrt{d\log(4n/\delta)}).$$

Therefore, as long as $d = \Omega(\log(4n/\delta))$, we have

$$\sigma_p^2 d/2 \le \|\boldsymbol{\xi}_i\|_2^2 \le 3\sigma_p^2 d/2.$$

Moreover, clearly $\langle \boldsymbol{\xi}_i, \boldsymbol{\xi}_{i'} \rangle$ has mean zero. For any $i, i'$ with $i \ne i'$, by Bernstein's inequality, with probability at least $1 - \delta/(2n^2)$ we have

$$|\langle \boldsymbol{\xi}_i, \boldsymbol{\xi}_{i'} \rangle| \le 2\sigma_p^2 \cdot \sqrt{d \log(4n^2/\delta)}.$$

Applying a union bound completes the proof. $\qquad\square$

The following lemma studies the inner product between a randomly initialized CNN convolutional filter $\mathbf{w}_{j,r}^{(0)}$, $j \in \{+1, -1\}$ and $r \in [m]$ and the signal/noise vectors in the training data. The calculations characterize how the neural network at initialization randomly captures signal and noise information.

**Lemma B.3.** *Suppose that $d \ge \Omega(\log(mn/\delta))$, $m = \Omega(\log(1/\delta))$. Then with probability at least $1 - \delta$,*

$$|\langle \mathbf{w}_{j,r}^{(0)}, \boldsymbol{\mu} \rangle| \le \sqrt{2\log(8m/\delta)} \cdot \sigma_0 \|\boldsymbol{\mu}\|_2,$$
$$|\langle \mathbf{w}_{j,r}^{(0)}, \boldsymbol{\xi}_i \rangle| \le 2\sqrt{\log(8mn/\delta)} \cdot \sigma_0 \sigma_p \sqrt{d}$$

*for all $r \in [m]$, $j \in \{\pm 1\}$ and $i \in [n]$. Moreover,*

$$\sigma_0 \|\boldsymbol{\mu}\|_2/2 \le \max_{r \in [m]} j \cdot \langle \mathbf{w}_{j,r}^{(0)}, \boldsymbol{\mu} \rangle \le \sqrt{2\log(8m/\delta)} \cdot \sigma_0 \|\boldsymbol{\mu}\|_2,$$
$$\sigma_0 \sigma_p \sqrt{d}/4 \le \max_{r \in [m]} j \cdot \langle \mathbf{w}_{j,r}^{(0)}, \boldsymbol{\xi}_i \rangle \le 2\sqrt{\log(8mn/\delta)} \cdot \sigma_0 \sigma_p \sqrt{d}$$

*for all $j \in \{\pm 1\}$ and $i \in [n]$.*

*Proof of Lemma B.3.* It is clear that for each $r \in [m]$, $j \cdot \langle \mathbf{w}_{j,r}^{(0)}, \boldsymbol{\mu} \rangle$ is a Gaussian random variable with mean zero and variance $\sigma_0^2 \|\boldsymbol{\mu}\|_2^2$. Therefore, by Gaussian tail bound and union bound, with probability at least $1 - \delta/4$,

$$j \cdot \langle \mathbf{w}_{j,r}^{(0)}, \boldsymbol{\mu} \rangle \le |\langle \mathbf{w}_{j,r}^{(0)}, \boldsymbol{\mu} \rangle| \le \sqrt{2\log(8m/\delta)} \cdot \sigma_0 \|\boldsymbol{\mu}\|_2.$$

Moreover, $\mathbb{P}(\sigma_0 \|\boldsymbol{\mu}\|_2/2 > j \cdot \langle \mathbf{w}_{j,r}^{(0)}, \boldsymbol{\mu} \rangle)$ is an absolute constant, and therefore by the condition on $m$, we have

$$\mathbb{P}\big(\sigma_0 \|\boldsymbol{\mu}\|_2/2 \le \max_{r \in [m]} j \cdot \langle \mathbf{w}_{j,r}^{(0)}, \boldsymbol{\mu} \rangle\big) = 1 - \mathbb{P}\big(\sigma_0 \|\boldsymbol{\mu}\|_2/2 > \max_{r \in [m]} j \cdot \langle \mathbf{w}_{j,r}^{(0)}, \boldsymbol{\mu} \rangle\big)$$
$$= 1 - \mathbb{P}\big(\sigma_0 \|\boldsymbol{\mu}\|_2/2 > j \cdot \langle \mathbf{w}_{j,r}^{(0)}, \boldsymbol{\mu} \rangle\big)^{2m}$$
$$\ge 1 - \delta/4.$$

By Lemma B.2, with probability at least $1 - \delta/4$, $\sigma_p \sqrt{d}/\sqrt{2} \le \|\boldsymbol{\xi}_i\|_2 \le \sqrt{3/2} \cdot \sigma_p \sqrt{d}$ for all $i \in [n]$. Therefore, the result for $\langle \mathbf{w}_{j,r}^{(0)}, \boldsymbol{\xi}_i \rangle$ follows the same proof as $j \cdot \langle \mathbf{w}_{j,r}^{(0)}, \boldsymbol{\mu} \rangle$. $\qquad\square$

## C  Signal-noise Decomposition Analysis

In this section, we establish a series of results on the signal-noise decomposition. These results are based on the conclusions in Section B, which hold with high probability. Denote by $\mathcal{E}_{\mathrm{prelim}}$ the event that all the results in Section B hold. Then for simplicity and clarity, we state all the results in this and the following sections conditional on $\mathcal{E}_{\mathrm{prelim}}$.

**Lemma C.1** (Restatement of Lemma 5.1). *The coefficients $\gamma_{j,r}^{(t)}, \overline{\rho}_{j,r,i}^{(t)}, \underline{\rho}_{j,r,i}^{(t)}$ defined in Definition 4.1 satisfy the following iterative equations:*

$$\gamma_{j,r}^{(0)}, \overline{\rho}_{j,r,i}^{(0)}, \underline{\rho}_{j,r,i}^{(0)} = 0,$$

$$\gamma_{j,r}^{(t+1)} = \gamma_{j,r}^{(t)} - \frac{\eta}{nm} \cdot \sum_{i=1}^{n} \ell_i'^{(t)} \cdot \sigma'(\langle \mathbf{w}_{j,r}^{(t)}, y_i \cdot \boldsymbol{\mu} \rangle) \cdot \|\boldsymbol{\mu}\|_2^2,$$

$$\overline{\rho}_{j,r,i}^{(t+1)} = \overline{\rho}_{j,r,i}^{(t)} - \frac{\eta}{nm} \cdot \ell_i'^{(t)} \cdot \sigma'(\langle \mathbf{w}_{j,r}^{(t)}, \boldsymbol{\xi}_i \rangle) \cdot \|\boldsymbol{\xi}_i\|_2^2 \cdot \mathbb{1}(y_i = j),$$

$$\underline{\rho}_{j,r,i}^{(t+1)} = \underline{\rho}_{j,r,i}^{(t)} + \frac{\eta}{nm} \cdot \ell_i'^{(t)} \cdot \sigma'(\langle \mathbf{w}_{j,r}^{(t)}, \boldsymbol{\xi}_i \rangle) \cdot \|\boldsymbol{\xi}_i\|_2^2 \cdot \mathbb{1}(y_i = -j)$$

*for all $r \in [m]$, $j \in \{\pm 1\}$ and $i \in [n]$.*

*Proof of Lemma C.1.* By our data model in Definition 3.1 and Gaussian initialization of the CNN weights, it is clear that with probability 1, the vectors are linearly independent. Therefore, the decomposition (4.1) is unique. Now consider $\widetilde{\gamma}_{j,r}^{(0)}, \widetilde{\rho}_{j,r,i}^{(0)} = 0$ and

$$\widetilde{\gamma}_{j,r}^{(t+1)} = \widetilde{\gamma}_{j,r}^{(t)} - \frac{\eta}{nm} \cdot \sum_{i=1}^{n} \ell_i'^{(t)} \cdot \sigma'(\langle \mathbf{w}_{j,r}^{(t)}, y_i \cdot \boldsymbol{\mu} \rangle) \cdot \|\boldsymbol{\mu}\|_2^2,$$

$$\widetilde{\rho}_{j,r,i}^{(t+1)} = \widetilde{\rho}_{j,r,i}^{(t)} - \frac{\eta}{nm} \cdot \ell_i'^{(t)} \cdot \sigma'(\langle \mathbf{w}_{j,r}^{(t)}, \boldsymbol{\xi}_i \rangle) \cdot \|\boldsymbol{\xi}_i\|_2^2 \cdot jy_i,$$

It is then easy to check by (3.1) that

$$\mathbf{w}_{j,r}^{(t)} = \mathbf{w}_{j,r}^{(0)} + j \cdot \widetilde{\gamma}_{j,r}^{(t)} \cdot \|\boldsymbol{\mu}\|_2^{-2} \cdot \boldsymbol{\mu} + \sum_{i=1}^{n} \widetilde{\rho}_{j,r,i}^{(t)} \|\boldsymbol{\xi}_i\|_2^{-2} \cdot \boldsymbol{\xi}_i.$$

Hence by the uniqueness of the decomposition we have $\gamma_{j,r}^{(t)} = \widetilde{\gamma}_{j,r}^{(t)}$ and $\rho_{j,r,i}^{(t)} = \widetilde{\rho}_{j,r,i}^{(t)}$. Then we have that

$$\rho_{j,r,i}^{(t)} = -\sum_{s=0}^{t-1} \frac{\eta}{nm} \cdot \ell_i'^{(s)} \cdot \sigma'(\langle \mathbf{w}_{j,r}^{(s)}, \boldsymbol{\xi}_i \rangle) \cdot \|\boldsymbol{\xi}_i\|_2^2 \cdot jy_i$$

Moreover, note that $\ell_i'^{(t)} < 0$ by the definition of the cross-entropy loss. Therefore,

$$\overline{\rho}_{j,r,i}^{(t)} = -\sum_{s=0}^{t-1} \frac{\eta}{nm} \cdot \ell_i'^{(s)} \cdot \sigma'(\langle \mathbf{w}_{j,r}^{(s)}, \boldsymbol{\xi}_i \rangle) \cdot \|\boldsymbol{\xi}_i\|_2^2 \cdot \mathbb{1}(y_i = j), \tag{C.1}$$

$$\underline{\rho}_{j,r,i}^{(t)} = -\sum_{s=0}^{t-1} \frac{\eta}{nm} \cdot \ell_i'^{(s)} \cdot \sigma'(\langle \mathbf{w}_{j,r}^{(s)}, \boldsymbol{\xi}_i \rangle) \cdot \|\boldsymbol{\xi}_i\|_2^2 \cdot \mathbb{1}(y_i = -j). \tag{C.2}$$

Writing out the iterative versions of (C.1) and (C.2) completes the proof. $\square$

We can futher plug the signal-noise decomposition (4.1) into the iterative formulas in Lemma C.1. By the second equation in Lemma C.1, we have

$$\gamma_{j,r}^{(t+1)} = \gamma_{j,r}^{(t)} - \frac{\eta}{nm} \cdot \sum_{i=1}^{n} \ell_i'^{(t)} \cdot \sigma'(y_i \cdot \langle \mathbf{w}_{j,r}^{(0)}, \boldsymbol{\mu} \rangle + y_i \cdot j \cdot \gamma_{j,r}^{(t)}) \cdot \|\boldsymbol{\mu}\|_2^2, \tag{C.3}$$

Moreover, by the third equation in Lemma C.1, we have

$$\overline{\rho}_{j,r,i}^{(t+1)} = \overline{\rho}_{j,r,i}^{(t)} - \frac{\eta}{nm} \cdot \ell_i'^{(t)} \sigma'\left( \langle \mathbf{w}_{j,r}^{(0)}, \boldsymbol{\xi}_i \rangle + \sum_{i'=1}^{n} \overline{\rho}_{j,r,i'}^{(t)} \frac{\langle \boldsymbol{\xi}_{i'}, \boldsymbol{\xi}_i \rangle}{\|\boldsymbol{\xi}_{i'}\|_2^2} + \sum_{i'=1}^{n} \underline{\rho}_{j,r,i'}^{(t)} \frac{\langle \boldsymbol{\xi}_{i'}, \boldsymbol{\xi}_i \rangle}{\|\boldsymbol{\xi}_{i'}\|_2^2} \right) \cdot \|\boldsymbol{\xi}_i\|_2^2 \tag{C.4}$$

if $j = y_i$, and $\overline{\rho}_{j,r,i}^{(t)} = 0$ for all $t \geq 0$ if $j = -y_i$. Similarly, by the last equation in Lemma C.1, we have

$$\underline{\rho}_{j,r,i}^{(t+1)} = \underline{\rho}_{j,r,i}^{(t)} + \frac{\eta}{nm} \cdot \ell_i'^{(t)} \sigma'\left( \langle \mathbf{w}_{j,r}^{(0)}, \boldsymbol{\xi}_i \rangle + \sum_{i'=1}^{n} \overline{\rho}_{j,r,i'}^{(t)} \frac{\langle \boldsymbol{\xi}_{i'}, \boldsymbol{\xi}_i \rangle}{\|\boldsymbol{\xi}_{i'}\|_2^2} + \sum_{i'=1}^{n} \underline{\rho}_{j,r,i'}^{(t)} \frac{\langle \boldsymbol{\xi}_{i'}, \boldsymbol{\xi}_i \rangle}{\|\boldsymbol{\xi}_{i'}\|_2^2} \right) \cdot \|\boldsymbol{\xi}_i\|_2^2 \tag{C.5}$$

if $j = -y_i$, and $\underline{\rho}_{j,r,i}^{(t)} = 0$ for all $t \geq 0$ if $j = y_i$.

We will now show that the parameter of the signal-noise decomposition will stay a reasonable scale during a long time of training. Let us consider the learning period $0 \leq t \leq T^*$, where $T^* = \eta^{-1}\text{poly}(\epsilon^{-1}, \|\boldsymbol{\mu}\|_2^{-1}, d^{-1}\sigma_p^{-2}, \sigma_0^{-1}, n, m, d)$ is the maximum admissible iterations. Note that we can consider any polynomial training time $T^*$. Denote $\alpha = 4\log(T^*)$. Here we list the exact conditions for $\eta, \sigma_0, d$ required by the proofs in this section, which are part of Condition 4.2:

$$\eta = O\Big(\min\{nm/(q\sigma_p^2 d), nm/(q2^{q+2}\alpha^{q-2}\sigma_p^2 d), nm/(q2^{q+2}\alpha^{q-2}\|\boldsymbol{\mu}\|_2^2)\}\Big), \tag{C.6}$$

$$\sigma_0 \leq [16\sqrt{\log(8mn/\delta)}]^{-1}\min\{\|\boldsymbol{\mu}\|_2^{-1}, (\sigma_p\sqrt{d})^{-1}\}, \tag{C.7}$$

$$d \geq 1024\log(4n^2/\delta)\alpha^2 n^2. \tag{C.8}$$

Denote $\beta = 2\max_{i,j,r}\{|\langle \mathbf{w}_{j,r}^{(0)}, \boldsymbol{\mu}\rangle|, |\langle \mathbf{w}_{j,r}^{(0)}, \boldsymbol{\xi}_i\rangle|\}$. By Lemma B.3, with probability at least $1 - \delta$, we can upper bound $\beta$ by $4\sqrt{\log(8mn/\delta)} \cdot \sigma_0 \cdot \max\{\|\boldsymbol{\mu}\|_2, \sigma_p\sqrt{d}\}$. Then, by (C.7) and (C.8), it is straightforward to verify the following inequality:

$$4\max\left\{\beta, 8n\sqrt{\frac{\log(4n^2/\delta)}{d}}\alpha\right\} \leq 1. \tag{C.9}$$

Suppose the conditions listed in (C.6), (C.7) and (C.8) hold, we claim that for $0 \leq t \leq T^*$ the following property holds.

**Proposition C.2** (Restatement of Proposition 5.3). *Under Condition 4.2, for $0 \leq t \leq T^*$, we have that*

$$0 \leq \gamma_{j,r}^{(t)}, \overline{\rho}_{j,r,i}^{(t)} \leq \alpha, \tag{C.10}$$

$$0 \geq \underline{\rho}_{j,r,i}^{(t)} \geq -\beta - 16n\sqrt{\frac{\log(4n^2/\delta)}{d}}\alpha \geq -\alpha. \tag{C.11}$$

*for all $r \in [m]$, $j \in \{\pm 1\}$ and $i \in [n]$.*

We will use induction to prove Proposition C.2. We first introduce several technical lemmas that will be used for the proof of Proposition C.2.

**Lemma C.3.** *For any $t \geq 0$, it holds that $\langle \mathbf{w}_{j,r}^{(t)} - \mathbf{w}_{j,r}^{(0)}, \boldsymbol{\mu}\rangle = j \cdot \gamma_{j,r}^{(t)}$ for all $r \in [m]$, $j \in \{\pm 1\}$.*

*Proof of Lemma C.3.* For any time $t \geq 0$, we have that

$$\langle \mathbf{w}_{j,r}^{(t)} - \mathbf{w}_{j,r}^{(0)}, \boldsymbol{\mu}\rangle = j \cdot \gamma_{j,r}^{(t)} + \sum_{i'=1}^{n} \overline{\rho}_{j,r,i'}^{(t)}\|\boldsymbol{\xi}_{i'}\|_2^{-2} \cdot \langle \boldsymbol{\xi}_{i'}, \boldsymbol{\mu}\rangle + \sum_{i'=1}^{n} \underline{\rho}_{j,r,i'}^{(t)}\|\boldsymbol{\xi}_{i'}\|_2^{-2} \cdot \langle \boldsymbol{\xi}_{i'}, \boldsymbol{\mu}\rangle$$

$$= j \cdot \gamma_{j,r}^{(t)},$$

where the equation is by our orthogonal assumption. $\square$

**Lemma C.4.** *Under Condition 4.2, suppose (C.10) and (C.11) hold at iteration $t$. Then*

$$\underline{\rho}_{j,r,i}^{(t)} - 8n\sqrt{\frac{\log(4n^2/\delta)}{d}}\alpha \leq \langle \mathbf{w}_{j,r}^{(t)} - \mathbf{w}_{j,r}^{(0)}, \boldsymbol{\xi}_i\rangle \leq \underline{\rho}_{j,r,i}^{(t)} + 8n\sqrt{\frac{\log(4n^2/\delta)}{d}}\alpha, \; j \neq y_i,$$

$$\overline{\rho}_{j,r,i}^{(t)} - 8n\sqrt{\frac{\log(4n^2/\delta)}{d}}\alpha \leq \langle \mathbf{w}_{j,r}^{(t)} - \mathbf{w}_{j,r}^{(0)}, \boldsymbol{\xi}_i\rangle \leq \overline{\rho}_{j,r,i}^{(t)} + 8n\sqrt{\frac{\log(4n^2/\delta)}{d}}\alpha, \; j = y_i$$

*for all $r \in [m]$, $j \in \{\pm 1\}$ and $i \in [n]$.*

*Proof of Lemma C.4.* For $j \neq y_i$, we have that $\overline{\rho}_{j,r,i}^{(t)} = 0$ and

$$\langle \mathbf{w}_{j,r}^{(t)} - \mathbf{w}_{j,r}^{(0)}, \boldsymbol{\xi}_i\rangle = \sum_{i'=1}^{n} \overline{\rho}_{j,r,i'}^{(t)}\|\boldsymbol{\xi}_{i'}\|_2^{-2} \cdot \langle \boldsymbol{\xi}_{i'}, \boldsymbol{\xi}_i\rangle + \sum_{i'=1}^{n} \underline{\rho}_{j,r,i'}^{(t)}\|\boldsymbol{\xi}_{i'}\|_2^{-2} \cdot \langle \boldsymbol{\xi}_{i'}, \boldsymbol{\xi}_i\rangle$$

$$\leq 4\sqrt{\frac{\log(4n^2/\delta)}{d}} \sum_{i' \neq i} |\overline{\rho}_{j,r,i'}^{(t)}| + 4\sqrt{\frac{\log(4n^2/\delta)}{d}} \sum_{i' \neq i} |\underline{\rho}_{j,r,i'}^{(t)}| + \underline{\rho}_{j,r,i}^{(t)}$$

$$\leq \underline{\rho}_{j,r,i}^{(t)} + 8n\sqrt{\frac{\log(4n^2/\delta)}{d}} \alpha,$$

where the second inequality is by Lemma B.2 and the last inequality is by $|\overline{\rho}_{j,r,i'}^{(t)}|, |\underline{\rho}_{j,r,i'}^{(t)}| \leq \alpha$ in (C.10) Similarly, for $y_i = j$, we have that $\underline{\rho}_{j,r,i}^{(t)} = 0$ and

$$\langle \mathbf{w}_{j,r}^{(t)} - \mathbf{w}_{j,r}^{(0)}, \boldsymbol{\xi}_i \rangle = \sum_{i'=1}^{n} \overline{\rho}_{j,r,i'}^{(t)} \|\boldsymbol{\xi}_{i'}\|_2^{-2} \cdot \langle \boldsymbol{\xi}_{i'}, \boldsymbol{\xi}_i \rangle + \sum_{i'=1}^{n} \underline{\rho}_{j,r,i'}^{(t)} \|\boldsymbol{\xi}_{i'}\|_2^{-2} \cdot \langle \boldsymbol{\xi}_{i'}, \boldsymbol{\xi}_i \rangle$$

$$\leq \overline{\rho}_{j,r,i}^{(t)} + 4\sqrt{\frac{\log(4n^2/\delta)}{d}} \sum_{i' \neq i} |\overline{\rho}_{j,r,i'}^{(t)}| + 4\sqrt{\frac{\log(4n^2/\delta)}{d}} \sum_{i' \neq i} |\underline{\rho}_{j,r,i'}^{(t)}|$$

$$\leq \overline{\rho}_{j,r,i}^{(t)} + 8n\sqrt{\frac{\log(4n^2/\delta)}{d}} \alpha,$$

where the first inequality is by Lemma B.1 and the second inequality is by $|\overline{\rho}_{j,r,i'}^{(t)}|, |\underline{\rho}_{j,r,i'}^{(t)}| \leq \alpha$ in (C.10). Similarly, we can show that $\langle \mathbf{w}_{j,r}^{(t)} - \mathbf{w}_{j,r}^{(0)}, \boldsymbol{\xi}_i \rangle \geq \underline{\rho}_{j,r,i}^{(t)} - 8n\sqrt{\log(4n^2/\delta)/d} \cdot \alpha$ and $\langle \mathbf{w}_{j,r}^{(t)} - \mathbf{w}_{j,r}^{(0)}, \boldsymbol{\xi}_i \rangle \geq \overline{\rho}_{j,r,i}^{(t)} - 8n\sqrt{\log(4n^2/\delta)/d} \cdot \alpha$, which completes the proof. $\qquad\square$

**Lemma C.5.** *Under Condition 4.2, suppose* (C.10) *and* (C.11) *hold at iteration $t$. Then*

$$\langle \mathbf{w}_{j,r}^{(t)}, y_i \boldsymbol{\mu} \rangle \leq \langle \mathbf{w}_{j,r}^{(0)}, y_i \boldsymbol{\mu} \rangle,$$

$$\langle \mathbf{w}_{j,r}^{(t)}, \boldsymbol{\xi}_i \rangle \leq \langle \mathbf{w}_{j,r}^{(0)}, \boldsymbol{\xi}_i \rangle + 8n\sqrt{\frac{\log(4n^2/\delta)}{d}} \alpha,$$

$$F_j(\mathbf{W}_j^{(t)}, \mathbf{x}_i) \leq 1$$

*for all $r \in [m]$ and $j \neq y_i$.*

*Proof of Lemma C.5.* For $j \neq y_i$, we have that

$$\langle \mathbf{w}_{j,r}^{(t)}, y_i \boldsymbol{\mu} \rangle = \langle \mathbf{w}_{j,r}^{(0)}, y_i \boldsymbol{\mu} \rangle + y_i \cdot j \cdot \gamma_{j,r}^{(t)} \leq \langle \mathbf{w}_{j,r}^{(0)}, y_i \boldsymbol{\mu} \rangle, \tag{C.12}$$

where the inequality is by $\gamma_{j,r}^{(t)} \geq 0$. In addition, we have

$$\langle \mathbf{w}_{j,r}^{(t)}, \boldsymbol{\xi}_i \rangle \leq \langle \mathbf{w}_{j,r}^{(0)}, \boldsymbol{\xi}_i \rangle + \underline{\rho}_{j,r,i}^{(t)} + 8n\sqrt{\frac{\log(4n^2/\delta)}{d}} \alpha \leq \langle \mathbf{w}_{j,r}^{(0)}, \boldsymbol{\xi}_i \rangle + 8n\sqrt{\frac{\log(4n^2/\delta)}{d}} \alpha, \tag{C.13}$$

where the first inequality is by Lemma C.4 and the second inequality is due to $\underline{\rho}_{j,r,i}^{(t)} \leq 0$. Then we can get that

$$F_j(\mathbf{W}_j^{(t)}, \mathbf{x}_i) = \frac{1}{m} \sum_{r=1}^{m} [\sigma(\langle \mathbf{w}_{j,r}^{(t)}, -j \cdot \boldsymbol{\mu} \rangle) + \sigma(\langle \mathbf{w}_{j,r}^{(t)}, \boldsymbol{\xi}_i \rangle)]$$

$$\leq 2^{q+1} \max_{j,r,i} \left\{ |\langle \mathbf{w}_{j,r}^{(0)}, \boldsymbol{\mu} \rangle|, |\langle \mathbf{w}_{j,r}^{(0)}, \boldsymbol{\xi}_i \rangle|, 8n\sqrt{\frac{\log(4n^2/\delta)}{d}} \alpha \right\}^{q}$$

$$\leq 1,$$

where the first inequality is by (C.12), (C.13) and the second inequality is by (C.9). $\qquad\square$

**Lemma C.6.** *Under Condition 4.2, suppose* (C.10) *and* (C.11) *hold at iteration $t$. Then*

$$\langle \mathbf{w}_{j,r}^{(t)}, y_i \boldsymbol{\mu} \rangle = \langle \mathbf{w}_{j,r}^{(0)}, y_i \boldsymbol{\mu} \rangle + \gamma_{j,r}^{(t)},$$

$$\langle \mathbf{w}_{j,r}^{(t)}, \boldsymbol{\xi}_i \rangle \leq \langle \mathbf{w}_{j,r}^{(0)}, \boldsymbol{\xi}_i \rangle + \overline{\rho}_{j,r,i}^{(t)} + 8n\sqrt{\frac{\log(4n^2/\delta)}{d}} \alpha$$

*for all $r \in [m]$, $j \in \{\pm 1\}$ and $i \in [n]$. If $\max\{\gamma_{j,r}^{(t)}, \overline{\rho}_{j,r,i}^{(t)}\} = O(1)$, we further have that $F_j(\mathbf{W}_j^{(t)}, \mathbf{x}_i) = O(1)$.*

*Proof of Lemma C.6.* For $j = y_i$, we have that

$$\langle \mathbf{w}_{j,r}^{(t)}, y_i \boldsymbol{\mu} \rangle = \langle \mathbf{w}_{j,r}^{(0)}, y_i \boldsymbol{\mu} \rangle + \gamma_{j,r}^{(t)}, \tag{C.14}$$

where the equation is by Lemma C.3. We also have that

$$\langle \mathbf{w}_{j,r}^{(t)}, \boldsymbol{\xi}_i \rangle \leq \langle \mathbf{w}_{j,r}^{(0)}, \boldsymbol{\xi}_i \rangle + \overline{\rho}_{j,r,i}^{(t)} + 8n\sqrt{\frac{\log(4n^2/\delta)}{d}}\alpha, \tag{C.15}$$

where the inequality is by Lemma C.4. If $\max\{\gamma_{j,r}^{(t)}, \overline{\rho}_{j,r,i}^{(t)}\} = O(1)$, we have following bound

$$F_j(\mathbf{W}_j^{(t)}, \mathbf{x}_i) = \frac{1}{m}\sum_{r=1}^{m}[\sigma(\langle \mathbf{w}_{j,r}^{(t)}, -j \cdot \boldsymbol{\mu}\rangle) + \sigma(\langle \mathbf{w}_{j,r}^{(t)}, \boldsymbol{\xi}_i\rangle)]$$

$$\leq 2 \cdot 3^q \max_{j,r,i}\left\{\gamma_{j,r}^{(t)}, \overline{\rho}_{j,r,i}^{(t)}, |\langle \mathbf{w}_{j,r}^{(0)}, \boldsymbol{\mu}\rangle|, |\langle \mathbf{w}_{j,r}^{(0)}, \boldsymbol{\xi}_i\rangle|, 8n\sqrt{\frac{\log(4n^2/\delta)}{d}}\alpha\right\}^q$$

$$= O(1),$$

where the first inequality is by (C.14), (C.15) and the second inequality is by (C.9) where $\beta = 2\max_{i,j,r}\{|\langle \mathbf{w}_{j,r}^{(0)}, \boldsymbol{\mu}\rangle|, |\langle \mathbf{w}_{j,r}^{(0)}, \boldsymbol{\xi}_i\rangle|\}$. $\qquad\square$

Now we are ready to prove Proposition C.2.

*Proof of Proposition C.2.* Our proof is based on induction. The results are obvious at $t = 0$ as all the coefficients are zero. Suppose that there exists $\widetilde{T} \leq T^*$ such that the results in Proposition C.2 hold for all time $0 \leq t \leq \widetilde{T} - 1$. We aim to prove that they also hold for $t = \widetilde{T}$.

We first prove that (C.11) holds for $t = \widetilde{T}$, i.e., $\underline{\rho}_{j,r,i}^{(t)} \geq -\beta - 16n\sqrt{\frac{\log(4n^2/\delta)}{d}}\alpha$ for $t = \widetilde{T}$, $r \in [m]$, $j \in \{\pm 1\}$ and $i \in [n]$. Notice that $\underline{\rho}_{j,r,i}^{(t)} = 0, \forall j = y_i$. Therefore, we only need to consider the case that $j \neq y_i$. When $\underline{\rho}_{j,r,i}^{(\widetilde{T}-1)} \leq -0.5\beta - 8n\sqrt{\frac{\log(4n^2/\delta)}{d}}\alpha$, by Lemma C.4 we have that

$$\langle \mathbf{w}_{j,r}^{(\widetilde{T}-1)}, \boldsymbol{\xi}_i \rangle \leq \underline{\rho}_{j,r,i}^{(\widetilde{T}-1)} + \langle \mathbf{w}_{j,r}^{(0)}, \boldsymbol{\xi}_i \rangle + 8n\sqrt{\frac{\log(4n^2/\delta)}{d}}\alpha \leq 0,$$

and thus

$$\underline{\rho}_{j,r,i}^{(\widetilde{T})} = \underline{\rho}_{j,r,i}^{(\widetilde{T}-1)} + \frac{\eta}{nm} \cdot \ell_i'^{(\widetilde{T}-1)} \cdot \sigma'(\langle \mathbf{w}_{j,r}^{(\widetilde{T}-1)}, \boldsymbol{\xi}_i\rangle) \cdot \mathbb{1}(y_i = -j)\|\boldsymbol{\xi}_i\|_2^2$$

$$= \underline{\rho}_{j,r,i}^{(\widetilde{T}-1)}$$

$$\geq -\beta - 16n\sqrt{\frac{\log(4n^2/\delta)}{d}}\alpha,$$

where the last inequality is by induction hypothesis. When $\underline{\rho}_{j,r,i}^{(\widetilde{T}-1)} \geq -0.5\beta - 8n\sqrt{\frac{\log(4n^2/\delta)}{d}}\alpha$, we have that

$$\underline{\rho}_{j,r,i}^{(\widetilde{T})} = \underline{\rho}_{j,r,i}^{(\widetilde{T}-1)} + \frac{\eta}{nm} \cdot \ell_i'^{(\widetilde{T}-1)} \cdot \sigma'(\langle \mathbf{w}_{j,r}^{(T-1)}, \boldsymbol{\xi}_i\rangle) \cdot \mathbb{1}(y_i = -j)\|\boldsymbol{\xi}_i\|_2^2$$

$$\geq -0.5\beta - 8n\sqrt{\frac{\log(4n^2/\delta)}{d}}\alpha - O\left(\frac{\eta\sigma_p^2 d}{nm}\right)\sigma'\left(0.5\beta + 8n\sqrt{\frac{\log(4n^2/\delta)}{d}}\alpha\right)$$

$$\geq -0.5\beta - 8n\sqrt{\frac{\log(4n^2/\delta)}{d}}\alpha - O\left(\frac{\eta q\sigma_p^2 d}{nm}\right)\left(0.5\beta + 8n\sqrt{\frac{\log(4n^2/\delta)}{d}}\alpha\right)$$

$$\geq -\beta - 16n\sqrt{\frac{\log(4n^2/\delta)}{d}}\alpha,$$

where we use $-\ell_i'^{(\widetilde{T}-1)} \leq 1$ and $\|\boldsymbol{\xi}_i\|_2 = O(\sigma_p^2 d)$ in the first inequality, the second inequality is by $0.5\beta + 8n\sqrt{\frac{\log(4n^2/\delta)}{d}}\alpha \leq 1$, and the last inequality is by $\eta = O\big(nm/(q\sigma_p^2 d)\big)$ in (C.6).

Next we prove (C.10) holds for $t = \widetilde{T}$. We have

$$
\begin{aligned}
|\ell_i'^{(t)}| &= \frac{1}{1 + \exp\{y_i \cdot [F_{+1}(\mathbf{W}_{+1}^{(t)}, \mathbf{x}_i) - F_{-1}(\mathbf{W}_{-1}^{(t)}, \mathbf{x}_i)]\}} \\
&\leq \exp\{-y_i \cdot [F_{+1}(\mathbf{W}_{+1}^{(t)}, \mathbf{x}_i) - F_{-1}(\mathbf{W}_{-1}^{(t)}, \mathbf{x}_i)]\} \\
&\leq \exp\{-F_{y_i}(\mathbf{W}_{y_i}^{(t)}, \mathbf{x}_i) + 1\}.
\end{aligned}
\tag{C.16}
$$

where the last inequality is due to Lemma C.5. Moreover, recall the update rule of $\gamma_{j,r}^{(t)}$ and $\overline{\rho}_{j,r,i}^{(t)}$,

$$
\gamma_{j,r}^{(t+1)} = \gamma_{j,r}^{(t)} - \frac{\eta}{nm} \cdot \sum_{i=1}^{n} \ell_i'^{(t)} \cdot \sigma'(\langle \mathbf{w}_{j,r}^{(t)}, y_i \cdot \boldsymbol{\mu} \rangle) \|\boldsymbol{\mu}\|_2^2,
$$

$$
\overline{\rho}_{j,r,i}^{(t+1)} = \overline{\rho}_{j,r,i}^{(t)} - \frac{\eta}{nm} \cdot \ell_i'^{(t)} \cdot \sigma'(\langle \mathbf{w}_{j,r}^{(t)}, \boldsymbol{\xi}_i \rangle) \cdot \mathbb{1}(y_i = j) \|\boldsymbol{\xi}_i\|_2^2.
$$

Let $t_{j,r,i}$ be the last time $t < T^*$ that $\overline{\rho}_{j,r,i}^{(t)} \leq 0.5\alpha$. Then we have that

$$
\overline{\rho}_{j,r,i}^{(\widetilde{T})} = \overline{\rho}_{j,r,i}^{(t_{j,r,i})} \underbrace{- \frac{\eta}{nm} \cdot \ell_i'^{(t_{j,r,i})} \cdot \sigma'(\langle \mathbf{w}_{j,r}^{(t_{j,r,i})}, \boldsymbol{\xi}_i \rangle) \cdot \mathbb{1}(y_i = j) \|\boldsymbol{\xi}_i\|_2^2}_{I_1}
$$

$$
\underbrace{- \sum_{t_{j,r,i} < t < T} \frac{\eta}{nm} \cdot \ell_i'^{(t)} \cdot \sigma'(\langle \mathbf{w}_{j,r}^{(t)}, \boldsymbol{\xi}_i \rangle) \cdot \mathbb{1}(y_i = j) \|\boldsymbol{\xi}_i\|_2^2}_{I_2}.
\tag{C.17}
$$

We first bound $I_1$ as follows,

$$
|I_1| \leq 2qn^{-1}m^{-1}\eta \left( \overline{\rho}_{j,r,i}^{(t_{j,r,i})} + 0.5\beta + 8n\sqrt{\frac{\log(4n^2/\delta)}{d}}\alpha \right)^{q-1} \sigma_p^2 d \leq q2^q n^{-1}m^{-1}\eta\alpha^{q-1}\sigma_p^2 d \leq 0.25\alpha,
$$

where the first inequality is by Lemmas C.4 and B.2, the second inequality is by $\beta \leq 0.1\alpha$ and $8n\sqrt{\frac{\log(4n^2/\delta)}{d}}\alpha \leq 0.1\alpha$, the last inequality is by $\eta \leq nm/(q2^{q+2}\alpha^{q-2}\sigma_p^2 d)$.

Second, we bound $I_2$. For $t_{j,r,i} < t < \widetilde{T}$ and $y_i = j$, we can lower bound $\langle \mathbf{w}_{j,r}^{(t)}, \boldsymbol{\xi}_i \rangle$ as follows,

$$
\begin{aligned}
\langle \mathbf{w}_{j,r}^{(t)}, \boldsymbol{\xi}_i \rangle &\geq \langle \mathbf{w}_{j,r}^{(0)}, \boldsymbol{\xi}_i \rangle + \overline{\rho}_{j,r,i}^{(t)} - 8n\sqrt{\frac{\log(4n^2/\delta)}{d}}\alpha \\
&\geq -0.5\beta + 0.5\alpha - 8n\sqrt{\frac{\log(4n^2/\delta)}{d}}\alpha \\
&\geq 0.25\alpha,
\end{aligned}
$$

where the first inequality is by Lemma C.4, the second inequality is by $\overline{\rho}_{j,r,i}^{(t)} > 0.5\alpha$ and $\langle \mathbf{w}_{j,r}^{(0)}, \boldsymbol{\xi}_i \rangle \geq -0.5\beta$ due to the definition of $t_{j,r,i}$ and $\beta$, the last inequality is by $\beta \leq 0.1\alpha$ and $8n\sqrt{\frac{\log(4n^2/\delta)}{d}}\alpha \leq 0.1\alpha$. Similarly, for $t_{j,r,i} < t < \widetilde{T}$ and $y_i = j$, we can also upper bound $\langle \mathbf{w}_{j,r}^{(t)}, \boldsymbol{\xi}_i \rangle$ as follows,

$$
\begin{aligned}
\langle \mathbf{w}_{j,r}^{(t)}, \boldsymbol{\xi}_i \rangle &\leq \langle \mathbf{w}_{j,r}^{(0)}, \boldsymbol{\xi}_i \rangle + \overline{\rho}_{j,r,i}^{(t)} + 8n\sqrt{\frac{\log(4n^2/\delta)}{d}}\alpha \\
&\leq 0.5\beta + \alpha + 8n\sqrt{\frac{\log(4n^2/\delta)}{d}}\alpha \\
&\leq 2\alpha,
\end{aligned}
$$

where the first inequality is by Lemma C.4, the second inequality is by induction hypothesis $\overline{\rho}_{j,r,i}^{(t)} \leq \alpha$, the last inequality is by $\beta \leq 0.1\alpha$ and $8n\sqrt{\frac{\log(4n^2/\delta)}{d}}\alpha \leq 0.1\alpha$. Thus, plugging the upper and lower bounds of $\langle \mathbf{w}_{j,r}^{(t)}, \boldsymbol{\xi}_i \rangle$ into $I_2$ gives

$$
|I_2| \leq \sum_{t_{j,r,i} < t < \widetilde{T}} \frac{\eta}{nm} \cdot \exp(-\sigma(\langle \mathbf{w}_{j,r}^{(t)}, \boldsymbol{\xi}_i \rangle) + 1) \cdot \sigma'(\langle \mathbf{w}_{j,r}^{(t)}, \boldsymbol{\xi}_i \rangle) \cdot \mathbb{1}(y_i = j) \|\boldsymbol{\xi}_i\|_2^2
$$

$$\leq \frac{eq2^q\eta T^*}{nm}\exp(-\alpha^q/4^q)\alpha^{q-1}\sigma_p^2 d$$
$$\leq 0.25T^*\exp(-\alpha^q/4^q)\alpha$$
$$\leq 0.25T^*\exp(-\log(T^*)^q)\alpha$$
$$\leq 0.25\alpha,$$

where the first inequality is by (C.16), the second inequality is by Lemma B.2, the third inequality is by $\eta = O\big(nm/(q2^{q+2}\alpha^{q-2}\sigma_p^2 d)\big)$ in (C.6), the fourth inequality is by our choice of $\alpha = 4\log(T^*)$ and the last inequality is due to the fact that $\log(T^*)^q \geq \log(T^*)$. Plugging the bound of $I_1, I_2$ into (C.17) completes the proof for $\overline{\rho}$. Similarly, we can prove that $\gamma_{j,r}^{(\widetilde{T})} \leq \alpha$ using $\eta = O\big(nm/(q2^{q+2}\alpha^{q-2}\|\boldsymbol{\mu}\|_2^2)\big)$ in (C.6). Therefore Proposition C.2 holds for $t = \widetilde{T}$, which completes the induction. $\qquad\square$

Based on Proposition C.2, we introduce some important properties of the training loss function for $0 \leq t \leq T^*$.

**Lemma C.7** (Restatement of Lemma 5.4). *Under Condition 4.2, for $0 \leq t \leq T^*$, the following result holds.*

$$\|\nabla L_S(\mathbf{W}^{(t)})\|_F^2 \leq O(\max\{\|\boldsymbol{\mu}\|_2^2, \sigma_p^2 d\})L_S(\mathbf{W}^{(t)}).$$

*Proof of Lemma C.7.* We first prove that

$$-\ell'\big(y_i f(\mathbf{W}^{(t)}, \mathbf{x}_i)\big) \cdot \|\nabla f(\mathbf{W}^{(t)}, \mathbf{x}_i)\|_F^2 = O(\max\{\|\boldsymbol{\mu}\|_2^2, \sigma_p^2 d\}). \qquad (C.18)$$

Without loss of generality, we suppose that $y_i = 1$ and $\mathbf{x}_i = [\boldsymbol{\mu}^\top, \boldsymbol{\xi}_i]$. Then we have that

$$\|\nabla f(\mathbf{W}^{(t)}, \mathbf{x}_i)\|_F \leq \frac{1}{m}\sum_{j,r}\left\|\left[\sigma'(\langle\mathbf{w}_{j,r}^{(t)}, \boldsymbol{\mu}\rangle)\boldsymbol{\mu} + \sigma'(\langle\mathbf{w}_{j,r}^{(t)}, \boldsymbol{\xi}_i\rangle)\boldsymbol{\xi}_i\right]\right\|_2$$

$$\leq \frac{1}{m}\sum_{j,r}\sigma'(\langle\mathbf{w}_{j,r}^{(t)}, \boldsymbol{\mu}\rangle)\|\boldsymbol{\mu}\|_2 + \frac{1}{m}\sum_{j,r}\sigma'(\langle\mathbf{w}_{j,r}^{(t)}, \boldsymbol{\xi}_i\rangle)\|\boldsymbol{\xi}_i\|_2$$

$$\leq 2q\left[F_{+1}(\mathbf{W}_{+1}^{(t)}, \mathbf{x}_i)\right]^{(q-1)/q}\max\{\|\boldsymbol{\mu}\|_2, 2\sigma_p\sqrt{d}\}$$

$$\qquad + 2q\left[F_{-1}(\mathbf{W}_{-1}^{(t)}, \mathbf{x}_i)\right]^{(q-1)/q}\max\{\|\boldsymbol{\mu}\|_2, 2\sigma_p\sqrt{d}\}$$

$$\leq 2q\left[F_{+1}(\mathbf{W}_{+1}^{(t)}, \mathbf{x}_i)\right]^{(q-1)/q}\max\{\|\boldsymbol{\mu}\|_2, 2\sigma_p\sqrt{d}\} + 2q\max\{\|\boldsymbol{\mu}\|_2, 2\sigma_p\sqrt{d}\},$$

where the first and second inequalities are by triangle inequality, the third inequality is by Jensen's inequality and Lemma B.2, and the last inequality is due to Lemma C.5. Denote $A = F_{+1}(\mathbf{W}_{+1}^{(t)}, \mathbf{x}_i)$. Then we have that $A \geq 0$, and besides, $F_{-1}(\mathbf{W}_{-1}^{(t)}, \mathbf{x}_i) \leq 1$ by Lemma C.5. Then we have that

$$-\ell'\big(y_i f(\mathbf{W}^{(t)}, \mathbf{x}_i)\big) \cdot \|\nabla f(\mathbf{W}^{(t)}, \mathbf{x}_i)\|_F^2$$

$$\leq -\ell'(A-1)\left(2q \cdot A^{(q-1)/q}\max\{\|\boldsymbol{\mu}\|_2, 2\sigma_p\sqrt{d}\} + 2q \cdot \max\{\|\boldsymbol{\mu}\|_2, 2\sigma_p\sqrt{d}\}\right)^2$$

$$= -4q^2\ell'(A-1)(A^{(q-1)/q}+1)^2 \cdot \max\{\|\boldsymbol{\mu}\|_2^2, 4\sigma_p^2 d\}$$

$$\leq \left(\max_{z>0} -4q^2\ell'(z-1)(z^{(q-1)/q}+1)^2\right)\max\{\|\boldsymbol{\mu}\|_2^2, 4\sigma_p^2 d\}$$

$$\overset{(i)}{=} O(\max\{\|\boldsymbol{\mu}\|_2^2, \sigma_p^2 d\}),$$

where (i) is by $\max_{z\geq 0} -4q^2\ell'(z-1)(z^{(q-1)/q}+1)^2 < \infty$ because $\ell'$ has an exponentially decaying tail. Now we can upper bound the gradient norm $\|\nabla L_S(\mathbf{W}^{(t)})\|_F$ as follows,

$$\|\nabla L_S(\mathbf{W}^{(t)})\|_F^2 \leq \left[\frac{1}{n}\sum_{i=1}^{n}\ell'\big(y_i f(\mathbf{W}^{(t)}, \mathbf{x}_i)\big)\|\nabla f(\mathbf{W}^{(t)}, \mathbf{x}_i)\|_F\right]^2$$

$$\leq \left[ \frac{1}{n} \sum_{i=1}^{n} \sqrt{-O(\max\{\|\boldsymbol{\mu}\|_2^2, \sigma_p^2 d\}) \ell'\big(y_i f(\mathbf{W}^{(t)}, \mathbf{x}_i)\big)} \right]^2$$

$$\leq O(\max\{\|\boldsymbol{\mu}\|_2^2, \sigma_p^2 d\}) \cdot \frac{1}{n} \sum_{i=1}^{n} -\ell'\big(y_i f(\mathbf{W}^{(t)}, \mathbf{x}_i)\big)$$

$$\leq O(\max\{\|\boldsymbol{\mu}\|_2^2, \sigma_p^2 d\}) L_S(\mathbf{W}^{(t)}),$$

where the first inequality is by triangle inequality, the second inequality is by (C.18), the third inequality is by Cauchy-Schwartz inequality and the last inequality is due to the property of the cross entropy loss $-\ell' \leq \ell$. $\qquad\square$

## D  Signal Learning

In this section, we consider the signal learning case under the condition that $n\|\boldsymbol{\mu}\|_2^q \geq \widetilde{\Omega}(\sigma_p^q(\sqrt{d})^q)$. We remind the readers that the proofs in this section are based on the results in Section B, which hold with high probability.

### D.1  First stage

**Lemma D.1** (Restatement of Lemma 5.5). *Under the same conditions as Theorem 4.3, in particular if we choose*

$$n \cdot \mathrm{SNR}^q \geq C \log(6/\sigma_0\|\boldsymbol{\mu}\|_2) 2^{2q+6} [4\log(8mn/\delta)]^{(q-1)/2}, \tag{D.1}$$

*where $C = O(1)$ is a positive constant, there exists time*

$$T_1 = \frac{C \log(6/\sigma_0\|\boldsymbol{\mu}\|_2) 2^{q+1} m}{\eta \sigma_0^{q-2} \|\boldsymbol{\mu}\|_2^q}$$

*such that*

- $\max_r \gamma_{j,r}^{(T_1)} \geq 2$ *for $j \in \{\pm 1\}$.*
- $|\rho_{j,r,i}^{(t)}| \leq \sigma_0 \sigma_p \sqrt{d}/2$ *for all $j \in \{\pm 1\}, r \in [m], i \in [n]$ and $0 \leq t \leq T_1$.*

*Proof of Lemma D.1.* Let

$$T_1^+ = \frac{nm\eta^{-1}\sigma_0^{2-q}\sigma_p^{-q}d^{-q/2}}{2^{q+4}q[4\log(8mn/\delta)]^{(q-1)/2}}. \tag{D.2}$$

We first prove the second bullet. Define $\Psi^{(t)} = \max_{j,r,i} |\rho_{j,r,i}^{(t)}| = \max_{j,r,i}\{\overline{\rho}_{j,r,i}^{(t)}, -\underline{\rho}_{j,r,i}^{(t)}\}$. We use induction to show that

$$\Psi^{(t)} \leq \sigma_0 \sigma_p \sqrt{d}/2 \tag{D.3}$$

for all $0 \leq t \leq T_1^+$. By definition, clearly we have $\Psi^{(0)} = 0$. Now suppose that there exists some $\widetilde{T} \leq T_1^+$ such that (D.3) holds for $0 < t \leq \widetilde{T} - 1$. Then by (C.4) and (C.5) we have

$$\Psi^{(t+1)} \leq \Psi^{(t)} + \max_{j,r,i}\left\{ \frac{\eta}{nm} \cdot |\ell_i'^{(t)}| \cdot \sigma'\left( \langle \mathbf{w}_{j,r}^{(0)}, \boldsymbol{\xi}_i \rangle + \sum_{i'=1}^{n} \Psi^{(t)} \cdot \frac{|\langle \boldsymbol{\xi}_{i'}, \boldsymbol{\xi}_i \rangle|}{\|\boldsymbol{\xi}_{i'}\|_2^2} + \sum_{i'=1}^{n} \Psi^{(t)} \cdot \frac{|\langle \boldsymbol{\xi}_{i'}, \boldsymbol{\xi}_i \rangle|}{\|\boldsymbol{\xi}_{i'}\|_2^2} \right) \cdot \|\boldsymbol{\xi}_i\|_2^2 \right\}$$

$$\leq \Psi^{(t)} + \max_{j,r,i}\left\{ \frac{\eta}{nm} \cdot \sigma'\left( \langle \mathbf{w}_{j,r}^{(0)}, \boldsymbol{\xi}_i \rangle + 2 \cdot \sum_{i'=1}^{n} \Psi^{(t)} \cdot \frac{|\langle \boldsymbol{\xi}_{i'}, \boldsymbol{\xi}_i \rangle|}{\|\boldsymbol{\xi}_{i'}\|_2^2} \right) \cdot \|\boldsymbol{\xi}_i\|_2^2 \right\}$$

$$= \Psi^{(t)} + \max_{j,r,i}\left\{ \frac{\eta}{nm} \cdot \sigma'\left( \langle \mathbf{w}_{j,r}^{(0)}, \boldsymbol{\xi}_i \rangle + 2\Psi^{(t)} + 2 \cdot \sum_{i'\neq i}^{n} \Psi^{(t)} \cdot \frac{|\langle \boldsymbol{\xi}_{i'}, \boldsymbol{\xi}_i \rangle|}{\|\boldsymbol{\xi}_{i'}\|_2^2} \right) \cdot \|\boldsymbol{\xi}_i\|_2^2 \right\}$$

$$\leq \Psi^{(t)} + \frac{\eta q}{nm} \cdot \left[ 2 \cdot \sqrt{\log(8mn/\delta)} \cdot \sigma_0\sigma_p\sqrt{d} + \left( 2 + \frac{4n\sigma_p^2 \cdot \sqrt{d\log(4n^2/\delta)}}{\sigma_p^2 d/2} \right) \cdot \Psi^{(t)} \right]^{q-1} \cdot 2\sigma_p^2 d$$

$$\leq \Psi^{(t)} + \frac{\eta q}{nm} \cdot \left(2 \cdot \sqrt{\log(8mn/\delta)} \cdot \sigma_0 \sigma_p \sqrt{d} + 4\Psi^{(t)}\right)^{q-1} \cdot 2\sigma_p^2 d$$

$$\leq \Psi^{(t)} + \frac{\eta q}{nm} \cdot \left(4 \cdot \sqrt{\log(8mn/\delta)} \cdot \sigma_0 \sigma_p \sqrt{d}\right)^{q-1} \cdot 2\sigma_p^2 d,$$

where the second inequality is by $|\ell_i'^{(t)}| \leq 1$, the third inequality is due to Lemmas B.2 and B.3, the fourth inequality follows by the condition that $d \geq 16n^2 \log(4n^2/\delta)$, and the last inequality follows by the induction hypothesis (D.3). Taking a telescoping sum over $t = 0, 1, \ldots, \widetilde{T} - 1$ then gives

$$\Psi^{(\widetilde{T})} \leq \widetilde{T} \cdot \frac{\eta q}{nm} \cdot \left(4 \cdot \sqrt{\log(8mn/\delta)} \cdot \sigma_0 \sigma_p \sqrt{d}\right)^{q-1} \cdot 2\sigma_p^2 d$$

$$\leq T_1^+ \cdot \frac{\eta q}{nm} \cdot \left(4 \cdot \sqrt{\log(8mn/\delta)} \cdot \sigma_0 \sigma_p \sqrt{d}\right)^{q-1} \cdot 2\sigma_p^2 d$$

$$\leq \frac{\sigma_0 \sigma_p \sqrt{d}}{2},$$

where the second inequality follows by $\widetilde{T} \leq T_1^+$ in our induction hypothesis. Therefore, by induction, we have $\Psi^{(t)} \leq \sigma_0 \sigma_p \sqrt{d}/2$ for all $t \leq T_1^+$.

Now, without loss of generality, let us consider $j = 1$ first. Denote by $T_{1,1}$ the last time for $t$ in the period $[0, T_1^+]$ satisfying that $\max_r \gamma_{1,r}^{(t)} \leq 2$. Then for $t \leq T_{1,1}$, $\max_{j,r,i}\{|\rho_{j,r,i}^{(t)}|\} = O(\sigma_0 \sigma_p \sqrt{d}) = O(1)$ and $\max_r \gamma_{1,r}^{(t)} \leq 2$. Therefore, by Lemmas C.5 and C.6, we know that $F_{-1}(\mathbf{W}_{-1}^{(t)}, \mathbf{x}_i), F_{+1}(\mathbf{W}_{+1}^{(t)}, \mathbf{x}_i) = O(1)$ for all $i$ with $y_i = 1$. Thus there exists a positive constant $C_1$ such that $-\ell_i'^{(t)} \geq C_1$ for all $i$ with $y_i = 1$.

By (C.3), for $t \leq T_{1,1}$ we have

$$\gamma_{1,r}^{(t+1)} = \gamma_{1,r}^{(t)} - \frac{\eta}{nm} \cdot \sum_{i=1}^n \ell_i'^{(t)} \cdot \sigma'(y_i \cdot \langle \mathbf{w}_{1,r}^{(0)}, \boldsymbol{\mu} \rangle + y_i \cdot \gamma_{1,r}^{(t)}) \cdot \|\boldsymbol{\mu}\|_2^2$$

$$\geq \gamma_{1,r}^{(t)} + \frac{C_1 \eta}{nm} \cdot \sum_{y_i=1} \sigma'(\langle \mathbf{w}_{1,r}^{(0)}, \boldsymbol{\mu} \rangle + \gamma_{1,r}^{(t)}) \cdot \|\boldsymbol{\mu}\|_2^2.$$

Denote $\widehat{\gamma}_{1,r}^{(t)} = \gamma_{1,r}^{(t)} + \langle \mathbf{w}_{1,r}^{(0)}, \boldsymbol{\mu} \rangle$ and let $A^{(t)} = \max_r \widehat{\gamma}_{1,r}^{(t)}$. Then we have

$$A^{(t+1)} \geq A^{(t)} + \frac{C_1 \eta}{nm} \cdot \sum_{y_i=1} \sigma'(A^{(t)}) \cdot \|\boldsymbol{\mu}\|_2^2$$

$$\geq A^{(t)} + \frac{C_1 \eta q \|\boldsymbol{\mu}\|_2^2}{4m} \left[A^{(t)}\right]^{q-1}$$

$$\geq \left(1 + \frac{C_1 \eta q \|\boldsymbol{\mu}\|_2^2}{4m} \left[A^{(0)}\right]^{q-2}\right) A^{(t)}$$

$$\geq \left(1 + \frac{C_1 \eta q \sigma_0^{q-2} \|\boldsymbol{\mu}\|_2^q}{2^q m}\right) A^{(t)},$$

where the second inequality is by the lower bound on the number of positive data in Lemma B.1, the third inequality is due to the fact that $A^{(t)}$ is an increasing sequence, and the last inequality follows by $A^{(0)} = \max_r \langle \mathbf{w}_{1,r}^{(0)}, \boldsymbol{\mu} \rangle \geq \sigma_0 \|\boldsymbol{\mu}\|_2/2$ proved in Lemma B.3. Therefore, the sequence $A^{(t)}$ will exponentially grow and we have that

$$A^{(t)} \geq \left(1 + \frac{C_1 \eta q \sigma_0^{q-2} \|\boldsymbol{\mu}\|_2^q}{2^q m}\right)^t A^{(0)} \geq \exp\left(\frac{C_1 \eta q \sigma_0^{q-2} \|\boldsymbol{\mu}\|_2^q}{2^{q+1} m} t\right) A^{(0)} \geq \exp\left(\frac{C_1 \eta q \sigma_0^{q-2} \|\boldsymbol{\mu}\|_2^q}{2^{q+1} m} t\right) \frac{\sigma_0 \|\boldsymbol{\mu}\|_2}{2},$$

where the second inequality is due to the fact that $1 + z \geq \exp(z/2)$ for $z \leq 2$ and our condition of $\eta$ and $\sigma_0$ listed in Condition 4.2, and the last inequality follows by Lemma B.3 and $A^{(0)} = \max_r \langle \mathbf{w}_{1,r}^{(0)}, \boldsymbol{\mu} \rangle$. Therefore, $A^{(t)}$ will reach 3 within $T_1 = \frac{\log(6/\sigma_0 \|\boldsymbol{\mu}\|_2) 2^{q+1} m}{C_1 \eta q \sigma_0^{q-2} \|\boldsymbol{\mu}\|_2^q}$ iterations. Since $\max_r \gamma_{1,r}^{(t)} \geq A^{(t)} - \max_r |\langle \mathbf{w}_{1,r}^{(0)}, \boldsymbol{\mu} \rangle| \geq A^{(t)} - 1$, $\max_r \gamma_{1,r}^{(t)}$ will reach 2 within $T_1$ iterations. We can next verify that

$$T_1 = \frac{\log(6/\sigma_0 \|\boldsymbol{\mu}\|_2) 2^{q+1} m}{C_1 \eta q \sigma_0^{q-2} \|\boldsymbol{\mu}\|_2^q} \leq \frac{nm\eta^{-1} \sigma_0^{2-q} \sigma_p^{-q} d^{-q/2}}{2^{q+5} q [4 \log(8mn/\delta)]^{(q-1)/2}} = T_1^+/2,$$

where the inequality holds due to our SNR condition in (D.1). Therefore, by the definition of $T_{1,1}$, we have $T_{1,1} \leq T_1 \leq T_1^+/2$, where we use the non-decreasing property of $\gamma$. The proof for $j = -1$ is similar, and we can prove that $\max_r \gamma_{-1,r}^{(T_{1,-1})} \geq 2$ while $T_{1,-1} \leq T_1 \leq T_1^+/2$, which completes the proof. $\qquad\square$

## D.2 Second Stage

By the results we get in the first stage we know that

$$\mathbf{w}_{j,r}^{(T_1)} = \mathbf{w}_{j,r}^{(0)} + j \cdot \gamma_{j,r}^{(T_1)} \cdot \frac{\boldsymbol{\mu}}{\|\boldsymbol{\mu}\|_2^2} + \sum_{i=1}^n \overline{\rho}_{j,r,i}^{(T_1)} \cdot \frac{\boldsymbol{\xi}_i}{\|\boldsymbol{\xi}_i\|_2^2} + \sum_{i=1}^n \underline{\rho}_{j,r,i}^{(T_1)} \cdot \frac{\boldsymbol{\xi}_i}{\|\boldsymbol{\xi}_i\|_2^2}.$$

And at the beginning of the second stage, we have following property holds:

- $\max_r \gamma_{j,r}^{(T_1)} \geq 2, \forall j \in \{\pm 1\}$.
- $\max_{j,r,i} |\rho_{j,r,i}^{(T_1)}| \leq \widehat{\beta}$ where $\widehat{\beta} = \sigma_0 \sigma_p \sqrt{d}/2$.

Lemma 5.1 implies that the learned feature $\gamma_{j,r}^{(t)}$ will not get worse, i.e., for $t \geq T_1$, we have that $\gamma_{j,r}^{(t+1)} \geq \gamma_{j,r}^{(t)}$, and therefore $\max_r \gamma_{j,r}^{(t)} \geq 2$. Now we choose $\mathbf{W}^*$ as follows:

$$\mathbf{w}_{j,r}^* = \mathbf{w}_{j,r}^{(0)} + 2qm \log(2q/\epsilon) \cdot j \cdot \frac{\boldsymbol{\mu}}{\|\boldsymbol{\mu}\|_2^2}.$$

Based on the above definition of $\mathbf{W}^*$, we have the following lemma.

**Lemma D.2.** *Under the same conditions as Theorem 4.4, we have that* $\|\mathbf{W}^{(T_1)} - \mathbf{W}^*\|_F \leq \widetilde{O}(m^{3/2}\|\boldsymbol{\mu}\|_2^{-1})$.

*Proof of Lemma D.2.* We have

$$
\begin{aligned}
\|\mathbf{W}^{(T_1)} - \mathbf{W}^*\|_F &\leq \|\mathbf{W}^{(T_1)} - \mathbf{W}^{(0)}\|_F + \|\mathbf{W}^{(0)} - \mathbf{W}^*\|_F \\
&\leq \sum_{j,r} \frac{\gamma_{j,r}^{(T_1)}}{\|\boldsymbol{\mu}\|_2} + \sum_{j,r,i} \frac{|\overline{\rho}_{j,r,i}^{(T_1)}|}{\|\boldsymbol{\xi}_i\|_2} + \sum_{j,r,i} \frac{|\underline{\rho}_{j,r,i}^{(T_1)}|}{\|\boldsymbol{\xi}_i\|_2} + O(m^{3/2}\log(1/\epsilon))\|\boldsymbol{\mu}\|_2^{-1} \\
&\leq \widetilde{O}(m\|\boldsymbol{\mu}\|^{-1}) + O(nm\sigma_0) + O(m^{3/2}\log(1/\epsilon))\|\boldsymbol{\mu}\|_2^{-1} \\
&\leq \widetilde{O}(m^{3/2}\|\boldsymbol{\mu}\|_2^{-1}),
\end{aligned}
$$

where the first inequality is by triangle inequality, the second inequality is by our decomposition of $\mathbf{W}^{(T_1)}$ and the definition of $\mathbf{W}^*$, the third inequality is by Proposition C.2 and Lemma D.1, and the last inequality is by our condition of $\sigma_0$ in Condition 4.2. $\qquad\square$

**Lemma D.3.** *Under the same conditions as Theorem 4.3, we have that* $y_i \langle \nabla f(\mathbf{W}^{(t)}, \mathbf{x}_i), \mathbf{W}^* \rangle \geq q \log(2q/\epsilon)$ *for all* $i \in [n]$ *and* $T_1 \leq t \leq T^*$.

*Proof of Lemma D.3.* Recall that $f(\mathbf{W}^{(t)}, \mathbf{x}_i) = (1/m) \sum_{j,r} j \cdot [\sigma(\langle \mathbf{w}_{j,r}, y_i \cdot \boldsymbol{\mu} \rangle) + \sigma(\langle \mathbf{w}_{j,r}, \boldsymbol{\xi}_i \rangle)]$, so we have

$$
\begin{aligned}
y_i \langle \nabla f(\mathbf{W}^{(t)}, \mathbf{x}_i), \mathbf{W}^* \rangle &= \frac{1}{m} \sum_{j,r} \sigma'(\langle \mathbf{w}_{j,r}^{(t)}, y_i \boldsymbol{\mu} \rangle) \langle \boldsymbol{\mu}, j \mathbf{w}_{j,r}^* \rangle + \frac{1}{m} \sum_{j,r} \sigma'(\langle \mathbf{w}_{j,r}^{(t)}, \boldsymbol{\xi}_i \rangle) \langle y_i \boldsymbol{\xi}_i, j \mathbf{w}_{j,r}^* \rangle \\
&= \frac{1}{m} \sum_{j,r} \sigma'(\langle \mathbf{w}_{j,r}^{(t)}, y_i \boldsymbol{\mu} \rangle) 2qm \log(2q/\epsilon) + \frac{1}{m} \sum_{j,r} \sigma'(\langle \mathbf{w}_{j,r}^{(t)}, y_i \boldsymbol{\mu} \rangle) \langle \boldsymbol{\mu}, j \mathbf{w}_{j,r}^{(0)} \rangle \\
&\quad + \frac{1}{m} \sum_{j,r} \sigma'(\langle \mathbf{w}_{j,r}^{(t)}, \boldsymbol{\xi}_i \rangle) \langle y_i \boldsymbol{\xi}_i, j \mathbf{w}_{j,r}^{(0)} \rangle \\
&\geq \frac{1}{m} \sum_{j,r} \sigma'(\langle \mathbf{w}_{j,r}^{(t)}, y_i \boldsymbol{\mu} \rangle) 2qm \log(2q/\epsilon) - \frac{1}{m} \sum_{j,r} \sigma'(\langle \mathbf{w}_{j,r}^{(t)}, y_i \boldsymbol{\mu} \rangle) \widetilde{O}(\sigma_0 \|\boldsymbol{\mu}\|_2)
\end{aligned}
$$

$$-\frac{1}{m}\sum_{j,r}\sigma'(\langle\mathbf{w}_{j,r}^{(t)},\boldsymbol{\xi}_i\rangle)\widetilde{O}(\sigma_0\sigma_p\sqrt{d}), \tag{D.4}$$

where the inequality is by Lemma B.3. Next we will bound the inner-product terms in (D.4) respectively. By Lemma C.6, we have that for $j = y_i$

$$\max_r\{\langle\mathbf{w}_{j,r}^{(t)}, y_i\boldsymbol{\mu}\rangle\} = \max_r\{\gamma_{j,r}^{(t)} + \langle\mathbf{w}_{j,r}^{(0)}, y_i\boldsymbol{\mu}\rangle\} \geq 2 - \widetilde{O}(\sigma_0\|\boldsymbol{\mu}\|_2) \geq 1. \tag{D.5}$$

We can also get the upper bound of the inner products between the parameter and the signal (noise) as follows,

$$|\langle\mathbf{w}_{j,r}^{(t)},\boldsymbol{\mu}\rangle| \overset{(i)}{\leq} |\langle\mathbf{w}_{j,r}^{(0)},\boldsymbol{\mu}\rangle| + |\gamma_{j,r}^{(t)}| \overset{(ii)}{\leq} \widetilde{O}(1)$$

$$|\langle\mathbf{w}_{j,r}^{(t)},\boldsymbol{\xi}_i\rangle| \overset{(iii)}{\leq} |\langle\mathbf{w}_{j,r}^{(0)},\boldsymbol{\xi}_i\rangle| + |\underline{\rho}_{j,r,i}^{(t)}| + |\overline{\rho}_{j,r,i}^{(t)}| + 8n\sqrt{\frac{\log(4n^2/\delta)}{d}}\alpha \overset{(iv)}{\leq} \widetilde{O}(1), \tag{D.6}$$

where (i) is by Lemma C.3, (iii) is by Lemma C.4, (ii) and (iv) are due to Proposition C.2. Plugging (D.5) and (D.6) into (D.4) gives,

$$y_i\langle\nabla f(\mathbf{W}^{(t)},\mathbf{x}_i),\mathbf{W}^*\rangle \geq 2q\log(2q/\epsilon) - \widetilde{O}(\sigma_0\|\boldsymbol{\mu}\|_2) - \widetilde{O}(\sigma_0\sigma_p\sqrt{d}) \geq q\log(2q/\epsilon),$$

where the last inequality is by $\sigma_0 \leq \widetilde{O}(m^{-2/(q-2)}n^{-1}) \cdot \min\{(\sigma_p\sqrt{d})^{-1}, \|\boldsymbol{\mu}\|_2^{-1}\}$ in Condition 4.2. This completes the proof. $\square$

**Lemma D.4.** *Under the same conditions as Theorem 4.3, we have that*

$$\|\mathbf{W}^{(t)} - \mathbf{W}^*\|_F^2 - \|\mathbf{W}^{(t+1)} - \mathbf{W}^*\|_F^2 \geq (2q-1)\eta L_S(\mathbf{W}^{(t)}) - \eta\epsilon$$

*for all $T_1 \leq t \leq T^*$.*

*Proof of Lemma D.4.* We first apply a proof technique similar to Lemma 2.6 in Ji and Telgarsky (2020). The difference between our analysis and Ji and Telgarsky (2020) is that here the neural network is $q$ homogeneous rather than 1 homogeneous.

$$\|\mathbf{W}^{(t)} - \mathbf{W}^*\|_F^2 - \|\mathbf{W}^{(t+1)} - \mathbf{W}^*\|_F^2$$

$$= 2\eta\langle\nabla L_S(\mathbf{W}^{(t)}),\mathbf{W}^{(t)} - \mathbf{W}^*\rangle - \eta^2\|\nabla L_S(\mathbf{W}^{(t)})\|_F^2$$

$$= \frac{2\eta}{n}\sum_{i=1}^n\ell_i'^{(t)}[qy_if(\mathbf{W}^{(t)},\mathbf{x}_i) - \langle\nabla f(\mathbf{W}^{(t)},\mathbf{x}_i),\mathbf{W}^*\rangle] - \eta^2\|\nabla L_S(\mathbf{W}^{(t)})\|_F^2$$

$$\geq \frac{2\eta}{n}\sum_{i=1}^n\ell_i'^{(t)}[qy_if(\mathbf{W}^{(t)},\mathbf{x}_i) - q\log(2q/\epsilon)] - \eta^2\|\nabla L_S(\mathbf{W}^{(t)})\|_F^2$$

$$\geq \frac{2q\eta}{n}\sum_{i=1}^n[\ell\big(y_if(\mathbf{W}^{(t)},\mathbf{x}_i)\big) - \epsilon/(2q)] - \eta^2\|\nabla L_S(\mathbf{W}^{(t)})\|_F^2$$

$$\geq (2q-1)\eta L_S(\mathbf{W}^{(t)}) - \eta\epsilon,$$

where the first inequality is by Lemma D.3, the second inequality is due to the convexity of the cross entropy function, and the last inequality is due to Lemma C.7. $\square$

**Lemma D.5** (Restatement of Lemma 5.6). *Under the same conditions as Theorem 4.3, let $T = T_1 + \left\lfloor\frac{\|\mathbf{W}^{(T_1)} - \mathbf{W}^*\|_F^2}{2\eta\epsilon}\right\rfloor = T_1 + \widetilde{O}(m^3\eta^{-1}\epsilon^{-1}\|\boldsymbol{\mu}\|_2^{-2})$. Then we have $\max_{j,r,i}|\rho_{j,r,i}^{(t)}| \leq 2\widehat{\beta} = \sigma_0\sigma_p\sqrt{d}$ for all $T_1 \leq t \leq T$. Besides,*

$$\frac{1}{t - T_1 + 1}\sum_{s=T_1}^t L_S(\mathbf{W}^{(s)}) \leq \frac{\|\mathbf{W}^{(T_1)} - \mathbf{W}^*\|_F^2}{(2q-1)\eta(t - T_1 + 1)} + \frac{\epsilon}{2q-1}$$

*for all $T_1 \leq t \leq T$, and we can find an iterate with training loss smaller than $\epsilon$ within $T$ iterations.*

*Proof of Lemma D.5.* By Lemma D.4, for any $t \in [T_1, T]$, we have that

$$\|\mathbf{W}^{(s)} - \mathbf{W}^*\|_F^2 - \|\mathbf{W}^{(s+1)} - \mathbf{W}^*\|_F^2 \geq (2q-1)\eta L_S(\mathbf{W}^{(s)}) - \eta\epsilon$$

holds for $s \leq t$. Taking a summation, we obtain that

$$\sum_{s=T_1}^{t} L_S(\mathbf{W}^{(s)}) \leq \frac{\|\mathbf{W}^{(T_1)} - \mathbf{W}^*\|_F^2 + \eta\epsilon(t - T_1 + 1)}{(2q-1)\eta} \tag{D.7}$$

for all $T_1 \leq t \leq T$. Dividing $(t - T_1 + 1)$ on both side of (D.7) gives that

$$\frac{1}{t - T_1 + 1} \sum_{s=T_1}^{t} L_S(\mathbf{W}^{(s)}) \leq \frac{\|\mathbf{W}^{(T_1)} - \mathbf{W}^*\|_F^2}{(2q-1)\eta(t - T_1 + 1)} + \frac{\epsilon}{2q-1}.$$

Then we can take $t = T$ and have that

$$\frac{1}{T - T_1 + 1} \sum_{s=T_1}^{T} L_S(\mathbf{W}^{(s)}) \leq \frac{\|\mathbf{W}^{(T_1)} - \mathbf{W}^*\|_F^2}{(2q-1)\eta(T - T_1 + 1)} + \frac{\epsilon}{2q-1} \leq \frac{3\epsilon}{2q-1} < \epsilon,$$

where we use the fact that $q > 2$ and our choice that $T = T_1 + \left\lfloor \frac{\|\mathbf{W}^{(T_1)} - \mathbf{W}^*\|_F^2}{2\eta\epsilon} \right\rfloor$. Because the mean is smaller than $\epsilon$, we can conclude that there exist $T_1 \leq t \leq T$ such that $L_S(\mathbf{W}^{(t)}) < \epsilon$.

Finally, we will prove that $\max_{j,r,i} |\rho_{j,r,i}^{(t)}| \leq 2\widehat{\beta}$ for all $t \in [T_1, T]$. Plugging $T = T_1 + \left\lfloor \frac{\|\mathbf{W}^{(T_1)} - \mathbf{W}^*\|_F^2}{2\eta\epsilon} \right\rfloor$ into (D.7) gives that

$$\sum_{s=T_1}^{T} L_S(\mathbf{W}^{(s)}) \leq \frac{2\|\mathbf{W}^{(T_1)} - \mathbf{W}^*\|_F^2}{(2q-1)\eta} = \widetilde{O}(\eta^{-1}m^3\|\boldsymbol{\mu}\|_2^2), \tag{D.8}$$

where the inequality is due to $\|\mathbf{W}^{(T_1)} - \mathbf{W}^*\|_F \leq \widetilde{O}(m^{3/2}\|\boldsymbol{\mu}\|_2^{-1})$ in Lemma D.2. Define $\Psi^{(t)} = \max_{j,r,i} |\rho_{j,r,i}^{(t)}|$. We will use induction to prove $\Psi^{(t)} \leq 2\widehat{\beta}$ for all $t \in [T_1, T]$. At $t = T_1$, by the definition of $\widehat{\beta}$, clearly we have $\Psi^{(T_1)} \leq \widehat{\beta} \leq 2\widehat{\beta}$. Now suppose that there exists $\widetilde{T} \in [T_1, T]$ such that $\Psi^{(t)} \leq 2\widehat{\beta}$ for all $t \in [T_1, \widetilde{T} - 1]$. Then for $t \in [T_1, \widetilde{T} - 1]$, by (C.4) and (C.5) we have

$$\Psi^{(t+1)} \leq \Psi^{(t)} + \max_{j,r,i} \left\{ \frac{\eta}{nm} \cdot |\ell_i'^{(t)}| \cdot \sigma'\left(\langle \mathbf{w}_{j,r}^{(0)}, \boldsymbol{\xi}_i \rangle + 2\sum_{i'=1}^{n} \Psi^{(t)} \cdot \frac{|\langle \boldsymbol{\xi}_{i'}, \boldsymbol{\xi}_i \rangle|}{\|\boldsymbol{\xi}_{i'}\|_2^2} \right) \cdot \|\boldsymbol{\xi}_{i'}\|_2^2 \right\}$$

$$= \Psi^{(t)} + \max_{j,r,i} \left\{ \frac{\eta}{nm} \cdot |\ell_i'^{(t)}| \cdot \sigma'\left(\langle \mathbf{w}_{j,r}^{(0)}, \boldsymbol{\xi}_i \rangle + 2\Psi^{(t)} + 2\sum_{i'\neq i}^{n} \Psi^{(t)} \cdot \frac{|\langle \boldsymbol{\xi}_{i'}, \boldsymbol{\xi}_i \rangle|}{\|\boldsymbol{\xi}_{i'}\|_2^2} \right) \cdot \|\boldsymbol{\xi}_{i'}\|_2^2 \right\}$$

$$\leq \Psi^{(t)} + \frac{\eta q}{nm} \cdot \max_i |\ell_i'^{(t)}| \cdot \left[ 2 \cdot \sqrt{\log(8mn/\delta)} \cdot \sigma_0 \sigma_p \sqrt{d} \right.$$

$$\left. + \left( 2 + \frac{4n\sigma_p^2 \cdot \sqrt{d\log(4n^2/\delta)}}{\sigma_p^2 d/2} \right) \cdot \Psi^{(t)} \right]^{q-1} \cdot 2\sigma_p^2 d$$

$$\leq \Psi^{(t)} + \frac{\eta q}{nm} \cdot \max_i |\ell_i'^{(t)}| \cdot \left( 2 \cdot \sqrt{\log(8mn/\delta)} \cdot \sigma_0 \sigma_p \sqrt{d} + 4 \cdot \Psi^{(t)} \right)^{q-1} \cdot 2\sigma_p^2 d,$$

where the second inequality is due to Lemmas B.2 and B.3, and the last inequality follows by the assumption that $d \geq 16n^2 \log(4n^2/\delta)$. Taking a telescoping sum over $t = 0, 1, \ldots, \widetilde{T} - 1$, we have that

$$\Psi^{(T)} \overset{(i)}{\leq} \Psi^{(T_1)} + \frac{\eta q}{nm} \sum_{s=T_1}^{\widetilde{T}-1} \max_i |\ell_i'^{(s)}| \widetilde{O}(\sigma_p^2 d)\widehat{\beta}^{q-1}$$

$$\overset{(ii)}{\leq} \Psi^{(T_1)} + \frac{\eta q}{nm} \widetilde{O}(\sigma_p^2 d)\widehat{\beta}^{q-1} \sum_{s=T_1}^{\widetilde{T}-1} \max_i \ell_i^{(s)}$$

$$\overset{(iii)}{\leq} \Psi^{(T_1)} + \widetilde{O}(\eta m^{-1}\sigma_p^2 d)\widehat{\beta}^{q-1} \sum_{s=T_1}^{\widetilde{T}-1} L_S(\mathbf{W}^{(s)})$$

$$\overset{(iv)}{\leq} \Psi^{(T_1)} + \widetilde{O}(m^2 \text{SNR}^{-2})\widehat{\beta}^{q-1}$$

$$\overset{(v)}{\leq} \widehat{\beta} + \widetilde{O}(m^2 n^{2/q}\widehat{\beta}^{q-2})\widehat{\beta}$$

$$\overset{(vi)}{\leq} 2\widehat{\beta},$$

where (i) is by out induction hypothesis that $\Psi^{(t)} \leq 2\widehat{\beta}$, (ii) is by $|\ell'| \leq \ell$, (iii) is by $\max_i \ell_i^{(s)} \leq \sum_i \ell_i^{(s)} = nL_S(\mathbf{W}^{(s)})$, (iv) is due to $\sum_{s=T_1}^{\widetilde{T}-1} L_S(\mathbf{W}^{(s)}) \leq \sum_{s=T_1}^{T} L_S(\mathbf{W}^{(s)}) = \widetilde{O}(\eta^{-1}m^3\|\boldsymbol{\mu}\|_2^2)$ in (D.8), (v) is by $n\text{SNR}^q \geq \widetilde{\Omega}(1)$, and (vi) is by the definition that $\widehat{\beta} = \sigma_0 \sigma_p \sqrt{d}/2$ and $\widetilde{O}(m^2 n^{2/q}\widehat{\beta}^{q-2}) = \widetilde{O}(m^2 n^{2/q}(\sigma_0 \sigma_p \sqrt{d})^{q-2}) \leq 1$ by Condition 4.2. Therefore, $\Psi^{(\widetilde{T})} \leq 2\widehat{\beta}$, which completes the induction. $\qquad\square$

### D.3 Population Loss

Consider a new data point $(\mathbf{x}, y)$ drawn from the distribution defined in Definition 3.1. Without loss of generality, we suppose that the first patch is the signal patch and the second patch is the noise patch, i.e., $\mathbf{x} = [y\boldsymbol{\mu}, \boldsymbol{\xi}]$. Moreover, by the signal-noise decomposition, the learned neural network has parameter

$$\mathbf{w}_{j,r}^{(t)} = \mathbf{w}_{j,r}^{(0)} + j \cdot \gamma_{j,r}^{(t)} \cdot \frac{\boldsymbol{\mu}}{\|\boldsymbol{\mu}\|_2^2} + \sum_{i=1}^{n} \overline{\rho}_{j,r,i}^{(t)} \cdot \frac{\boldsymbol{\xi}_i}{\|\boldsymbol{\xi}_i\|_2^2} + \sum_{i=1}^{n} \underline{\rho}_{j,r,i}^{(t)} \cdot \frac{\boldsymbol{\xi}_i}{\|\boldsymbol{\xi}_i\|_2^2}$$

for $j \in \{\pm 1\}$ and $r \in [m]$.

**Lemma D.6.** *Under the same conditions as Theorem 4.3, we have that $\max_{j,r} |\langle \mathbf{w}_{j,r}^{(t)}, \boldsymbol{\xi}_i\rangle| \leq 1/2$ for all $0 \leq t \leq T$.*

*Proof.* We can get the upper bound of the inner products between the parameter and the noise as follows:

$$|\langle \mathbf{w}_{j,r}^{(t)}, \boldsymbol{\xi}_i\rangle| \overset{(i)}{\leq} |\langle \mathbf{w}_{j,r}^{(0)}, \boldsymbol{\xi}_i\rangle| + |\underline{\rho}_{j,r,i}^{(t)}| + |\overline{\rho}_{j,r,i}^{(t)}| + 8n\sqrt{\frac{\log(4n^2/\delta)}{d}}\alpha$$

$$\overset{(ii)}{\leq} 2\sqrt{\log(8mn/\delta)} \cdot \sigma_0 \sigma_p \sqrt{d} + \sigma_0 \sigma_p \sqrt{d} + 8n\sqrt{\frac{\log(4n^2/\delta)}{d}}\alpha$$

$$\overset{(iii)}{\leq} 1/2$$

for all $j \in \{\pm 1\}$, $r \in [m]$ and $i \in [n]$, where (i) is by Lemma C.3, (ii) is due to $|\langle \mathbf{w}_{j,r}^{(0)}, \boldsymbol{\xi}_i\rangle| \leq 2\sqrt{\log(8mn/\delta)} \cdot \sigma_0 \sigma_p \sqrt{d}$ in Lemma B.3 and $\max_{j,r,i} |\rho_{j,r,i}^{(t)}| \leq \sigma_0 \sigma_p \sqrt{d}$ in Lemma D.5, and (iii) is due to our condition of $\sigma_0 \leq \widetilde{O}(m^{-2/(q-2)}n^{-1}) \cdot (\sigma_p \sqrt{d})^{-1}$ and $d \geq \widetilde{\Omega}(m^2 n^4)$ in Condition 4.2. $\qquad\square$

**Lemma D.7.** *Under the same conditions as Theorem 4.3, with probability at least $1 - 4mT \cdot \exp(-C_2^{-1}\sigma_0^{-2}\sigma_p^{-2}d^{-1})$, we have that $\max_{j,r} |\langle \mathbf{w}_{j,r}^{(t)}, \boldsymbol{\xi}\rangle| \leq 1/2$ for all $0 \leq t \leq T$, where $C_2 = \widetilde{O}(1)$.*

*Proof of Lemma D.7.* Let $\widetilde{\mathbf{w}}_{j,r}^{(t)} = \mathbf{w}_{j,r}^{(t)} - j \cdot \gamma_{j,r}^{(t)} \cdot \frac{\boldsymbol{\mu}}{\|\boldsymbol{\mu}\|_2^2}$, then we have that $\langle \widetilde{\mathbf{w}}_{j,r}^{(t)}, \boldsymbol{\xi}\rangle = \langle \mathbf{w}_{j,r}^{(t)}, \boldsymbol{\xi}\rangle$ and

$$\|\widetilde{\mathbf{w}}_{j,r}^{(t)}\|_2 \leq \widetilde{O}(\sigma_0 \sqrt{d} + n\sigma_0) = \widetilde{O}(\sigma_0 \sqrt{d}), \tag{D.9}$$

where the equality is due to $d \geq \widetilde{\Omega}(m^2 n^4)$ by Condition 4.2.

By (D.9), $\max_{j,r} \|\widetilde{\mathbf{w}}_{j,r}^{(t)}\|_2 \leq C_1 \sigma_0 \sqrt{d}$, where $C_1 = \widetilde{O}(1)$. Clearly $\langle \widetilde{\mathbf{w}}_{j,r}^{(t)}, \boldsymbol{\xi} \rangle$ is a Gaussian distribution with mean zero and standard deviation smaller than $C_1 \sigma_0 \sigma_p \sqrt{d}$. Therefore, the probability is bounded by

$$\mathbb{P}\big(|\langle \widetilde{\mathbf{w}}_{j,r}^{(t)}, \boldsymbol{\xi} \rangle| \geq 1/2\big) \leq 2 \exp\bigg( - \frac{1}{8 C_1^2 \sigma_0^2 \sigma_p^2 d} \bigg).$$

Applying a union bound over $j, r, t$ completes the proof. $\qquad\square$

**Lemma D.8** (Restatement of Lemma 5.7). *Let $T$ be defined in Lemma 5.5 respectively. Under the same conditions as Theorem 4.3, for any $0 \leq t \leq T$ with $L_S(\mathbf{W}^{(t)}) \leq 1$, it holds that $L_\mathcal{D}(\mathbf{W}^{(t)}) \leq 6 \cdot L_S(\mathbf{W}^{(t)}) + \exp(-n^2)$.*

*Proof of Lemma D.8.* Let event $\mathcal{E}$ be the event that Lemma D.7 holds. Then we can divide $L_\mathcal{D}(\mathbf{W}^{(t)})$ into two parts:

$$\mathbb{E}\big[\ell\big(yf(\mathbf{W}^{(t)}, \mathbf{x})\big)\big] = \underbrace{\mathbb{E}[\mathbb{1}(\mathcal{E})\ell\big(yf(\mathbf{W}^{(t)}, \mathbf{x})\big)]}_{I_1} + \underbrace{\mathbb{E}[\mathbb{1}(\mathcal{E}^c)\ell\big(yf(\mathbf{W}^{(t)}, \mathbf{x})\big)]}_{I_2}. \qquad (D.10)$$

In the following, we bound $I_1$ and $I_2$ respectively.

**Bounding $I_1$:** Since $L_S(\mathbf{W}^{(t)}) \leq 1$, there must exist one $(\mathbf{x}_i, y_i)$ such that $\ell\big(y_i f(\mathbf{W}^{(t)}, \mathbf{x}_i)\big) \leq L_S(\mathbf{W}^{(t)}) \leq 1$, which implies that $y_i f(\mathbf{W}^{(t)}, \mathbf{x}_i) \geq 0$. Therefore, we have that

$$\exp(-y_i f(\mathbf{W}^{(t)}, \mathbf{x}_i)) \overset{(i)}{\leq} 2 \log\big(1 + \exp(-y_i f(\mathbf{W}^{(t)}, \mathbf{x}_i))\big) = 2\ell\big(y_i f(\mathbf{W}^{(t)}, \mathbf{x}_i)\big) \leq 2 L_S(\mathbf{W}^{(t)}), \tag{D.11}$$

where (i) is by $z \leq 2\log(1 + z), \forall z \leq 1$. If event $\mathcal{E}$ holds, we have that

$$\begin{aligned}
|yf(\mathbf{W}^{(t)}, \mathbf{x}) - y_i f(\mathbf{W}^{(t)}, \mathbf{x}_i)| &\leq \frac{1}{m} \sum_{j,r} \sigma(\langle \mathbf{w}_{j,r}, \boldsymbol{\xi}_i \rangle) + \frac{1}{m} \sum_{j,r} \sigma(\langle \mathbf{w}_{j,r}, \boldsymbol{\xi} \rangle) \\
&\leq \frac{1}{m} \sum_{j,r} \sigma(1/2) + \frac{1}{m} \sum_{j,r} \sigma(1/2) \\
&\leq 1, \qquad\qquad\qquad\qquad\qquad\qquad\qquad\qquad (D.12)
\end{aligned}$$

where the second inequality is by $\max_{j,r} |\langle \mathbf{w}_{j,r}^{(t)}, \boldsymbol{\xi} \rangle| \leq 1/2$ in Lemma D.7 and $\max_{j,r} |\langle \mathbf{w}_{j,r}^{(t)}, \boldsymbol{\xi}_i \rangle| \leq 1/2$ in Lemma D.6. Thus we have that

$$\begin{aligned}
I_1 &\leq \mathbb{E}[\mathbb{1}(\mathcal{E}) \exp(-yf(\mathbf{W}^{(t)}, \mathbf{x}))] \\
&\leq e \cdot \mathbb{E}[\mathbb{1}(\mathcal{E}) \exp(-y_i f(\mathbf{W}^{(t)}, \mathbf{x}_i))] \\
&\leq 2e \cdot \mathbb{E}[\mathbb{1}(\mathcal{E}) L_S(\mathbf{W}^{(t)})],
\end{aligned}$$

where the first inequality is by the property of cross-entropy loss that $\ell(z) \leq \exp(-z)$ for all $z$, the second inequality is by (D.12), and the third inequality is by (D.11). Dropping the event in the expectation gives $I_1 \leq 6 L_S(\mathbf{W}^{(t)})$.

**Bounding $I_2$:** Next we bound the second term $I_2$. We choose an arbitrary training data $(\mathbf{x}_{i'}, y_{i'})$ such that $y_{i'} = y$. Then we have

$$\begin{aligned}
\ell\big(yf(\mathbf{W}^{(t)}, \mathbf{x})\big) &\leq \log(1 + \exp(F_{-y}(\mathbf{W}^{(t)}, \mathbf{x}))) \\
&\leq 1 + F_{-y}(\mathbf{W}^{(t)}, \mathbf{x}) \\
&= 1 + \frac{1}{m} \sum_{j=-y, r \in [m]} \sigma(\langle \mathbf{w}_{j,r}^{(t)}, y\boldsymbol{\mu} \rangle) + \frac{1}{m} \sum_{j=-y, r \in [m]} \sigma(\langle \mathbf{w}_{j,r}^{(t)}, \boldsymbol{\xi} \rangle) \\
&\leq 1 + F_{-y_{i'}}(\mathbf{W}_{-y_{i'}}, \mathbf{x}_{i'}) + \frac{1}{m} \sum_{j=-y, r \in [m]} \sigma(\langle \mathbf{w}_{j,r}^{(t)}, \boldsymbol{\xi} \rangle)
\end{aligned}$$

$$\leq 2 + \frac{1}{m} \sum_{j=-y, r \in [m]} \sigma(\langle \mathbf{w}_{j,r}^{(t)}, \boldsymbol{\xi} \rangle)$$

$$\leq 2 + \widetilde{O}((\sigma_0 \sqrt{d})^q) \|\boldsymbol{\xi}\|^q, \tag{D.13}$$

where the first inequality is due to $F_y(\mathbf{W}^{(t)}, \mathbf{x}) \geq 0$, the second inequality is by the property of cross-entropy loss, i.e., $\log(1 + \exp(z)) \leq 1 + z$ for all $z \geq 0$, the third inequality is by $\frac{1}{m} \sum_{j=-y, r \in [m]} \sigma(\langle \mathbf{w}_{j,r}^{(t)}, y\boldsymbol{\mu} \rangle) \leq F_{-y}(\mathbf{W}_{-y}, \mathbf{x}_{i'}) = F_{-y_{i'}}(\mathbf{W}_{-y_{i'}}, \mathbf{x}_{i'})$, the fourth inequality is by $F_{-y_{i'}}(\mathbf{W}_{-y_{i'}}, \mathbf{x}_{i'}) \leq 1$ in Lemma C.5, and the last inequality is due to $\langle \widetilde{\mathbf{w}}_{j,r}^{(t)}, \boldsymbol{\xi} \rangle = \langle \mathbf{w}_{j,r}^{(t)}, \boldsymbol{\xi} \rangle \leq \|\widetilde{\mathbf{w}}_{j,r}^{(t)}\|_2 \|\boldsymbol{\xi}\|_2 \leq \widetilde{O}(\sigma_0 \sqrt{d}) \|\boldsymbol{\xi}\|_2$ in (D.9). Then we further have that

$$I_2 \leq \sqrt{\mathbb{E}[\mathbb{1}(\mathcal{E}^c)]} \cdot \sqrt{\mathbb{E}\left[ \ell \big( y f(\mathbf{W}^{(t)}, \mathbf{x}) \big)^2 \right]}$$

$$\leq \sqrt{\mathbb{P}(\mathcal{E}^c)} \cdot \sqrt{4 + \widetilde{O}((\sigma_0 \sqrt{d})^{2q}) \mathbb{E}[\|\boldsymbol{\xi}\|_2^{2q}]}$$

$$\leq \exp[-\widetilde{\Omega}(\sigma_0^{-2} \sigma_p^{-2} d^{-1}) + \text{polylog}(d)]$$

$$\leq \exp(-n^2),$$

where the first inequality is by Cauchy-Schwartz inequality, the second inequality is by (D.13), the third inequality is by Lemma D.7 and the fact that $\sqrt{4 + \widetilde{O}((\sigma_0 \sqrt{d})^{2q}) \mathbb{E}[\|\boldsymbol{\xi}\|_2^{2q}]} = O(\text{poly}(d))$, and the last inequality is by our condition $\sigma_0 \leq \widetilde{O}(m^{-2/(q-2)} n^{-1}) \cdot (\sigma_p \sqrt{d})^{-1}$ in Condition 4.2. Plugging the bounds of $I_1, I_2$ into (D.10) completes the proof. $\qquad \square$

# E  Noise Memorization

In this section, we will consider the noise memorization case under the condition that $\sigma_p^q (\sqrt{d})^q \geq \widetilde{\Omega}(n \|\boldsymbol{\mu}\|_2^q)$. We remind the readers that the proofs in this section are based on the results in Section B, which hold with high probability.

We also remind readers that $\alpha = 4 \log(T^*)$ is defined in Appendix C. Denote $\bar{\beta} = \min_i \max_r \langle \mathbf{w}_{y_i,r}^{(0)}, \boldsymbol{\xi}_i \rangle$. The following lemma provides a lower bound of $\bar{\beta}$.

**Lemma E.1.** *Under the same conditions as Theorem 4.4, if in particular*

$$\sigma_0 \geq 80n \sqrt{\frac{\log(4n^2/\delta)}{d}} \alpha \cdot \min\{(\sigma_p \sqrt{d})^{-1}, \|\boldsymbol{\mu}\|_2^{-1}\}, \tag{E.1}$$

*then we have that $\bar{\beta} \geq \sigma_0 \sigma_p \sqrt{d}/4 \geq 20n \sqrt{\frac{\log(4n^2/\delta)}{d}} \alpha$.*

*Proof of Lemma E.1.* Because $\sigma_p^q (\sqrt{d})^q \geq \widetilde{\Omega}(n \|\boldsymbol{\mu}\|_2^q)$, we have that $\sigma_p \sqrt{d} \geq \|\boldsymbol{\mu}\|_2$. Therefore we have that

$$\bar{\beta} \geq \sigma_0 \sigma_p \sqrt{d}/4$$

$$= \sigma_0/4 \cdot \max\{\sigma_p \sqrt{d}, \|\boldsymbol{\mu}\|_2\}$$

$$\geq 20n \sqrt{\frac{\log(4n^2/\delta)}{d}} \alpha,$$

where the first inequality is by Lemma B.3 and the last inequality is by our lower bound condition of $\sigma_0$ in (E.1). $\qquad \square$

## E.1  First Stage

**Lemma E.2.** *Under the same conditions as Theorem 4.4, in particular if we choose*

$$n^{-1} \text{SNR}^{-q} \geq \frac{C 2^{q+2} \log\big(20/(\sigma_0 \sigma_p \sqrt{d})\big) \big(\sqrt{2 \log(8m/\delta)}\big)^{q-2}}{0.15^{q-2}}, \tag{E.2}$$

*where $C = O(1)$ is a positive constant, then there exist*

$$T_1 = \frac{C \log\left(20/(\sigma_0 \sigma_p \sqrt{d})\right) 4mn}{0.15^{q-2} \eta q \sigma_0^{q-2} (\sigma_p \sqrt{d})^q}$$

*such that*

- $\max_{j,r} \overline{\rho}_{j,r,i}^{(T_1)} \geq 2$ *for all $i \in [n]$.*
- $\max_{j,r} \gamma_{j,r}^{(t)} = \widetilde{O}(\sigma_0 \|\boldsymbol{\mu}\|_2)$ *for all $0 \leq t \leq T_1$.*
- $\max_{j,r,i} |\underline{\rho}_{j,r,i}^{(t)}| = \widetilde{O}(\sigma_0 \sigma_p \sqrt{d})$ *for all $0 \leq t \leq T_1$.*

*Proof of Lemma E.2.* Let

$$T_1^+ = \frac{m}{\eta q 2^{q-1} (\sqrt{2\log(8m/\delta)})^{q-2} \sigma_0^{q-2} \|\boldsymbol{\mu}\|_2^q}. \tag{E.3}$$

By Proposition C.2, we have that $\underline{\rho}_{j,r,i}^{(t)} \geq -\beta - 16n\sqrt{\frac{\log(4n^2/\delta)}{d}}\alpha \geq -\beta - \bar{\beta}$ for all $j \in \{\pm 1\}$, $r \in [m]$, $i \in [n]$ and $0 \leq t \leq T^*$. Since $\underline{\rho}_{j,r,i}^{(t)} \leq 0$ and $\bar{\beta} \leq \beta = \widetilde{O}(\sigma_0 \sigma_p \sqrt{d})$, we have that $\max_{j,r,i} |\underline{\rho}_{j,r,i}^{(t)}| = \widetilde{O}(\sigma_0 \sigma_p \sqrt{d})$. Next, we will carefully compute the growth of the $\gamma_{j,r}^{(t)}$.

$$\gamma_{j,r}^{(t+1)} = \gamma_{j,r}^{(t)} - \frac{\eta}{nm} \cdot \sum_{i=1}^{n} \ell_i'^{(t)} \cdot \sigma'(\langle \mathbf{w}_{j,r}^{(t)}, y_i \cdot \boldsymbol{\mu} \rangle) \|\boldsymbol{\mu}\|_2^2$$

$$\leq \gamma_{j,r}^{(t)} + \frac{\eta}{nm} \cdot \sum_{i=1}^{n} \sigma'(|\langle \mathbf{w}_{j,r}^{(0)}, \boldsymbol{\mu} \rangle| + \gamma_{j,r}^{(t)}) \|\boldsymbol{\mu}\|_2^2,$$

where the inequality is by $|\ell'| \leq 1$. Let $A^{(t)} = \max_{j,r}\{\gamma_{j,r}^{(t)} + |\langle \mathbf{w}_{j,r}^{(0)}, \boldsymbol{\mu} \rangle|\}$, then we have that

$$A^{(t+1)} \leq A^{(t)} + \frac{\eta q \|\boldsymbol{\mu}\|_2^2}{m} [A^{(t)}]^{q-1}. \tag{E.4}$$

We will use induction to prove that $A^{(t)} \leq 2A^{(0)}$ for $t \leq T_1^+$. By definition, clearly we have that $A^{(0)} \leq 2A^{(0)}$. Now suppose that there exists some $\widetilde{T} \leq T_1^+$ such that $A^{(t)} \leq 2A^{(0)}$ holds for $0 \leq t \leq \widetilde{T} - 1$. Taking a telescoping sum of (E.4) gives that

$$A^{(\widetilde{T})} \leq A^{(0)} + \sum_{s=0}^{\widetilde{T}} \frac{\eta q \|\boldsymbol{\mu}\|_2^2}{m} [A^{(s)}]^{q-1}$$

$$\leq A^{(0)} + \frac{\eta q \|\boldsymbol{\mu}\|_2^2 T_1^+ 2^{q-1}}{m} [A^{(0)}]^{q-1}$$

$$\leq A^{(0)} + \frac{\eta q \|\boldsymbol{\mu}\|_2^2 T_1^+ 2^{q-1}}{m} [\sqrt{2\log(8m/\delta)} \cdot \sigma_0 \|\boldsymbol{\mu}\|_2]^{q-2} A^{(0)}$$

$$\leq 2A^{(0)},$$

where the second inequality is by our induction hypothesis, the third inequality is by $A_0 \leq \sqrt{2\log(8m/\delta)} \cdot \sigma_0 \|\boldsymbol{\mu}\|_2$ in Lemma B.3, and the last inequality is by (E.3). Thus we have that $A^{(t)} \leq 2A^{(0)}$ for all $t \leq T_1^+$. Therefore, $\max_{j,r} \gamma_{j,r}^{(t)} \leq A^{(t)} + \max_{j,r}\{|\langle \mathbf{w}_{j,r}^{(0)}, \boldsymbol{\mu} \rangle|\} \leq 3A^{(0)}$ for all $0 \leq t \leq T_1^+$. Recall that

$$\overline{\rho}_{j,r,i}^{(t+1)} = \overline{\rho}_{j,r,i}^{(t)} - \frac{\eta}{nm} \cdot \ell_i'^{(t)} \cdot \sigma'(\langle \mathbf{w}_{j,r}^{(t)}, \boldsymbol{\xi}_i \rangle) \cdot \mathbb{1}(y_i = j) \|\boldsymbol{\xi}_i\|_2^2.$$

For $y_i = j$, Lemma C.4 implies that

$$\langle \mathbf{w}_{j,r}^{(t)}, \boldsymbol{\xi}_i \rangle \geq \langle \mathbf{w}_{j,r}^{(0)}, \boldsymbol{\xi}_i \rangle + \overline{\rho}_{j,r,i}^{(t)} - 8n\sqrt{\frac{\log(4n^2/\delta)}{d}}\alpha$$

$$\geq \overline{\rho}_{j,r,i}^{(t)} + \langle \mathbf{w}_{j,r}^{(0)}, \boldsymbol{\xi}_i \rangle - 0.4\overline{\beta},$$

where the last inequality is by $\overline{\beta} \geq 20n\sqrt{\frac{\log(4n^2/\delta)}{d}}\alpha$. Now let $B_i^{(t)} = \max_{j=y_i,r}\{\overline{\rho}_{j,r,i}^{(t)} + \langle \mathbf{w}_{j,r}^{(0)}, \boldsymbol{\xi}_i \rangle - 0.4\overline{\beta}\}$. For each $i$, denote by $T_1^{(i)}$ the last time in the period $[0, T_1^+]$ satisfying that $\overline{\rho}_{j,r,i}^{(t)} \leq 2$. Then for $t \leq T_1^{(i)}$, $\max_{j,r}\{|\overline{\rho}_{j,r,i}^{(t)}|, |\underline{\rho}_{j,r,i}^{(t)}|\} = O(1)$ and $\max_{j,r}\gamma_{j,r}^{(t)} \leq 3A^{(0)} = O(1)$. Therefore, by Lemmas C.5 and C.6, we know that $F_{-1}(\mathbf{W}^{(t)}, \mathbf{x}_i), F_{+1}(\mathbf{W}^{(t)}, \mathbf{x}_i) = O(1)$. Thus there exists a positive constant $C_1$ such that $-\ell_i'^{(t)} \geq C_1$ for all $0 \leq t \leq T_1^{(i)}$. It is also easy to check that $B_i^{(0)} \geq 0.6\overline{\beta} \geq 0.15\sigma_0\sigma_p\sqrt{d}$. Then we can carefully compute the growth of $B_i^{(t)}$,

$$B_i^{(t+1)} \geq B_i^{(t)} + \frac{C_1\eta q\sigma_p^2 d}{2nm}[B_i^{(t)}]^{q-1}$$

$$\geq B_i^{(t)} + \frac{C_1\eta q\sigma_p^2 d}{2nm}[B_i^{(0)}]^{q-2}B_i^{(t)}$$

$$\geq \left(1 + \frac{C_1 0.15^{q-2}\eta q\sigma_0^{q-2}(\sigma_p^2\sqrt{d})^q}{2nm}\right)B_i^{(t)},$$

where the second inequality is by the non-decreasing property of $B_i^{(t)}$. Therefore, $B_i^{(t)}$ is an exponentially increasing sequence and we have that

$$B_i^{(t)} \geq \left(1 + \frac{C_1 0.15^{q-2}\eta q\sigma_0^{q-2}(\sigma_p^2\sqrt{d})^q}{2nm}\right)^t B_i^{(0)}$$

$$\geq \exp\left(\frac{C_1 0.15^{q-2}\eta q\sigma_0^{q-2}(\sigma_p^2\sqrt{d})^q}{4nm}t\right)B_i^{(0)}$$

$$\geq \exp\left(\frac{C_1 0.15^{q-2}\eta q\sigma_0^{q-2}(\sigma_p^2\sqrt{d})^q}{4nm}t\right) \cdot 0.15\sigma_0\sigma_p\sqrt{d},$$

where the second inequality is due to the fact that $1 + z \geq \exp(z/2)$ for $z \leq 2$ and our conditions of $\eta$ and $\sigma_0$ listed in Condition 4.2, and the last inequality is due to $B_i^{(0)} \geq 0.15\sigma_0\sigma_p\sqrt{d}$. Therefore, $B_i^{(t)}$ will reach 3 within $T_1 = \frac{\log\left(20/(\sigma_0\sigma_p\sqrt{d})\right)4nm}{C_1 0.15^{q-2}\eta q\sigma_0^{q-2}(\sigma_p^2\sqrt{d})^q}$ iterations. Since $\max_{j=y_i,r}\overline{\rho}_{j,r,i}^{(t)} \geq B_i^{(t)} - \max_{j=y_i,r}|\langle \mathbf{w}_{j,r}^{(0)}, \xi_i \rangle| + 0.4\overline{\beta} \geq B_i^{(t)} - 1$, $\max_{j=y_i,r}\overline{\rho}_{j,r,i}^{(t)}$ will reach 2 within $T_1$ iterations. We can next verify that

$$T_1 = \frac{\log\left(20/(\sigma_0\sigma_p\sqrt{d})\right)4mn}{C_1 0.15^{q-2}\eta q\sigma_0^{q-2}(\sigma_p^2\sqrt{d})^q} \leq \frac{m}{\eta q 2^q(\sqrt{2\log(8m/\delta)})^{q-2}\sigma_0^{q-2}\|\boldsymbol{\mu}\|_2^q} = T_1^+/2,$$

where the inequality holds due to our SNR condition in (E.2). Therefore, by the definition of $T_1^{(i)}$, we have $T_1^{(i)} \leq T_1 \leq T_1^+/2$, where we use the non-decreasing property of $\overline{\rho}_{j,r,i}$. This completes the proof. $\qquad\square$

## E.2 Second Stage

By the signal-noise decompositon, at the end of the first stage, we have

$$\mathbf{w}_{j,r}^{(T_1)} = \mathbf{w}_{j,r}^{(0)} + j \cdot \gamma_{j,r}^{(T_1)} \cdot \frac{\boldsymbol{\mu}}{\|\boldsymbol{\mu}\|_2^2} + \sum_{i=1}^n \overline{\rho}_{j,r,i}^{(T_1)} \cdot \frac{\boldsymbol{\xi}_i}{\|\boldsymbol{\xi}_i\|_2^2} + \sum_{i=1}^n \underline{\rho}_{j,r,i}^{(T_1)} \cdot \frac{\boldsymbol{\xi}_i}{\|\boldsymbol{\xi}_i\|_2^2}$$

for $j \in \{\pm 1\}$ and $r \in [m]$. By the results we get in the first stage, we know that at the beginning of this stage, we have following property holds:

- $\max_r \overline{\rho}_{y_i,r,i}^{(T_1)} \geq 2$ for all $i \in [n]$.
- $\max_{j,r,i} |\underline{\rho}_{j,r,i}^{(T_1)}| = \widetilde{O}(\sigma_0\sigma_p\sqrt{d})$.
- $\max_{j,r} \gamma_{j,r}^{(T_1)} \leq \widehat{\beta}'$, where $\widehat{\beta}' = \widetilde{O}(\sigma_0\|\boldsymbol{\mu}\|_2)$.

Note that Lemma 5.1 implies that the learned noise $\overline{\rho}_{j,r,i}^{(t)}$ will not decrease, i.e., $\overline{\rho}_{j,r,i}^{(t+1)} \geq \overline{\rho}_{j,r,i}^{(t)}$. Therefore, for all data index $i$, we have $\max_r \overline{\rho}_{y_i,r,i}^{(t)} \geq 2$ for all $t \geq T_1$. Now we choose $\mathbf{W}^*$ as follows

$$\mathbf{w}_{j,r}^* = \mathbf{w}_{j,r}^{(0)} + 2qm\log(2q/\epsilon)) \left[ \sum_{i=1}^{n} \mathbb{1}(j = y_i) \cdot \frac{\boldsymbol{\xi}_i}{\|\boldsymbol{\xi}_i\|_2} \right].$$

Based on the definition of $\mathbf{W}^*$, we have the following lemma.

**Lemma E.3.** *Under the same conditions as Theorem 4.4, we have that* $\|\mathbf{W}^{(T_1)} - \mathbf{W}^*\|_F \leq \widetilde{O}(m^2 n^{1/2}\sigma_p^{-1}d^{-1/2})$.

*Proof of Lemma E.3.* We have

$$\|\mathbf{W}^{(T_1)} - \mathbf{W}^*\|_F \leq \|\mathbf{W}^{(T_1)} - \mathbf{W}^{(0)}\|_F + \|\mathbf{W}^{(0)} - \mathbf{W}^*\|_F$$

$$\leq \sum_{j,r} \gamma_{j,r}^{(T_1)}\|\boldsymbol{\mu}\|_2^{-1} + O(\sqrt{m}) \max_{j,r} \left\| \sum_{i=1}^{n} \overline{\rho}_{j,r,i}^{(T_1)} \cdot \frac{\boldsymbol{\xi}_i}{\|\boldsymbol{\xi}_i\|_2^2} + \sum_{i=1}^{n} \underline{\rho}_{j,r,i}^{(T_1)} \cdot \frac{\boldsymbol{\xi}_i}{\|\boldsymbol{\xi}_i\|_2^2} \right\|_2$$

$$+ O(m^{3/2}n^{1/2}\log(1/\epsilon)\sigma_p^{-1}d^{-1/2})$$

$$\leq \widetilde{O}(m^{3/2}n^{1/2}\sigma_p^{-1}d^{-1/2}),$$

where the first inequality is by triangle inequality, the second inequality is by our decomposition of $\mathbf{W}^{(T_1)}, \mathbf{W}^*$ and Lemma B.2 (notice that different noises are almost orthogonal), and the last inequality is by Proposition C.2 and Lemma E.2. This completes the proof. $\qquad\square$

**Lemma E.4.** *Under the same conditions as Theorem 4.4, we have that*

$$y_i\langle \nabla f(\mathbf{W}^{(t)}, \mathbf{x}_i), \mathbf{W}^* \rangle \geq q\log(2q/\epsilon)$$

*for all* $T_1 \leq t \leq T^*$.

*Proof of Lemma E.4.* Recall that $f(\mathbf{W}^{(t)}, \mathbf{x}_i) = (1/m)\sum_{j,r} j \cdot \left[ \sigma(\langle \mathbf{w}_{j,r}, y_i \cdot \boldsymbol{\mu}\rangle) + \sigma(\langle \mathbf{w}_{j,r}, \boldsymbol{\xi}_i\rangle) \right]$, so we have

$$y_i\langle \nabla f(\mathbf{W}^{(t)}, \mathbf{x}_i), \mathbf{W}^* \rangle$$

$$= \frac{1}{m}\sum_{j,r} \sigma'(\langle \mathbf{w}_{j,r}^{(t)}, y_i\boldsymbol{\mu}\rangle)\langle \boldsymbol{\mu}, j\mathbf{w}_{j,r}^*\rangle + \frac{1}{m}\sum_{j,r} \sigma'(\langle \mathbf{w}_{j,r}^{(t)}, \boldsymbol{\xi}_i\rangle)\langle y_i\boldsymbol{\xi}_i, j\mathbf{w}_{j,r}^*\rangle$$

$$= \frac{1}{m}\sum_{j,r}\sum_{i'=1}^{n} \sigma'(\langle \mathbf{w}_{j,r}^{(t)}, \boldsymbol{\xi}_i\rangle)2qm\log(2q/\epsilon) \mathbb{1}(j = y_{i'}) \cdot \frac{\langle \boldsymbol{\xi}_{i'}, \boldsymbol{\xi}_i\rangle}{\|\boldsymbol{\xi}_{i'}\|_2}$$

$$+ \frac{1}{m}\sum_{j,r} \sigma'(\langle \mathbf{w}_{j,r}^{(t)}, y_i\boldsymbol{\mu}\rangle)\langle \boldsymbol{\mu}, j\mathbf{w}_{j,r}^{(0)}\rangle + \frac{1}{m}\sum_{j,r} \sigma'(\langle \mathbf{w}_{j,r}^{(t)}, \boldsymbol{\xi}_i\rangle)\langle y_i\boldsymbol{\xi}_i, j\mathbf{w}_{j,r}^{(0)}\rangle$$

$$\geq \frac{1}{m}\sum_{j=y_i,r} \sigma'(\langle \mathbf{w}_{j,r}^{(t)}, \boldsymbol{\xi}_i\rangle)2qm\log(2q/\epsilon) - \frac{1}{m}\sum_{j,r}\sum_{i'\neq i} \sigma'(\langle \mathbf{w}_{j,r}^{(t)}, \boldsymbol{\xi}_i\rangle)2qm\log(2q/\epsilon)\frac{|\langle \boldsymbol{\xi}_{i'}, \boldsymbol{\xi}_i\rangle|}{\|\boldsymbol{\xi}_{i'}\|_2}$$

$$- \frac{1}{m}\sum_{j,r} \sigma'(\langle \mathbf{w}_{j,r}^{(t)}, y_i\boldsymbol{\mu}\rangle)\widetilde{O}(\sigma_0\|\boldsymbol{\mu}\|_2) - \frac{1}{m}\sum_{j,r} \sigma'(\langle \mathbf{w}_{j,r}^{(t)}, \boldsymbol{\xi}_i\rangle)\widetilde{O}(\sigma_0\sigma_p\sqrt{d})$$

$$\geq \frac{1}{m}\sum_{j=y_i,r} \sigma'(\langle \mathbf{w}_{j,r}^{(t)}, \boldsymbol{\xi}_i\rangle)2qm\log(2q/\epsilon) - \frac{1}{m}\sum_{j,r} \sigma'(\langle \mathbf{w}_{j,r}^{(t)}, \boldsymbol{\xi}_i\rangle)\widetilde{O}(mnd^{-1/2})$$

$$- \frac{1}{m}\sum_{j,r} \sigma'(\langle \mathbf{w}_{j,r}^{(t)}, y_i\boldsymbol{\mu}\rangle)\widetilde{O}(\sigma_0\|\boldsymbol{\mu}\|_2) - \frac{1}{m}\sum_{j,r} \sigma'(\langle \mathbf{w}_{j,r}^{(t)}, \boldsymbol{\xi}_i\rangle)\widetilde{O}(\sigma_0\sigma_p\sqrt{d}), \qquad (\text{E.5})$$

where the first inequality is by Lemma B.3 and the last inequality is by Lemma B.2. Next we will bound the inner-product terms in (D.4) respectively. By Lemma C.6, we have that

$$|\langle \mathbf{w}_{j,r}^{(t)}, y_i\boldsymbol{\mu}\rangle| \leq |\langle \mathbf{w}_{j,r}^{(0)}, y_i\boldsymbol{\mu}\rangle| + \gamma_{j,r}^{(t)} \leq \widetilde{O}(1), \qquad (\text{E.6})$$

where the last inequality is by Proposition C.2.

For $j = y_i$, we can bound the inner product between the parameter and the noise as follows

$$\max_{j,r} \langle \mathbf{w}_{j,r}^{(t)}, \boldsymbol{\xi}_i \rangle \geq \max_{j,r} \left[ \langle \mathbf{w}_{j,r}^{(0)}, \boldsymbol{\xi}_i \rangle + \overline{\rho}_{j,r,i}^{(t)} \right] - 8n\sqrt{\frac{\log(4n^2/\delta)}{d}} \alpha \geq 1, \tag{E.7}$$

where the first inequality is by Lemma C.4, the second inequality is by Lemma E.2.

For $j = -y_i$, we can bound the inner product between the parameter and the noise as follows

$$\langle \mathbf{w}_{j,r}^{(t)}, \boldsymbol{\xi}_i \rangle \leq \langle \mathbf{w}_{j,r}^{(0)}, \boldsymbol{\xi}_i \rangle + 8n\sqrt{\frac{\log(4n^2/\delta)}{d}} \alpha \leq 1, \tag{E.8}$$

where the first inequality is by Lemma C.5 and the last inequality is by Lemma B.3 and the conditions of $\sigma_0$ and $d$ in Condition 4.2. Therefore, plugging (E.6), (E.7), (E.8) into (E.5) gives

$$y_i \langle \nabla f(\mathbf{W}^{(t)}, \mathbf{x}_i), \mathbf{W}^* \rangle \geq 2q \log(2q/\epsilon)) - \widetilde{O}(mnd^{-1/2}) - \widetilde{O}(\sigma_0 \|\boldsymbol{\mu}\|_2) - \widetilde{O}(\sigma_0 \sigma_p \sqrt{d})$$
$$\geq q \log(2q/\epsilon),$$

where the last inequality is by $d \geq \widetilde{\Omega}(m^2 n^4)$ and $\sigma_0 \leq \widetilde{O}(m^{-2/(q-2)} n^{-1}) \cdot \min\{(\sigma_p \sqrt{d})^{-1}, \|\boldsymbol{\mu}\|_2^{-1}\}$ in Condition 4.2. $\qquad\square$

**Lemma E.5.** *Under the same conditions as Theorem 4.4, we have that*

$$\|\mathbf{W}^{(t)} - \mathbf{W}^*\|_F^2 - \|\mathbf{W}^{(t+1)} - \mathbf{W}^*\|_F^2 \geq (2q-1)\eta L_S(\mathbf{W}^{(t)}) - \eta\epsilon$$

*for all $T_1 \leq t \leq T^*$.*

*Proof of Lemma E.5.* The proof is exactly same as the proof of Lemma D.4.

$$\|\mathbf{W}^{(t)} - \mathbf{W}^*\|_F^2 - \|\mathbf{W}^{(t+1)} - \mathbf{W}^*\|_F^2$$
$$= 2\eta \langle \nabla L_S(\mathbf{W}^{(t)}), \mathbf{W}^{(t)} - \mathbf{W}^* \rangle - \eta^2 \|\nabla L_S(\mathbf{W}^{(t)})\|_F^2$$
$$= \frac{2\eta}{n} \sum_{i=1}^n \ell_i'^{(t)} [qy_i f(\mathbf{W}^{(t)}, \mathbf{x}_i) - \langle \nabla f(\mathbf{W}^{(t)}, \mathbf{x}_i), \mathbf{W}^* \rangle] - \eta^2 \|\nabla L_S(\mathbf{W}^{(t)})\|_F^2$$
$$\geq \frac{2\eta}{n} \sum_{i=1}^n \ell_i'^{(t)} [qy_i f(\mathbf{W}^{(t)}, \mathbf{x}_i) - q\log(6/\epsilon)] - \eta^2 \|\nabla L_S(\mathbf{W}^{(t)})\|_F^2$$
$$\geq \frac{2q\eta}{n} \sum_{i=1}^n [\ell(y_i f(\mathbf{W}^{(t)}, \mathbf{x}_i)) - \epsilon/(2q)] - \eta^2 \|\nabla L_S(\mathbf{W}^{(t)})\|_F^2$$
$$\geq (2q-1)\eta L_S(\mathbf{W}^{(t)}) - \eta\epsilon,$$

where the first inequality is by Lemma E.4, the second inequality is due to the convexity of the cross entropy function and the last inequality is due to Lemma C.7. $\qquad\square$

**Lemma E.6.** *Under the same conditions as Theorem 4.4, let $T = T_1 + \left\lfloor \frac{\|\mathbf{W}^{(T_1)} - \mathbf{W}^*\|_F^2}{2\eta\epsilon} \right\rfloor = T_1 + \widetilde{O}(\eta^{-1}\epsilon^{-1}m^3 nd^{-1}\sigma_p^{-2})$. Then we have $\max_{j,r} \gamma_{j,r}^{(t)} \leq 2\widehat{\beta}'$, $\max_{j,r,i} |\rho_{j,r,i}^{(t)}| = \widetilde{O}(\sigma_0 \sigma_p \sqrt{d})$ for all $T_1 \leq t \leq T$. Besides,*

$$\frac{1}{t - T_1 + 1} \sum_{s=T_1}^t L_S(\mathbf{W}^{(s)}) \leq \frac{\|\mathbf{W}^{(T_1)} - \mathbf{W}^*\|_F^2}{(2q-1)\eta(t - T_1 + 1)} + \frac{\epsilon}{(2q-1)}$$

*for all $T_1 \leq t \leq T$, and we can find an iterate with training loss smaller than $\epsilon$ within $T$ iterations.*

*Proof of Lemma E.6.* By Lemma E.5, for any $T_1 \leq t \leq T$, we obtain that

$$\|\mathbf{W}^{(s)} - \mathbf{W}^*\|_F^2 - \|\mathbf{W}^{(s+1)} - \mathbf{W}^*\|_F^2 \geq (2q-1)\eta L_S(\mathbf{W}^{(s)}) - \eta\epsilon \tag{E.9}$$

holds for $T_1 \leq s \leq t$. Taking a summation, we have that

$$\sum_{s=T_1}^{t} L_S(\mathbf{W}^{(s)}) \leq \frac{\|\mathbf{W}^{(T_1)} - \mathbf{W}^*\|_F^2 + \eta\epsilon(t - T_1 + 1)}{(2q - 1)\eta}$$

$$\overset{(i)}{\leq} \frac{2\|\mathbf{W}^{(T_1)} - \mathbf{W}^*\|_F^2}{(2q - 1)\eta}$$

$$\overset{(ii)}{=} \widetilde{O}(\eta^{-1}m^3nd^{-1}\sigma_p^{-2}), \tag{E.10}$$

where (i) is by $t \leq T_2$ and (ii) is by Lemma E.3 Then we can use induction to prove that $\max_{j,r} \gamma_{j,r}^{(t)} \leq 2\widehat{\beta}'$ for all $t \in [T_1, T]$. Clearly, by the definition of $\widehat{\beta}'$, we have $\max_{j,r} \gamma_{j,r}^{(T_1)} \leq \widehat{\beta}' \leq 2\widehat{\beta}'$. Now suppose that there exists $\widetilde{T} \in [T_1, T]$ such that $\max_{j,r} \gamma_{j,r}^{(t)} \leq 2\widehat{\beta}'$ for all $t \in [T_1, \widetilde{T} - 1]$. Then by (C.3), we have

$$\gamma_{j,r}^{(\widetilde{T})} = \gamma_{j,r}^{(T_1)} - \frac{\eta}{nm} \sum_{s=T_1}^{\widetilde{T}-1} \sum_{i=1}^{n} \ell_i'^{(t)} \cdot \sigma'(\langle \mathbf{w}_{j,r}^{(t)}, y_i \cdot \boldsymbol{\mu} \rangle)\|\boldsymbol{\mu}\|_2^2,$$

$$\overset{(i)}{\leq} \gamma_{j,r}^{(T_1)} + \frac{q3^{q-1}\eta}{nm}\|\boldsymbol{\mu}\|_2^2\widehat{\beta}'^{q-1} \sum_{s=T_1}^{\widetilde{T}-1} \sum_{i=1}^{n} |\ell_i'^{(t)}|$$

$$\overset{(ii)}{\leq} \gamma_{j,r}^{(T_1)} + q3^{q-1}\eta m^{-1}\|\boldsymbol{\mu}\|_2^2\widehat{\beta}'^{q-1} \sum_{s=T_1}^{\widetilde{T}-1} L_S(\mathbf{W}^{(s)})$$

$$\overset{(iii)}{\leq} \gamma_{j,r}^{(T_1)} + \widehat{\beta}'^{q-1}\widetilde{O}(m^2n\mathrm{SNR}^2)$$

$$\overset{(iv)}{\leq} \gamma_{j,r}^{(T_1)} + \widehat{\beta}'^{q-1}\widetilde{O}(m^2n^{1-2/q})$$

$$\overset{(v)}{\leq} 2\widehat{\beta}'$$

for all $j \in \{\pm 1\}$ and $r \in [m]$, where (i) is by induction hypothesis $\max_{j,r} \gamma_{j,r}^{(t)} \leq 2\widehat{\beta}'$, (ii) is by $|\ell'| \leq \ell$, (iii) is by (E.10), (iv) is by $n^{-1}\mathrm{SNR}^{-q} \geq \widetilde{\Omega}(1)$, and $(v)$ is by $\widehat{\beta}' = \widetilde{O}(\sigma_0\|\boldsymbol{\mu}\|_2)$ and $\widehat{\beta}'^{q-2}\widetilde{O}(m^2n^{1-2/q}) = \widetilde{O}(m^2n^{1-2/q}(\sigma_0\|\boldsymbol{\mu}\|_2)^{q-2}) \leq 1$ by Condition 4.2. Therefore, we have $\max_{j,r} \gamma_{j,r}^{(\widetilde{T})} \leq 2\widehat{\beta}'$, which completes the induction. $\qquad\square$

### E.3 Population Loss

**Lemma E.7** (4th statement of Theorem 4.4). *Under the same conditions as Theorem 4.4, within $\widetilde{O}(\eta^{-1}n\sigma_0^{2-q}\sigma_p^{-q}d^{-q/2} + \eta^{-1}\epsilon^{-1}m^3n\sigma_p^{-2}d^{-1})$ iterations, we can find $\mathbf{W}^{(T)}$ such that $L_S(\mathbf{W}^{(T)}) \leq \epsilon$. Besides, for any $0 \leq t \leq T$ we have that $L_{\mathcal{D}}(\mathbf{W}^{(t)}) \geq 0.1$.*

*Proof of Lemma E.7.* Given a new example $(x, y)$, we have that

$$\|\mathbf{w}_{j,r}^{(t)}\|_2 = \left\| \mathbf{w}_{j,r}^{(0)} + j \cdot \gamma_{j,r}^{(t)} \cdot \frac{\boldsymbol{\mu}}{\|\boldsymbol{\mu}\|_2^2} + \sum_{i=1}^{n} \overline{\rho}_{j,r,i}^{(t)} \cdot \frac{\boldsymbol{\xi}_i}{\|\boldsymbol{\xi}_i\|_2^2} + \sum_{i=1}^{n} \underline{\rho}_{j,r,i}^{(t)} \cdot \frac{\boldsymbol{\xi}_i}{\|\boldsymbol{\xi}_i\|_2^2} \right\|_2$$

$$\overset{(i)}{\leq} \|\mathbf{w}_{j,r}^{(0)}\|_2 + \frac{\gamma_{j,r}^{(t)}}{\|\boldsymbol{\mu}\|_2} + \sum_{i=1}^{n} \frac{\overline{\rho}_{j,r,i}^{(t)}}{\|\boldsymbol{\xi}_i\|_2} + \sum_{i=1}^{n} \frac{|\underline{\rho}_{j,r,i}^{(t)}|}{\|\boldsymbol{\xi}_i\|_2}$$

$$\overset{(ii)}{=} O(\sigma_0\sqrt{d}) + \widetilde{O}(n\sigma_p^{-1}d^{-1/2}),$$

where (i) is by triangle inequality and (ii) is by $\max_{j,r} \gamma_{j,r}^{(t)} = \widetilde{O}(\sigma_0\|\boldsymbol{\mu}\|_2)$ in Lemma E.6 and $\max_{i,j,r} |\rho_{j,r,i}| \leq 4\log(T^*)$ in Proposition 5.3.

Therefore, we have that $\langle \mathbf{w}_{j,r}^{(t)}, \boldsymbol{\xi} \rangle \sim \mathcal{N}(0, \sigma_p^2\|\mathbf{w}_{j,r}^{(t)}\|_2^2)$. So with probability $1 - 1/(4m)$,

$$|\langle \mathbf{w}_{j,r}^{(t)}, \boldsymbol{\xi} \rangle| \leq \widetilde{O}(\sigma_0\sigma_p\sqrt{d} + nd^{-1/2}).$$

Since the signal vector $\boldsymbol{\mu}$ is orthogonal to noises, by $\max_{j,r} \gamma_{j,r}^{(t)} \leq 2\widehat{\beta}' = \widetilde{O}(\sigma_0 \|\boldsymbol{\mu}\|_2)$ in Lemma E.6, we also have that $|\langle \mathbf{w}_{j,r}^{(t)}, \boldsymbol{\mu} \rangle| \leq |\langle \mathbf{w}_{j,r}^{(0)}, y_i \boldsymbol{\mu} \rangle| + \gamma_{j,r}^{(t)} = \widetilde{O}(\sigma_0 \|\boldsymbol{\mu}\|_2)$. Now by union bound, with probability at least $1 - 1/2$, we have that

$$
\begin{aligned}
F_j(\mathbf{W}_j^{(t)}, \mathbf{x}) &= \frac{1}{m} \sum_{r=1}^{m} \sigma(\langle \mathbf{w}_{j,r}^{(t)}, y\boldsymbol{\mu} \rangle) + \frac{1}{m} \sum_{r=1}^{m} \sigma(\langle \mathbf{w}_{j,r}^{(t)}, \boldsymbol{\xi} \rangle) \\
&\leq \max_r |\langle \mathbf{w}_{j,r}^{(t)}, \boldsymbol{\mu} \rangle|^q + \max_r |\langle \mathbf{w}_{j,r}^{(t)}, \boldsymbol{\xi} \rangle|^q \\
&\leq \widetilde{O}(\sigma_0^q \sigma_p^q d^{q/2} + n^q d^{-q/2} + \sigma_0^q \|\boldsymbol{\mu}\|_2^2) \\
&\leq 1,
\end{aligned}
$$

where the last inequality is by $\sigma_0 \leq \widetilde{O}(n^{-1} m^{-2/(q-2)} \cdot \min\{(\sigma_p \sqrt{d})^{-1}, \|\boldsymbol{\mu}\|_2^{-1}\})$ and $d \geq \widetilde{\Omega}(m^2 n^4)$ in Condition 4.2. Therefore, with probability at least $1 - 1/2$, we have that

$$
\ell\big(y \cdot f(\mathbf{W}^{(t)}, \mathbf{x})\big) \geq \log(1 + e^{-1}).
$$

Thus $L_{\mathcal{D}}(\mathbf{W}^{(t)}) \geq \log(1 + e^{-1}) \cdot 0.5 \geq 0.1$. This completes the proof. $\qquad\square$

## F   Experiments

We present simulations on synthetic data and experiments on real-world data to back up our theoretical analysis. The code and data for our experiments can be found on Github [‡].

**Synthetic-data experiments.**   Here we generate synthetic data exactly following Definition 3.1. Specifically, we set dimension $d = 400$. Since the learning problem is rotation-invariant, without loss of generality, we set $\boldsymbol{\mu} \mathbin{/\mkern-5mu/} [1, 0, \ldots, 0]^\top$. We then generate the noise vector $\xi$ in Definition 3.1 so that its first entry is zero, while the rest of the entries are standard Gaussian random vectors. In this way, to perform experiments under different SNRs, it suffices to change $\|\mu\|_2$, or equivalently, change the first entry of $\boldsymbol{\mu}$.

We train a two-layer CNN model defined in Section 3 with RELU[3] activation function. The number of filters is set as $m = 10$. We use the default initialization method in PyTorch to initialize the CNN parameters, and train the CNN with full-batch gradient descent with a learning rate of $0.01$ for $50$ epochs. We consider different training data sizes $n$ ranging from 1 to 100, and different SNRs ranging from 0 to 1.0. Note that in all these training sample size and SNR settings, our training setup can guarantee a training loss smaller than 0.05. After training, we also calculate the test losses for each case using 100 test data points. The results are given in Figure 2.

It is clear that the results shown in Figure 2 match our theoretical results very well: the test loss values depend on both sample size and SNR, and a larger sample size or a higher SNR can both lead to smaller test losses. Moreover, Figure 2 shows a clear phase transition between benign and harmful overfitting, and is consistent with our illustration in Figure 1.

**Real-data experiments.**   We further conduct real-world experiments on the MNIST data set (Deng, 2012), which consists of gray-scale hand-written digits of size $28 \times 28$. Note that for a given data set, we cannot accurately define the SNR, as it is not clear which part of the image is signal or noise. Therefore, in order to verify our theory, for each image, we first multiply each pixel in the image with a factor which we call "scaled SNR", and then add standard Gaussian random noises to the outer regions with a width of 5. In this way, we can roughly use the scaled SNR to represent the signal-to-noise ratio in the data. Two examples of the modified images with scaled SNRs 1 and 0.2 are given in Figure 3.

We train a simple ReLU CNN model which has two convolutional layers each followed by a max-pooling layer, and a fully-connected layer that gives the final output of the network. The first convolutional layer has 32 output channels (i.e., 32 different filters), with filter size 5, stride 1 and padding 0. The second convolutional layer has 64 output channels, with filter size 3, stride 1 and padding 0. Both max-pooling layers are of size 2 and stride 2.

---

[‡]https://github.com/uclaml/Benign-Overfitting-CNN

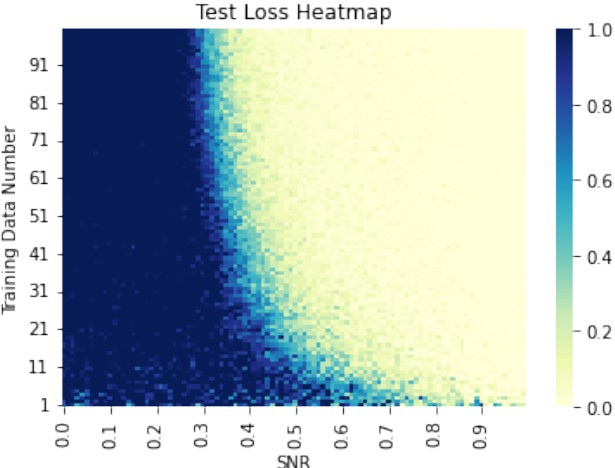

Figure 2: Heatmap of test losses on synthetic data under different training data sizes and SNRs. High test losses are marked in blue and low test losses are marked in yellow.

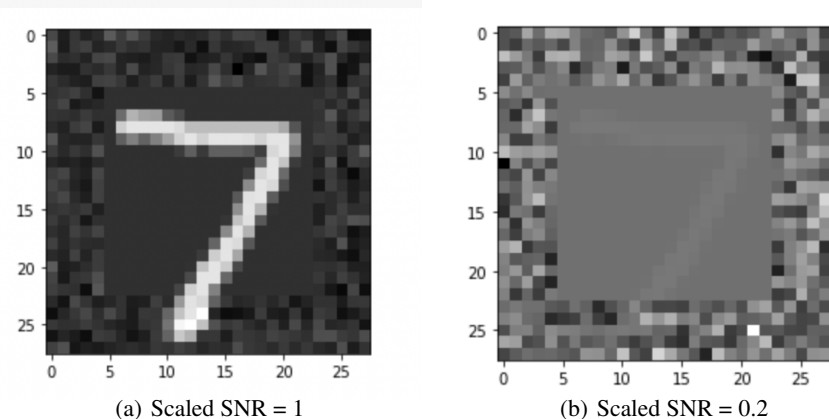

(a) Scaled SNR = 1         (b) Scaled SNR = 0.2

Figure 3: Illustration of modified MNIST images. (a) shows an example of a modified digit 7 with Scaled SNR = 1, and (b) shows the same image with Scaled SNR = 0.2.

We train the network with mini-batch stochastic gradient descent. We set the mini-batch size to be 128, and the learning rate to be 0.1. The network is trained for 2000 epochs in each setting. We consider different training data sizes $n$ ranging from 3000 to 7500, and different scaled SNRs ranging from 0.01 to 0.2. Again, in all these training sample size and SNR settings, our training setup can guarantee a training loss smaller than 0.05. After training, we calculate the test losses for each case using the 10000 test data (modified in the same way as training data). The results are given in Figure 4.

Clearly, the results in Figure 4 also match our theoretical results very well, and show a phase transition between benign and harmful overfitting. Note that for this set of experiments, the setup does not satisfy our assumptions in many aspects:

- The CNN has two convolution layers and two max-pooling layers.

- The convolutions are with stride 1 and therefore the patches in the data have overlaps. This also implies that the noise patches in an image are not independent.

- The activation function is ReLU instead of $\mathrm{ReLU}^q$ with $q > 2$.

- The noise patches and signal patches are not orthogonal to each other, and are sometimes mixed together, as is shown in Figure 3.

Nevertheless, the experiment results still corroborate our theory to a certain extent. This demonstrates that our study of benign and harmful overfitting in CNNs captures the nature of real-world image classification problems.

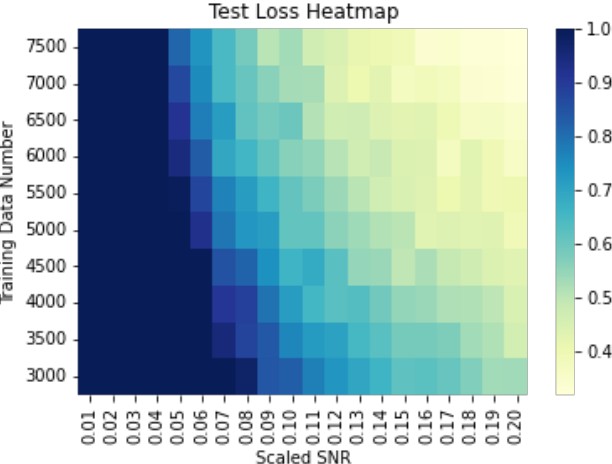

Figure 4: Heatmap of test losses on modified MNIST images under different training data sizes and scaled SNRs. High test losses are marked in blue and low test losses are marked in yellow.