# OpenReview forum: "Benign Overfitting in Two-layer Convolutional Neural Networks"
_NeurIPS.cc/2022/Conference — NeurIPS 2022 Accept_

### Official Review · Reviewer_sf8Q · 2022-06-20

**Rating:** 7
**Confidence:** 4
**Soundness:** 3 good
**Presentation:** 3 good
**Contribution:** 3 good

**Summary:**

Paper studies the benign overfitting phenomenon in training a two-layer convolutional neural network (CNN).
Authors show that when the signal-to-noise ratio satisfies a certain condition, a two-layer CNN trained by gradient descent can achieve arbitrarily small training and test loss. Otherwise, overfitting becomes harmful and the obtained CNN can only achieve constant level test loss.
Then paper shows a sharp phase transition between benign overfitting and harmful overfitting, driven by the signal-to-noise ratio.

**Questions:**

It would be nice to have some hint to a general property of benign overfitting.
How much do you think this result is general and holds for a wider class of function?

**Limitations:**

Some practical example would improved the quality of the paper.

**Strengths And Weaknesses:**

Strengths
- intriguing property
- experiments convincing

Weaknesses
- theoretical analysis is very technical and hard to follow
- some practical example would improved the paper

---

> ### Author Response · Authors · 2022-08-02
> **Response to Reviewer sf8Q**
>
> Thanks for your supportive comments. Below are our answers to your questions.
>
> Q1. theoretical analysis is very technical and hard to follow
>
> A1. Thanks for pointing it out. We have revised the proofs in the paper to make them easier to follow.
>
> Q2. some practical example would improved the paper
>
> A2. Thanks for your suggestions. In Appendix F in the revised supplementary materials, we have added experiments on both synthetic and real data to demonstrate that our theoretical results match the practice well.
>
> Q3. It would be nice to have some hint to a general property of benign overfitting. How much do you think this result is general and holds for a wider class of function?
>
> A3. The benign overfitting phenomenon has been observed in various practical settings, especially in the practice of training neural networks (Zhang et al., (2017)). However, existing theoretical studies on this phenomenon mostly focus on simplified settings such as linear models. We believe extending the theory of benign overfitting to more general settings is an important future work direction, and our work serves as a foundation towards more practical results.

---

### Official Review · Reviewer_s7mX · 2022-07-12

**Rating:** 7
**Confidence:** 4
**Soundness:** 3 good
**Presentation:** 4 excellent
**Contribution:** 2 fair

**Summary:**

This paper characterizes when a convolutional neural network with a certain architecture and on a certain data model generalizes well while fitting the data perfectly. The analysis uses a signal-to-noise (SNR) ratio and shows that harmful overfitting can be avoided with a small sample size if SNR is large enough.

**Questions:**

See weaknesses section

**Limitations:**

See weaknesses section

**Strengths And Weaknesses:**

The paper attempts at an important problem and analyzes the performance of a neural network without approximating it with a convex linear model. I have not check the proofs in the appendix but the math in the main paper is correct and well explained. The two step approach in the analysis might have applications beyond the current problem. Nevertheless I am leaning towards rejection since this machinery relies on the following restrictive assumptions on both the data and the network:

1. The images have a noise patch and a signal patch. Each feature corresponds to either signal or noise through the dataset unlike a real dataset where the signal can be anywhere in the image. The label in the studied data model is a fixed vector $\mu$ for the positive class and $-\mu$ for the negative class and the noise patch is a multi-dimensional Gaussian r.v., while in a typical dataset signal and noise are entwined. Does this theory extend to a more realistic data model e.g. by replacing $\mu$ and $-\mu$ with the mean of feature vectors in the positive and negative class?

2. In the equation below line 140 it appears that the network's first layer processes the signal and noise separately, that is, it applies the filter on the first patch and then strides to the second patch and applies the filter on it. This large value of stride does not represent any common CNN architecture. A typical CNN would slide over the image in more fine-grained steps.

3. The second layers weights are initialized to $\pm 1$ and are assumed to be constant through the training. The hidden layer activation is polynomial ReLU with degree at least 2.

4. An important motivation for this work is that removing the large hidden layer width assumption in NTK. The analysis in the submission, however, still requires an overparametrization assumption (Condition 4.2). From the omega notation in the main paper it is not clear how restrictive this requirement is and how large the input and hidden layer width need to be for the results to hold.

Minor comments:
- In Line 305 the claim that small variations in loss across data points at the beginning of training matches the behavior of neural networks in practice needs citation.
----------------------------------------------
Update: Thanks for the response. The clarifications in the rebuttal and the discussion with reviewer UZNG answered my first comment. The added experiments also show a consistency between the trend shown by theory and the one that occurs in a realistic setting without any of the unrealistic assumptions.

Regarding comment 4: My understanding is that large-width neural networks (at least some fully-connected ones) can be _well approximated_ with the NTK even if their width is finite. See theorem 3.2 in [1] for example. It is likely that under condition 4.2 one can directly analyze the NTK and then bound the difference between the behavior of a wide convolutional neural network and the NTK without the other assumptions.

Regarding asymptotic notation: The omega notation is completely fine. I was only admitting that, due to the omega notation, I do not have a clear idea if Condition 4.2 makes the behavior of the network close to NTK or not.

[1] Arora, Sanjeev, et al. "On exact computation with an infinitely wide neural net." NeurIPS 2019.

---

> ### Author Response · Authors · 2022-08-02
> **Response to Reviewer s7mX**
>
> Thanks for your helpful comments. We have added experiments and revised the paper to address your concerns. Below are our answers to your questions.
>
> Q1. The images have a noise patch and a signal patch. Each feature corresponds to either signal or noise through the dataset unlike a real dataset where the signal can be anywhere in the image.
>
> A1. This is a misunderstanding. We would like to clarify that the locations of signal in the data inputs are not fixed. For example, in data point #1, the first patch may be the signal and the second patch may be noise, while in another data point #2, probably the second patch is the signal and the first patch is the noise. Therefore, our data model captures the nature of image data that signal can be anywhere in the image. We have clarified this in the revision.
>
> Q2. The label in the studied data model is a fixed vector $\boldsymbol{\mu}$ for the positive class and $-\boldsymbol{\mu}$ for the negative class and the noise patch is a multi-dimensional Gaussian r.v., while in a typical dataset signal and noise are entwined. Does this theory extend to a more realistic data model e.g. by replacing  $\boldsymbol{\mu}$ and  $-\boldsymbol{\mu}$ with the mean of feature vectors in the positive and negative class?
>
> A2. We would like to point out that assuming the features of the two classes to be $\boldsymbol{\mu}$ and $-\boldsymbol{\mu}$ is also natural, as the data are usually centered to have zero mean in practice. Our theory also applies to the setting where the features of the two classes of data are two orthogonal vectors $\boldsymbol{\mu}_1$ and $\boldsymbol{\mu}_2$. Under this new setting with two signal vectors $\boldsymbol{\mu}_1$ and $\boldsymbol{\mu}_2$, our theoretical results would be the same.
>
> Q3. In the equation below line 140 it appears that the network's first layer processes the signal and noise separately, that is, it applies the filter on the first patch and then strides to the second patch and applies the filter on it. This large value of stride does not represent any common CNN architecture. A typical CNN would slide over the image in more fine-grained steps.
>
> A3. Thanks for pointing it out. We agree that assuming a large stride is a limitation of our analysis. However, this is also the case for most existing theoretical studies of CNNs (Allen-Zhu and Li (2020b), Zou et al (2021a)), and benign overfitting for CNNs is not well-studied even under this large stride assumption. We believe our work is a cornerstone of this line of research, and serves as a foundation towards more practical results. In Appendix F in the revised supplementary materials, we have added real-data experiments to show that even for multi-layer CNNs with stride $1$, the experiment results, to a certain extent, still corroborate our theory.
>
> Q4. An important motivation for this work is that removing the large hidden layer width assumption in NTK. The analysis in the submission, however, still requires an overparametrization assumption (Condition 4.2).
>
> A4. This is also a misunderstanding. Over-parameterization does not necessarily imply that the training of neural networks is in the NTK regime. Moreover, our motivation is not to remove the large hidden layer width assumption in NTK. Instead, even though the CNNs we study are over-parameterized, we can still demonstrate that the learning regime studied in this paper is beyond the NTK regime. Moreover, we would like to point out that over-parameterization is naturally required since our goal is to study benign and harmful overfitting. Without over-parameterization, we cannot guarantee overfitting.
>
> Q5. From the omega notation in the main paper it is not clear how restrictive this requirement is and how large the input and hidden layer width need to be for the results to hold.
>
> A5. We would like to point out that the asymptotic notations are standard in theoretical analyses. In Appendix F in the revised supplementary materials, we use experiments on both synthetic and real data to demonstrate that the requirements are not restrictive and can cover practical settings.

---

### Official Review · Reviewer_UZNG · 2022-07-14

**Rating:** 5
**Confidence:** 3
**Soundness:** 3 good
**Presentation:** 3 good
**Contribution:** 3 good

**Summary:**

The paper studies the training dynamics and generalization behavior of GD on two-layer CNNs. In particular, it proves that when the signal-to noise ratio is large enough, GD can achieve small training and test loss; otherwise, the test loss may be large and bad overfitting occurs. Together, it shows a phase transition between benign and harmful overfitting.


**Questions:**

If the position of the signal is not fixed, or if the signal and noise is not separable in the CNN model, will the result keep the same?

**Limitations:**

Yes.

**Strengths And Weaknesses:**

The studied problem is important in understanding the generalization power of neural networks, and the result is relatively complete for the phase transition demonstrated. The technical analysis of the GD dynamics looks solid, and the decomposition approach may potentially be applied to other settings.

I have some questions in the setting of the studied problem. In Definition 3.1, the data point is composed of $x_1$ and $x_2$ representing signal and noise, separately. In this definition, the position of the signal is either $x_1$ or $x_2$, indicating that it is fixed in one of the two positions. Moreover, in the expression of CNN (Line 140), $x_1$ and $x_2$ seems to be separable, so the signal and noise seems to be separable in the setting. I am not quite sure whether this is reasonable in practice, as the position of signal usually varies (for example, the “signal” of an image may be in any corner of the image).

Another minor issue: In Section 3, the subscript of $x$ refer to both the category of signal or noise (with subscript 1, 2) and the index of data (with subscript 1, …, n). This may cause confusion.

---

> ### Author Response · Authors · 2022-08-02
> **Response to Reviewer UZNG**
>
> Thanks for your positive comments. We address your questions as follows.
>
> Q1  "In this definition, the position of the signal is either $x_1$ or $x_2$, indicating that it is fixed in one of the two positions. Moreover, in the expression of CNN (Line 140), $x_1$ and $x_2$ seems to be separable." "If the position of the signal is not fixed, or if the signal and noise is not separable in the CNN model, will the result keep the same?"
>
> A1 This is a misunderstanding. We would like to clarify that the locations of signal in the data inputs are not fixed. For example, in data point #1, the first patch may be the signal and the second patch may be noise, and in data point #2, probably the second patch is the signal and the first patch is the noise. Therefore, our data model captures the nature of image data that position of signal usually varies. We have clarified this in the revision.
>
> Q2  In Section 3, the subscript of $\mathbf{x}$ refer to both the category of signal or noise (with subscript 1, 2) and the index of data (with subscript 1, …, n). This may cause confusion.
>
> A2 Thanks for pointing it out. We have changed notations to avoid confusion. We now use superscripts $\mathbf{x}^{(1)}$ and  $\mathbf{x}^{(2)}$ to denote the two patches in a data input.

---

> > ### Comment · Reviewer_UZNG · 2022-08-07
> > **Clarification on Q1**
> >
> > Thank you for your reply. I'd like to make some clarifications on Q1.
> >
> > I understand that the location is not fixed and I understand the example that the author gives. However, my concern is that the signal seems to be in **either the first patch or the second patch**, whereas my expectation for "not fixed" is that the signal can be **anywhere in the two patches**. For example, half of the signal is in the first patch and half is in the second patch. Is such case included in your results? If not, what are the expected results in this case?
> >
> > The intuition behind this question is that your CNN model processes the two patches separately. If the signal and the noise are in different patches, then this implies that your model processes signal and noise separately. This is a rather strong assumption and does not match practice in my opinion.
> >
> > I notice that Reviewer s7mx asked similar questions, and I think I share the same understandings and opinions with him/her in this question. Could you please address more about it?

---

> > > ### Author Response · Authors · 2022-08-07
> > > **Re: Clarification on Q1**
> > >
> > > Thank you for your follow-up questions.
> > >
> > > Our analysis can cover the case that the signal appears in multiple patches. For example, for the case you mentioned where half of the signal is in one patch and the other half is in another, it can be covered by our analysis with slight modification. This is because our theory can be directly generalized to the case where the data contains multiple signal patches. When the signal $\mu$ fall into two patches, it will break down into two parts $\mu_{1}$ and $\mu_{2}$ that fall into different **signal patches**. By pigeonhole principle, at least one of $\lVert \mu_{1} \rVert_{2}$ and $\lVert\mu_{2}\rVert_{2} $ is of the same order as $\lVert\mu\rVert_{2}$ $\big(\lVert\mu\rVert_{2}^{2} = \lVert\mu_{1}\rVert_{2}^{2} + \lVert\mu_{2}\rVert_{2}^{2} \big)$, the SNR will only change by a constant order (In this case, at most $\sqrt{2}$). Therefore, the phase transition results in Theorem 4.3, 4.4 will still hold while only the constants hidden in the $\tilde{\Omega}$ and $\tilde{O}$ notation will change.
> > >
> > >
> > > Our analysis can be easily modified to handle the setting where the signal patch also contains noises (i.e., the signal and the noise co-occur in the same patch). The proof is still based on the signal-noise decomposition. The difference is that in this case, we need to add $\rho_{j, r, i, p}\cdot \xi_{i, p}$ in Decomposition (4.1) where $\xi_{i, p}$ indicates different noises in the different patches (can be in the signal patch) and $\rho_{j, r, i, p}$ are the corresponding coefficients. Besides, there will be multiple terms $\langle w_{j,r}^{(t)}, y_{i}\cdot\mu\rangle + \langle w_{j,r}^{(t)}, y_{i}\cdot\xi_{i, p}\rangle$ inside the activation function in equation (5.2). Nevertheless, this difference will not affect our analysis of signal learning and noise memorization. For example, in the benign overfitting regime, a modified version of Lemma 5.5 can still hold where we need to upper bound noise coefficient $\rho_{j,r, i,p}$ instead of $\rho_{j, r, i}$. When noises appear in all $P$ patches, the resulting SNR will also depend on $P$. By modifying the subsequent analysis accordingly, we can still show how a CNN learns signals or memorizes noises during the gradient descent trajectory as well as the phase transition phenomenon.
> > >
> > > Since this is the first paper towards understanding the benign overfitting in CNNs, we choose to avoid introducing unnecessary complexities in the data/CNN models and the analysis to make the result easy to follow. This will also benefit the readers who would like to use our key proof technique to analyze their own problems. In the future, we will extend our analysis to more complicated data/CNN models.

---

### Official Review · Reviewer_AJsJ · 2022-07-16

**Rating:** 7
**Confidence:** 3
**Soundness:** 3 good
**Presentation:** 3 good
**Contribution:** 3 good

**Summary:**

This manuscript analyses the benign overfitting phenomenon for deep neural networks. In detail, the authors decompose the signal information and the noise during the training process and derive the corresponding convergence result according to the signal-to-noise-ratio (SNR). According to the theorems, with smaller SNR, the noise will be dropped and the harmful overfitting converts to benign overfitting.

**Questions:**

Please see Weaknesses.

**Ethics Review Area:**

["I don’t know"]

**Limitations:**

Please see Weaknesses.

**Strengths And Weaknesses:**

Pros:

- This manuscript deeply investigates benign overfitting and clearly show the transition boundary between harmful overfitting and benign overfitting;

- The signal-noise decomposition is novel and can promote analysis in deep learning theory;

- The paper is well-writting and easy to follow.

Cons:

This paper comprehensively analyse the phase transition of benign overftting via SNR. However, there are two major points that can undermine the contributions of the paper: (1) Lack of ancillary experiments. The results are fully theoretical and rely on many assumptions (Condition 4.2) and Figure 1 is the schematic diagram according to theoretical results. If there are some empirical study based on real models and datasets, the conclusion will be more convincing; (2) Strong assumptions. The paper relys on many assumptions on dimiension, sample size, parameter size, etc. I recommend the authors to apply the assumtions to real empirical situation and show how the assumptions close to the real scenarios.

Some minor writing suggestions:
- Line 23: "the the classical" $\rightarrow$ "the classical";
- Line 128: "signal-to-noise ratio"should be followed by the abbr. SNR for better understandingl;
- Line 201: "that$n^{-1}$" $\rightarrow$ "that $n^{-1}$";
- Line 211: ",$n$SNR" $\rightarrow$ ", $n$SRN";

The reason for socre: Though the paper lacks empircal study, I appreciate for its solid theoretical analysis on the transition boundary between harmful and benigh overfitting. The transion boundary will be an important prespective to understand the success of neural nets.

---

> ### Author Response · Authors · 2022-08-02
> **Response to Reviewer AJsJ**
>
> Thanks for your positive comments! We address your questions and concerns as follows.
>
> Q1. Lack of ancillary experiments.
>
> A1. Thanks for pointing it out. We have added simulation results to justify our theory in Appendix F in the revised supplementary materials. Our experiment results match our theory well and demonstrate the phase transition between signal learning and noise memorization on both synthetic and real data.
>
> Q2. Strong assumptions. The paper relys on many assumptions on dimiension, sample size, parameter size, etc. I recommend the authors to apply the assumptions to real empirical situations and show how the assumptions close to the real scenarios.
>
> A2. The assumptions in this paper are all common assumptions made in previous studies of benign overfitting. In Appendix F in the revised supplementary materials, we have added real-data experiments to show that our theoretical results hold under practical settings.

---

### Meta-Review · Area_Chair_MjPV · 2022-08-29

**Recommendation:** Accept
**Confidence:** Certain

**Metareview:**

This paper studies the benign overfitting phenomenon in training a two-layer convolutional neural network (CNN). Most importantly, this paper introduces SNR ratio in the analysis of benign overfitting. Under Condition 4.2, this paper provides the phase transition regime between "benign overfitting" and "harmful overfitting" in terms of SNR. A common concern from reviewers is the lack of connection to the practical settings. Still, all the reviewers agree to accept. I also think the theoretical result of this paper is interesting to the community, and I recommend accept.


**Award:**

No

---

### Decision · Program_Chairs · 2022-09-14

Accept